# Scale-teaching: Robust Multi-scale Training for Time Series Classification with Noisy Labels

**Zhen Liu**
South China University of Technology
Guangzhou, China
`cszhenliu@mail.scut.edu.cn`

**Peitian Ma**
South China University of Technology
Guangzhou, China
`ma_scuter@163.com`

**Dongliang Chen**
South China University of Technology
Guangzhou, China
`ytucdl@foxmail.com`

**Wenbin Pei**
Dalian University of Technology
Dalian, China
`peiwenbin@dlut.edu.cn`

**Qianli Ma**[*]
South China University of Technology
Guangzhou, China
`qianlima@scut.edu.cn`

## Abstract

Deep Neural Networks (DNNs) have been criticized because they easily overfit noisy (incorrect) labels. To improve the robustness of DNNs, existing methods for image data regard samples with small training losses as correctly labeled data (small-loss criterion). Nevertheless, time series' discriminative patterns are easily distorted by external noises (i.e., frequency perturbations) during the recording process. This results in training losses of some time series samples that do not meet the small-loss criterion. Therefore, this paper proposes a deep learning paradigm called *Scale-teaching* to cope with time series noisy labels. Specifically, we design a fine-to-coarse cross-scale fusion mechanism for learning discriminative patterns by utilizing time series at different scales to train multiple DNNs simultaneously. Meanwhile, each network is trained in a cross-teaching manner by using complementary information from different scales to select small-loss samples as clean labels. For unselected large-loss samples, we introduce multi-scale embedding graph learning via label propagation to correct their labels by using selected clean samples. Experiments on multiple benchmark time series datasets demonstrate the superiority of the proposed Scale-teaching paradigm over state-of-the-art methods in terms of effectiveness and robustness.

## 1 Introduction

Time series classification has recently received much attention in deep learning [1, 2]. Essentially, the success of Deep Neural Networks (DNNs) is driven by a large amount of well-labeled data. However, human errors [3] and sensor failures [4] produce noisy (incorrect) labels in time series datasets. For example, in electrocardiogram diagnosis [5], physicians with different experiences tend to make inconsistent category judgments. In recent studies [6, 7], DNNs have shown their powerful learning ability, which, however, makes it relatively easier to overfit noisy labels and inevitably degenerate

---

[*]Qianli Ma is the corresponding author.

37th Conference on Neural Information Processing Systems (NeurIPS 2023).

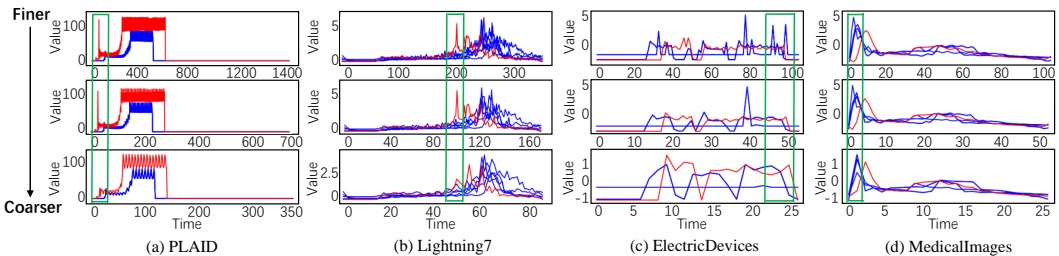

Figure 1: Illustration of time series samples *from the same category* at different time scales. Among all samples in the same category, red indicates the one with the largest variance, and blue indicates a few samples with the smallest variance.

the robustness of models. Moreover, time series data has complex temporal dynamics that make it challenging to manually correct noisy labels [8].

To cope with noisy labels, existing studies on label-noise learning [9, 10] use the memory effect of DNNs to select samples with small losses for training. DNNs memorize the data with clean labels first, and then those with noisy labels in classification training (small-loss criterion) [11]. It is worth noting that the small-loss criterion is not affected by the choice of training optimizations and network structures [12], and is widely utilized for label-noise learning in computer vision [13, 14]. However, the small loss criterion cannot always be applied to time series because the discriminative patterns of time series data are easily distorted by external noises [15, 16]. For example, in a smart grid, distortions may occur due to sampling frequency perturbations, imprecise sensors, or random differences in energy consumption [17]. Such distortions can make it difficult for DNNs to learn the appropriate discriminant patterns of time series, resulting in large training losses for some clean labeled samples. In addition, the small-loss criterion only utilizes the data's label information and does not consider the inherent properties of time series features (i.e., multi-scale information).

Multi-scale properties are crucial in time series classification tasks. In recent years, multi-scale convolution [16], dynamic skip connections [18, 19] and adaptive convolution kernel size [20] have been utilized to learn discriminative patterns of time series. Furthermore, according to related studies [2, 20, 21], the selection of appropriate time scales for time series data can facilitate DNNs to learn class-characteristic patterns. With correct labels, the above studies indicate that the multi-scale properties of time series data can help DNNs learn appropriate discriminative patterns for mitigating the negative effects of time series recording noises. Nevertheless, it remains an open challenge as to how the multi-scale properties of time series can be used for label-noise learning.

To this end, we propose a deep learning paradigm, named Scale-teaching, for time-series classification with noisy labels. In particular, we design a fine-to-coarse cross-scale fusion mechanism for obtaining robust time series embeddings in the presence of noisy labels. We select four time series datasets from the UCR archive [22] to explain our motivation. As shown in Figure 1, in the single scale case (top row), the red and blue samples from the same category have large differences in certain local regions (the green rectangle in Figure 1). By downsampling the time series from fine to coarse, some local regions between the red and blue samples did become similar. Meanwhile, existing studies [12, 23] show that multiple DNNs with random initialization have classification divergence for noisy labeled samples, but are consistent for clean labeled samples. The above findings inspire us to utilize multiple DNNs to combine robust embeddings at different scales to deal with noisy labels. Nonetheless, the coarse scale discards many local regions in the fine scale (as in Figure 1 (c)), which may degenerate the classification performance. Hence, we propose the Scale-teaching paradigm, which can better preserve the local discriminative patterns of fine scale while dealing with distortions.

More specifically, the proposed Scale-teaching paradigm performs the cross-scale embedding fusion in the finer-to-coarser direction by utilizing time series at different scales to train multiple DNNs simultaneously. The cross-scale embedding fusion exploits complementary information from different scales to learn discriminative patterns. This enables the learned embeddings to be more robust to distortions and noisy labels. During training, clean labels are selected through cross-teaching on those networks with the learned embeddings. The small-loss samples in training are used as (clean) labeled data, and the unselected large-loss samples are used as (noisy) unlabeled data. Moreover,

multi-scale embedding graph learning is introduced to establish relationships between labeled and unlabeled samples for noisy label correction. Based on the multi-scale embedding graph, the label propagation theory [24] is employed to correct noisy labels. This drives the model to better fit time series category distribution. The contributions are summarized as follows:

- We propose a deep learning paradigm, called Scale-teaching, for time-series label-noise learning. In particular, a cross-scale fusion mechanism is designed to help the model select more reliable clean labels by exploiting complementary information from different scales.

- We further introduce multi-scale embedding graph learning for noisy label correction using the selected clean labels based on the label propagation theory. Unlike conventional image label-noise learning methods focused on sample loss levels, our approach uses well-learned multi-scale time series embeddings for noise label correction at sample feature levels.

- Extensive experiments on multiple benchmark time series datasets show that the proposed Scale-teaching paradigm achieves a state-of-the-art classification performance. In addition, multi-scale analyses and ablation studies indicate that the use of multi-scale information can effectively improve the robustness of Scale-teaching against noisy labels.

## 2 Related Work

**Label-noise Learning.** Existing label-noise learning studies focus mainly on image data [10]. These studies can be broadly classified into three categories: (1) designing noise-robust objective functions [25, 26] or regularization strategies [27, 28]; (2) detecting and correcting noisy labels [13, 29, 30]; (3) transition-matrix-based [31, 32] and semi-supervised-based [14, 33] methods. In contrast to the methodologies in the first and third categories, approaches categorized under the second category have received considerable attention in recent years [7, 34]. Methods of the second category can be further divided into sample selection and label correction. The common methods of sample selection are the Co-teaching family [12, 13, 23] and FINE [35]. Label correction [36, 37] attempts to correct noisy labels by either using prediction results of classifiers or pseudo-labeling techniques. Recently, SREA [4] utilizes pseudo-labels generated based on a clustering task to correct time-series noisy labels. Although the above methods can improve the robustness of DNNs, how the multi-scale properties of time series are exploited for label-noise learning has not been explored.

**Multi-scale Time Series Modeling.** In recent years, multi-scale properties have gradually gained attention in various time series downstream tasks [18, 38], such as time series classification, prediction, and anomaly detection [39]. For example, Cui et al. [16] employ multiple convolutional network channels of different scales to learn temporal patterns that facilitate time series classification. Chen et al. [19] design a time-aware multi-scale RNN model for human action prediction. Wang et al. [40] introduce a multi-scale one-class RNN for time series anomaly detection. Also, recent studies [41, 42, 43, 44] indicate that multi-scale properties can effectively improve the performance of long-term time series prediction. Unlike prior work, we utilize multiple DNNs with identical architectures to separately capture discriminative temporal patterns across various scales. This enables us to acquire robust embeddings for handling noisy labels via a cross-scale fusion strategy.

**Label Propagation.** Label propagation (LP) is a graph-based inductive inference method [24, 45] that can propagate pseudo-labels to unlabeled graph nodes using labeled graph nodes. Since LP can utilize the feature information of data to obtain pseudo-labels of unlabeled samples, related works employ LP in few-shot learning [46] and semi-supervised learning [47, 48]. Generally speaking, DNNs have the powerful capability for feature extraction, and the learned embeddings tend to be similar within classes and different between classes. Each sample contains feature and label information. Intuitively, the embeddings of samples with noisy labels obtained by DNNs closely align with the true class distribution when the DNNs do not fit noisy labels in the early training stages. Naturally, we create a nearest-neighbor graph based on well-learned multi-scale time series embeddings at the feature level. Subsequently, we employ LP theory to correct the labels of unselected noisy samples using the labels of clean samples chosen by the DNNs. This approach leverages robust multi-scale embeddings to address the issue of noisy labels.

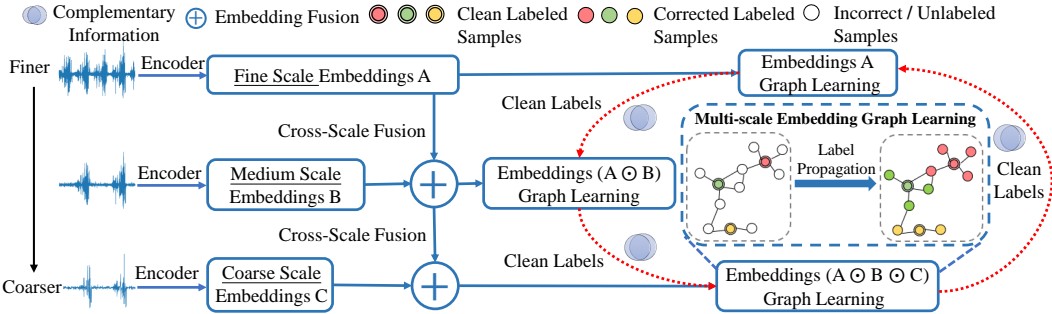

Figure 2: The Scale-teaching paradigm's general architecture comprises two core processes: (i) clean label selection and (ii) noisy label correction. In the clean label selection phase, networks A, B, and C engage in cross-scale fusion, moving from fine to coarse (A→B, B→C)). They employ clean labels acquired through cross-teaching (A→B, B→C, C→A) to guide their respective classification training. In the noisy label correction phase, pseudo labels derived from multi-scale embeddings graph learning are employed as corrected labels for time series not selected as clean labeled samples.

# 3 Proposed Approach

## 3.1 Problem Definition

Given a noisy labeled time series dataset $\mathcal{D} = \{(\mathcal{X}_i, \widehat{y}_i)\}_{i=1}^N$, it contains $N$ time series, where $\mathcal{X}_i \in R^{L \times T}$, $L$ denotes the number of variables, and $T$ is the length of variable. $\widehat{y}_i \in \{1, \ldots, C\}$ is the observed label of $\mathcal{X}_i$ with $\eta$ probability of being a noisy label. Our goal is to enable the DNNs trained on the noisy labeled training dataset $\mathcal{D}_{train}$ to correctly predict the ground-truth labels of the given time series in the test set. Specifically, the problem to be addressed in this paper consists of two steps. The first is to select clean labeled time series from $\mathcal{D}_{train}$, and the second is to perform noisy label correction for time series in $\mathcal{D}_{train}$ that have not been selected as clean labels.

## 3.2 Model Architecture

The overall architecture of Scale-teaching is shown in Figure 2. While this figure illustrates Scale-teaching with input time series at three scales, it can be extended to models with more scales, exceeding three. We utilize a consistent structural encoder to learn embeddings for each input scale sequence. Each encoder undergoes training at two levels: embedding learning for clean sample selection at the feature level and label correction with the multi-scale embeddings. For embedding learning, we propose a cross-scale fusion (Section 3.3) mechanism from fine to coarse to obtain robust embeddings. This approach enables the selection of more dependable clean labels through the small-loss criterion. Specifically, embeddings (A ⊙ B ⊙ C) encompass multi-scale information from fine, medium, and coarse scale sequences derived from the same time series. Regarding noisy label correction, we introduce multi-scale embedding graph learning (Section 3.4) based on label propagation, utilizing the selected clean samples to correct the labels of unselected large-loss samples.

## 3.3 Cross-scale Fusion for Clean Label Selection

After downsampling the original time series at different scales, it eliminates some of the differences in local regions between samples of the same category (as in Figure 1). However, the downsampled sequences (i.e., coarse scale) discard many local regions of the original time series. This tends to degrade the model's classification performance if the downsampled sequences are used directly for classification (please refer to Table 2 in the Experiments section). Meanwhile, existing studies [12, 23] on label-noise learning show that DNNs with different random initializations have high consistency in classification results for clean labeled samples in the early training period, while there is disagreement in the classification of noisy labeled samples. Based on the above findings, we utilize multiple DNNs (or encoders) with different random initializations to learn embeddings of different downsampled scale sequences separately, and perform cross-scale fusion. On the one hand, we exploit complementary

information between adjacent scale embeddings to promote learned embeddings to be more robust for classification. On the other hand, we leverage the divergence in the classification of noisy labeled samples by different DNNs to mitigate the negative impact of noisy labels in training. In this way, we can utilize the cross-scale fusion embeddings for classification, thus better using the small loss criterion [11, 29] for clean label selection. Specifically, downsampling is employed to generate different scale sequences from the same time series. Given a time series $\mathcal{X}_i = \{x_1, x_2, \ldots, x_T\}$, supposing the downsampling ratio is $k$. Then, we only keep data points in $\mathcal{X}_i$ as follows:

$$\mathcal{X}_i^k = \{x_{k*j}\}, j = 1, 2, \ldots, \frac{T}{k}, \tag{1}$$

where $k \in [1, T/2]$, and a larger $k$ indicates that $\mathcal{X}_i^k$ is coarser. As shown in Figure 2, time series with multiple downsampling intervals (i.e., $k$ = 1, 2, 4) is treated as the input data for training. To better utilize the small-loss criterion for clean label selection, each time series sample performs cross-scale fusion from fine to coarse (i.e., A→B, B→C) in the embedding space, which is mathematically defined as:

$$v_i^k = f\left(r_i^k \left\| v_i^{k-t} \right\| \left(r_i^k - v_i^{k-t}\right) \left\| \left(r_i^k \cdot v_i^{k-t}\right)\right.\right), \tag{2}$$

where $r_i^k$ represents the single-scale embedding acquired by learning $\mathcal{X}_i^k$ through an encoder. Meanwhile, $v_i^k$ (or $v_i^{k-t}$) denotes the embedding of the time series $X_i^k$ (or $X_i^{k-t}$) after performing cross-scale fusion. Here, $t$ denotes the interval between adjacent downsampling ratios, and $\|$ signifies the concatenation of two vectors to form a new vector. Notably, when $k = 1$, we employ the single-scale for classification training, resulting in $v_i^k = r_i^k$. By combining $(r_i^k - v_i^{k-t})$ and $(r_i^k \cdot v_i^{k-t})$ for vector concatenation, $v_i^k$ can capture more nuanced discriminative information between $r_i^k$ and $v_i^{k-t}$ than that of simply concatenating $r_i^k$ with $v_i^{k-t}$. The function $f(\cdot)$ represents a two-layer nonlinear network mapping function for fusing information of $r_i^k$ and $v_i^{k-t}$. Additionally, $v_i^k$ has the same dimension as $r_i^k$ and serves as the input data for the multi-scale embedding graph learning process.

### 3.4 Multi-scale Embedding Graph Learning for Noisy Label Correction

We now present the multi-scale embedding graph learning module for correcting noisy labels. This module incorporates selected clean labels using label propagation theory. The process consists of two stages: graph construction and noisy label correction.

**Graph Construction.** It is assumed that the set of cross-fusion embeddings obtained from a batch of time series is defined as $V = \{v_1^k, v_2^k, \ldots, v_M^k\}$, where $M$ is the batch size. Intuitively, samples close to each other in the feature space have a high probability of belonging to the same class. However, in label-noise learning, $v_i^k$ obtained from the current iterative training of the model may have unstable information, resulting in large deviations in the information of the nearest-neighbor samples of $v_i^k$. To address this issue, the proposed approach performs a momentum update [49] on $v_i^k$ during training, which is defined as:

$$\bar{v}_i^k[e] = \alpha v_i^k[e] + (1 - \alpha)\bar{v}_i^k[e - 1], \tag{3}$$

where $e$ is the current training epoch and $\alpha$ denotes the momentum update parameter.

The multi-scale embeddings nearest-neighbor graph can be created by using Euclidean distance among different $\bar{v}_i^k$. A common approach is the use of the Gaussian similarity function [45] to obtain the nearest-neighbor graph edge weight, which is defined as:

$$W_{ij} = \exp\left(-\frac{1}{2}d\left(\frac{\bar{v}_i^k}{\sigma}, \frac{\bar{v}_j^k}{\sigma}\right)\right), \tag{4}$$

where $d(\cdot)$ is the Euclidean distance function and $\sigma$ is a fixed parameter. $W \in R^{M \times M}$ is a symmetric adjacency matrix, and the element $W_{ij}$ denotes the nearest-neighbor edge weight between the embedding $v_i^k$ and $v_j^k$ (note that larger values indicate closer proximity). Then, $W$ is normalized based on the graph laplacians [50] to obtain $Q = D^{-1/2}WD^{-1/2}$, where $D = diag(W1_n)$ is a diagonal matrix. Specifically, the $K$ neighbors with the largest values in each row of $Q$ are employed to create the nearest-neighbor graph. It is noteworthy that the embeddings in each mini-batch are utilized to generate the nearest-neighbor graph, thus obtaining $Q$ within short computational time.

**Noisy Label Correction.** Specifically, small training loss samples acquired by DNNs in the early training period can be considered as clean samples, while samples with large training losses are considered as noisy ones [12, 14]. The above learning pattern of DNNs has been mathematically validated [11] (see Appendix A for details). Under this criterion, prior studies [12, 13, 23] have typically employed samples with small losses after a $e_{warm}$ warm-up training as clean labels. Following [29], we extend the small-loss sample selection process to operate within each class, thereby enhancing the overall quality of the chosen clean labels. In our method, samples chosen with clean labels are considered labeled data, whereas unselected samples are treated as unlabeled data.

We utilize clean samples selected from time series at different scales in a cross-teaching manner (as in Figure 2). This could explore complementary information from different scale fusion embeddings to deal with noisy labels. It is supposed that there is a corresponding one-hot encoding matrix $Y \in R^{M \times C}$ ($Y_{ij} \in \{0, 1\}$) for the cross-fusion embeddings $V$. If $y_i$ is identified as a clean label, we employ $y_i$ to set $Y_i$ as a one-hot encoded label. Otherwise, all the elements in $Y_i$ are identified as zero. Through $Y$, the pseudo-label of each node in the nearest-neighbor graph $Q$ can be obtained in an iterative way based on the label propagation theory. The specific solution formula is defined as:

$$F_{t+1} = \beta Q F_t + (1 - \beta)Y, \tag{5}$$

where $F_t \in R^{M \times C}$ denotes the predicted pseudo-label of the $t$-th iteration and $\beta \in (0, 1)$ is a hyperparameter. Naturally, $F_t$ has a closed-form solution [24] defined as follows:

$$\mathcal{F} = (I - \beta Q)^{-1}Y, \tag{6}$$

where $\mathcal{F} \in R^{M \times C}$ is the final pseudo-labels and $I$ denotes the identity matrix. Finally, the corrected label obtained for an unselected large-loss sample $X_i$ is defined as:

$$y_i = \arg\max_c \mathcal{F}_i^c, \tag{7}$$

However, $\mathcal{F}$ is the estimated pseudo-labels, which inevitably contain some incorrect labels. To address this issue, two strategies are used to improve the quality of pseudo-labels in $\mathcal{F}$. For the first strategy, the model continues training $e_{update}$ epochs by using small-loss samples after $e_{warm}$ epochs warm-up training to improve the robustness of the multi-scale embeddings. Then, the noisy label correction is performed after ($e_{warm} + e_{update}$) epoch. For the second strategy, a dynamic threshold $\varphi_e(c) = \frac{\delta_e(c)}{\max(\delta_e)}\gamma$ is utilized for each class [51] to select the pseudo-labels with a high confidence for noisy label correction, where $\delta_e(c)$ is the number of labeled samples contained in class $c$ in the $e$-th epoch, and $\gamma$ is a constant threshold.

**Overall Training.** Finally, each encoder utilizes the selected clean samples in combination with multi-scale embedding graph learning to perform noisy label correction for unselected large-loss samples. Combining the training data of the selected clean labels and those of corrected labels, the proposed Scale-teaching paradigm utilizes cross-entropy for time-series label-noise learning. Please refer to Algorithm 1 in the Appendix for the specific pseudo-code of Scale-teaching.

## 4 Experiments

### 4.1 Experiment Setup

**Datasets.** We use three time series benchmarks (four individual large datasets [3, 52, 53], UCR 128 archive [22], and UEA 30 archive [54]) for experiments. Among the four individual large datasets, HAR [52] and UniMiB-SHAR [3] are human activity recognition scenarios; FD-A [53] is the mechanical fault diagnosis scenario; Sleep-EDF [52] belongs to the sleep stage classification scenario. The UCR archive [22] contains 128 univariate time series datasets from different real-world scenarios. The UEA archive [54] contains 30 multivariate time series datasets from real-world scenarios. For details on the above datasets, please refer to Appendix B. Since all the datasets in three time series benchmarks are correctly labeled, we utilize a label transformation matrix $T$ to add noises to the original correct labels [4], where $T_{ij}$ is the probability of label $i$ being flipped to $j$. We use three types of noisy labels for evaluations, namely Symmetric (Sym) noise, Asymmetric (Asym) noise, and Instance-dependent (Ins) noise. Symmetric (Asymmetric) noise randomly replaces a true label with other labels with an equal (unequal) probability. Instance noise [55] means that the noisy label is instance-dependent. Like [4, 12, 23], we use the test set with correct labels for evaluations.

Table 1: Test classification accuracy results compared with baselines on three time series benchmarks. The best results are **bold**, and the second best results are underlined. When P-value < 0.05, it indicates that the performance of Scale-teaching is statistically significant than the baseline.

| Dataset | Noise Ratio | Metric | Standard | Mixup | Co-teaching | FINE | SREA | SELC | CULCU | Scale-teaching |
|---|---|---|---|---|---|---|---|---|---|---|
| Four individual large datasets | Sym 20% | Avg Rank | 4.75 | 4.75 | 4.50 | 7.50 | 6.50 | 4.50 | 2.50 | **1.00** |
| | Sym 50% | Avg Rank | 4.75 | 4.50 | 4.75 | 7.25 | 5.75 | 4.50 | 3.25 | **1.25** |
| | Asym 40% | Avg Rank | 5.00 | 5.50 | 3.75 | 7.50 | 5.75 | 4.00 | 3.25 | **1.00** |
| | Ins 40% | Avg Rank | 4.75 | 4.25 | 4.25 | 7.25 | 6.00 | 4.75 | 3.50 | **1.00** |
| UCR 128 archive | Sym 20% | Avg Rank | 4.15 | 4.33 | 3.61 | 7.50 | 6.16 | 3.48 | 3.54 | **3.02** |
| | | P-value | 1.90E-04 | 4.06E-05 | 1.90E-03 | 1.49E-34 | 1.70E-17 | 3.04E-03 | 8.57E-03 | - |
| | Sym 50% | Avg Rank | 4.31 | 4.57 | 4.05 | 6.43 | 5.89 | 3.56 | 3.86 | **3.11** |
| | | P-value | 3.15E-05 | 1.70E-05 | 4.02E-04 | 7.48E-19 | 1.22E-15 | 1.40E-02 | 4.93E-03 | - |
| | Asym 40% | Avg Rank | 4.38 | 4.80 | 3.93 | 6.91 | 5.91 | 3.30 | 3.67 | **2.95** |
| | | P-value | 1.62E-05 | 3.53E-07 | 6.10E-04 | 1.93E-23 | 9.82E-14 | 1.89E-02 | 2.24E-02 | - |
| | Ins 40% | Avg Rank | 4.05 | 4.52 | 4.02 | 7.04 | 6.18 | 3.30 | 3.77 | **2.95** |
| | | P-value | 1.43E-05 | 1.81E-06 | 2.43E-04 | 9.81E-26 | 2.36E-17 | 3.27E-02 | 1.54E-02 | - |
| UEA 30 archive | Sym 20% | Avg Rank | 5.03 | 5.20 | 3.83 | 6.37 | 4.77 | 3.73 | 4.00 | **2.73** |
| | | P-value | 6.61E-04 | 3.33E-04 | 2.69E-02 | 2.37E-05 | 1.14E-02 | 2.63E-02 | 3.93E-02 | - |
| | Sym 50% | Avg Rank | 5.17 | 5.73 | 4.23 | 6.23 | 3.93 | 3.83 | 4.30 | **2.43** |
| | | P-value | 2.98E-04 | 7.40E-05 | 1.59E-02 | 9.35E-05 | 1.67E-02 | 1.08E-02 | 3.75E-02 | - |
| | Asym 40% | Avg Rank | 5.60 | 4.77 | 4.40 | 6.13 | 4.20 | 4.00 | 3.97 | **2.73** |
| | | P-value | 3.81E-03 | 6.17E-03 | 1.63E-02 | 9.33E-05 | 1.36E-02 | 2.62E-02 | 3.88E-02 | - |
| | Ins 40% | Avg Rank | 5.20 | 4.77 | 4.33 | 6.60 | 4.27 | 4.20 | 3.77 | **2.60** |
| | | P-value | 6.08E-04 | 2.92E-03 | 1.20E-02 | 2.55E-05 | 5.52E-03 | 1.08E-02 | 3.47E-02 | - |

**Baselines.** We select seven methods for comparative analyses, namely 1) Standard: direct training of the model using cross-entropy with all noisy labels; 2) Mixup [56]; 3) Co-teaching [12]; 4) FINE [35]; 5) SREA [4]; 6) SELC [37]; and 7) CULCU [23]. Among them, Standard, Mixup, and Co-teaching are the benchmark methods for label-noise learning. FINE, SELC, and CULCU are the state-of-the-art methods that do not need to focus on data types, and SREA is the state-of-the-art method in time series domain. In addition, for fair comparisons, all the baselines and the proposed Scale-teaching paradigm use the same encoder and classifier. We focus on the ability of different label-noise learning paradigms to cope with time series noise labels, rather than the classification performance achieved by using fully correct labels. Hence, considering the trade-off between the running time and classification performance, we choose FCN [57] as the encoder of Scale-teaching. For more details of baselines, please refer to Appendix C.

**Implementation Details.** Based on the experience [19, 44] in time series modeling, we utilize three different sampling intervals 1, 2, 4 as the input muti-scale series data for Scale-teaching. We use Adam as the optimizer. The learning rate is set to 1e-3, the maximum batch size is set to 256, and the maximum epoch is set to 200. $e_{warm}$ is set to 30 and $e_{update}$ is set to 90. $\alpha$ in Eq. 3 is set to 0.9, $\sigma$ in Eq. 4 is set to 0.25, $\beta$ in Eq. 5 is set to 0.99, the largest neighbor $K$ is set to 10, and $\gamma$ is set to 0.99. In addition, following the parameter settings suggested in [23], we linearly decay the learning rate to zero from the 80-th epoch to 200-th epoch. For a comprehensive understanding of the hyperparameter selection and the implementation of the small-loss criterion applied to Scale-teaching, please consult Appendix C. To reduce random errors, we utilize the mean test classification accuracy of the last five epochs of the model on the test set as experimental results. All the experiments are independently conducted five times with five different seeds, and the average classification accuracy and rank are reported. Finally, we build our model using PyTorch 1.10 platform with 2 NVIDIA GeForce RTX 3090 GPUs. Our implementation of Scale-teaching is available at `https://github.com/qianlima-lab/Scale-teaching`.

## 4.2 Main Results

We evaluate each time series benchmack using four noise ratios, Sym 20%, Sym 50%, Asym 40%, and Ins 40%. Due to space constraints, we only give the average ranking of all the methods on each benchmark in Table 1. Please refer to Appendix D for the specific test classification accuracies. Besides, for UCR 128 and UEA 30 archives, we use the Wilcoxon signed rank test (P-value) [58] to analyze the classification performance of baselines. As shown in Table 1, the proposed Scale-teaching paradigm achieves the best Avg Rank in all the cases. It is found that Mixup [56] and FINE [35] perform worse than the Standard method in most cases. For Mixup, the complex dynamic properties

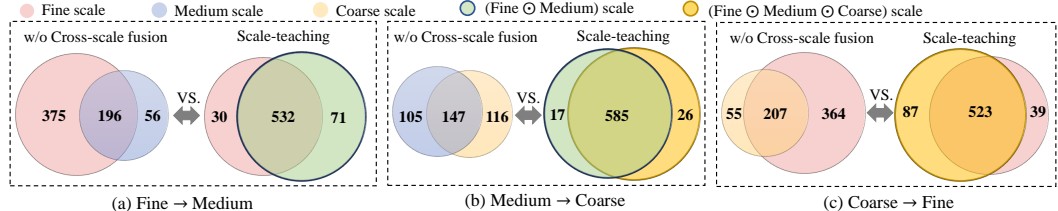

Figure 3: Venn diagram of the average number of correctly classified samples for the different scale sequences of UCR 128 archive with Sym 20% noisy labels. The numbers in the figure indicate the complements and intersections of classification results at different scales.

Table 2: The test classification accuracy (%) results of different scale classifiers on UCR 128 archive. The best results are **bold**, and the second best results are underlined. When P-value < 0.05, it indicates that the performance of Scale-teaching's coarse scale classifier is significant than other classifiers.

| Method | | w/o Cross-scale fusion | | | Scale-teaching | | |
|---|---|---|---|---|---|---|---|
| Noise Ratio | Metric | Fine | Medium | Coarse | Fine | Medium | Coarse |
| | Avg Acc | 65.13 | 30.11 | 28.17 | 59.67 | 68.17 | **68.70** |
| Sym 20% | Avg Rank | 2.38 | 5.09 | 5.37 | 3.20 | 2.17 | **2.11** |
| | P-value | 1.89E-03 | 2.85E-37 | 2.07E-40 | 1.58E-09 | 3.74E-02 | - |
| | Avg Acc | 49.61 | 29.01 | 28.87 | 47.75 | 51.93 | **52.87** |
| Asym 40% | Avg Rank | 2.64 | 4.78 | 4.75 | 3.01 | 2.45 | **2.27** |
| | P-value | 1.94E-03 | 6.78E-25 | 1.59E-27 | 1.80E-07 | 2.80E-02 | - |

of the original time series are destroyed probably due to the mixture of two different time series mechanisms. FINE uses embeddings of the input data to select clean labels. Although FINE achieves advanced classification performance for image data, it is difficult to be used directly for time series data because its discriminative patterns are easily distorted by external noises. SREA [4] has a good performance on the UEA 30 archive, while it performs poorly on the other benchmarks. Meanwhile, Co-teaching [12], SELC [37], and CULCU [23] are more robust against time series noisy labels in different cases, further indicating that the small-loss criterion is also applicable to time series.

## 4.3 Multi-scale Analysis

To explain the multi-scale mechanism in the Scale-teaching paradigm, we add an ablation study based on Scale-teaching (w/o cross-scale fusion). We select the UCR 128 archive to analyze the classification results obtained by the fine, medium, and coarse scale classifiers. As shown in Figure 3, the classification results of different scale sequences have evident complementary information. Scale-teaching can effectively use complementary information between cross-scale to obtain more robust embeddings and clean labels. In response to the tendency of the coarse scale to ignore discriminative patterns in fine scale (please see Table 2), our proposed cross-scale fusion mechanism can effectively improve the classification performance of medium and coarse scales while retaining complementarity. Please refer to Appendix E for the specific classification results of Figure 3 and Table 2. In Appendix E, we also analyze the order and size of the downsampled input scale sequence for Scale-teaching.

Scale-teaching utilizes multi-scale embeddings to generate the nearest-neighbor graph, and uses clean labels selected for noisy label correction. To explore the distribution of different classes of embeddings, we employ t-SNE [59] for dimensionality reduction visualization. Specifically, we utilize the UniMiB-SHAR dataset containing Sym 20% noisy labels for visualization. As shown in Figure 4, we find that the embeddings learned by Scale-teaching are more discriminative across classes than the Standard and CULCU methods that use a single scale series for training. The above results suggest that Scale-teaching can effectively exploit the complementary information between different scales, prompting the learned embeddings to be more discriminative between classes. In addition, we choose the FD-A dataset for t-SNE visualization, and please refer to Appendix E.

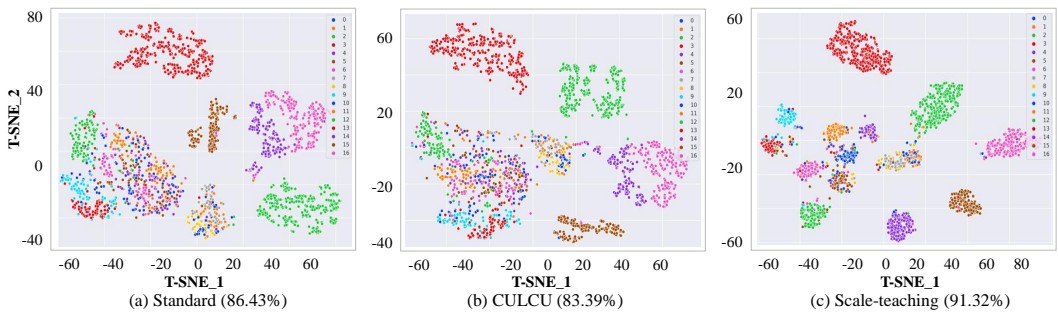

(a) Standard (86.43%)   (b) CULCU (83.39%)   (c) Scale-teaching (91.32%)

Figure 4: t-SNE visualization of the learned embeddings on the UnimiB-SHAR dataset with Sym 20% noisy labels (values in parentheses are the test classification accuracies).

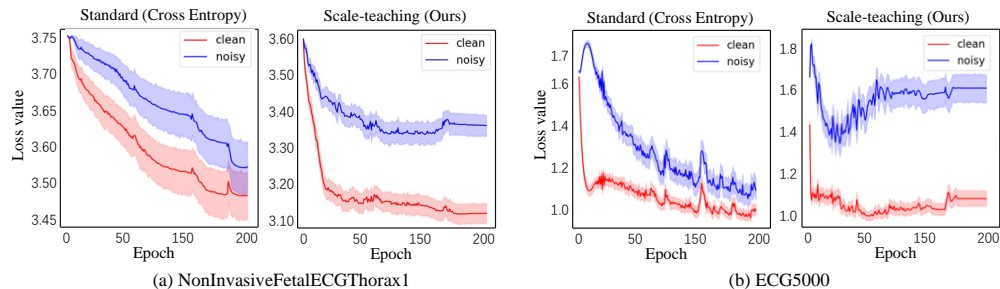

(a) NonInvasiveFetalECGThorax1   (b) ECG5000

Figure 5: The change of loss values for clean and noisy time series samples under Aysm 40% noise labels. The solid line and shading indicate the mean and standard deviation loss values of all clean (or noisy) training samples within each epoch.

### 4.4 Small-loss Analysis

To analyze the application of the small-loss criterion to time series data, we visualize the change of loss values for ground-truth clean and noisy time series samples during training. Specifically, Figure 5 shows the change in loss values of the models trained by the Standard method and Scale-teaching on two UCR time series datasets. When the model is trained with the Standard method, differences can be found in the loss values of clean and noisy samples in the network early training, especially in Figure 5 (b). The Standard method makes the model gradually fit the noisy samples as the training proceeds, while Scale-teaching improves the ability of the model to handle noisy labels. To further prove its effectiveness, we selected two other UCR datasets for the loss value change analysis, which have the same pattern as Figure 5. Also, we report the HAR and UniMiB-SHAR dataset's loss value probability distributions of clean and noisy samples. For more details, please refer to Appendix F.

### 4.5 Ablation Study

To verify the robustness of each module in Scale-teaching, the ablation experiments have been conducted in the HAR and UniMiB-SHAR datasets, and the results are shown in Table 3. Specifically, (1) **w/o cross-scale fusion**: the cross-scale embedding fusion from fine to coarse mechanism is ablated; (2) **only single scale**: only the original time series is used for training; (3) **w/o graph learning**: the multi-scale embedding graph learning module for noisy label correction is ablated; (4) **w/o moment**: the embedding momentum update mechanism (Eq. 3) is ablated; (5) **w/o dynamic threshold**: using a dynamic threshold to select high-quality propagation pseudo-labels is ablated.

As shown in Table 3, the cross-scale fusion strategy (w/o cross-scale fusion) and the clean labels cross-teaching mechanism (only single scale) can effectively improve the classification performance of Scale-teaching, especially on the UniMiB-SHAR dataset with a large number of classes. Meanwhile, in terms of label correction based on multi-scale embedding graph learning, the results of the corresponding ablation module show that improving the stability of embedding (w/o moment) and

Table 3: The test classification accuracy (%) results of ablation study (values in parentheses denote drop accuracy).

| Method | HAR | | UniMiB-SHAR | |
|---|---|---|---|---|
| | Sym 50% | Asym 40% | Sym 50% | Asym 40% |
| Scale-teaching | **90.17** | **89.62** | **81.31** | **70.68** |
| w/o cross-scale fusion | 88.47 (-1.70) | 87.64 (-1.98) | 73.32 (-7.99) | 61.62 (-9.06) |
| only single scale | 89.01 (-1.06) | 88.11 (-1.51) | 69.89 (-11.42) | 60.32 (-10.36) |
| w/o graph learning | 88.06 (-2.11) | 87.65 (-1.97) | 79.72 (-1.59) | 68.87 (-1.81) |
| w/o moment | 89.76 (-0.41) | 88.76 (-0.86) | 80.57 (-0.74) | 69.85 (-0.83) |
| w/o dynamic threshold | 89.12 (-1.05) | 88.75 (-0.87) | 77.42 (-3.89) | 69.53 (-1.15) |

Table 4: The test classification accuracy (%) results on four individual large datasets without noisy labels. The best results are **bold**, and the second best results are underlined.

| Dataset | Standard | Mixup | Co-teaching | FINE | SREA | SELC | CULCU | Scale-teaching |
|---|---|---|---|---|---|---|---|---|
| HAR | 93.29 | 95.42 | 93.77 | 93.13 | 93.02 | 93.76 | **94.75** | 94.72 |
| UniMiB-SHAR | 89.14 | 84.84 | 88.24 | 88.14 | 65.51 | 89.28 | 89.46 | **93.61** |
| FD-A | 99.93 | 99.91 | **99.96** | 68.22 | 90.25 | 99.82 | 99.95 | **99.96** |
| Sleep-EDF | 84.93 | 84.67 | 85.37 | 84.62 | 79.42 | 84.82 | **85.54** | 85.34 |

selecting high-quality pseudo-labels (w/o dynamic threshold) can effectively improve the performance of label correction based on graph learning.

Furthermore, we select the four individual large datasets without noisy labels for evaluation. As shown in Table 4, Scale-teaching's classification performance is still better than most baselines. It's worth mentioning that SREA [4] employs an unsupervised time series reconstruction loss as an auxiliary task, which reduces the model's classification performance without noisy labels. We also provide the corresponding test classification results for Tables 3 and 4 under the F1-score metric in Appendix G. Additionally, we find the running time of Scale-teaching, which is faster than FINE, SREA and CULCU for datasets with a larger number of samples or longer length of the sequence. We further analyze the classification performance of the proposed Scale-teaching paradigm and time series classification methods [15, 60] in Appendix G.

## 5 Conclusions

**Limitations.** The input scales of our proposed Scale-teaching paradigm can only select a fixed number of scales for training, and the running time will increase as the number of scales increases.

**Conclusion.** In this paper, we propose a deep learning paradigm for time-series classification with noisy labels called Scale-teaching. Experiments on the three time series benchmarks show that the Scale-teaching paradigm can utilize the multi-scale properties of time series to effectively handle noisy labels. Comprehensive analyses on multi-scale and ablation studies demonstrate the robustness of the Scale-teaching paradigm. In the future, we will explore the design of scale-adaptive time-series label-noise learning models.

## Acknowledgments

We thank the anonymous reviewers for their helpful feedbacks. We thank Professor Eamonn Keogh and all the people who have contributed to the UCR 128 archive, UEA 30 archive, and the four large individual time series classification datasets. The work described in this paper was partially funded by the National Natural Science Foundation of China (Grant Nos. 62272173, 61872148, 62206041), the Natural Science Foundation of Guangdong Province (Grant Nos. 2022A1515010179, 2019A1515010768), the Science and Technology Planning Project of Guangdong Province (Grant No. 2023A0505050106), the Fundamental Research Funds for the Central Universities under grants DUT22RC(3)015. The authors would like to thank Siying Zhu, Huawen Feng, Yu Chen, and Junlong Liu from SCUT for their review and helpful suggestions.

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

# Supplementary Material:
# Scale-teaching: Robust Multi-scale Training for Time Series Classification with Noisy Labels

## A  Small-loss Criterion

DNNs have been widely known to first learn simple and generalized patterns, which is achieved by learning clean data. After that, the networks gradually overfit noisy ones. In other words, when we train a model with a dataset containing incorrectly labeled samples, we can consider the samples with small training losses as clean ones and use them to update the model. Formally, let $f^*$ be the target concept which determines the true label of $x$ and model $g^* = g(\boldsymbol{x}; \Theta^*)$ minimizing the expected loss, i.e.,

$$\Theta^* = \arg\min_{\Theta} \mathbb{E}_{(\boldsymbol{x}, \tilde{y})} \left[ \ell_{CE}(g(\boldsymbol{x}; \Theta), \tilde{y}) \right]. \tag{8}$$

Then, the small-loss criterion can be stated as follows[1]:

**Theorem 1.** *Suppose $g$ is $\epsilon$-close to $g^*$, i.e., $\|g - g^*\|_\infty = \epsilon$, for two examples $(\boldsymbol{x}_1, \tilde{y})$ and $(\boldsymbol{x}_2, \tilde{y})$, assume $f^*(\boldsymbol{x}_1) = \tilde{y}$ and $f^*(\boldsymbol{x}_2) \neq \tilde{y}$, if $T$ satisfies the diagonally-dominant condition $T_{ii} > \max\{\max_{j \neq i} T_{ij}, \max_{j \neq i} T_{ji}\}, \forall i$, and $\epsilon < \frac{1}{2} \cdot (T_{\tilde{y}\tilde{y}} - T_{f^*(\boldsymbol{x}_2)\tilde{y}})$, then $\ell_{CE}(g(\boldsymbol{x}_1), \tilde{y}) < \ell_{CE}(g(\boldsymbol{x}_2), \tilde{y})$.*

The work [11] provides the proof of this theorem. It shows that during training, the model can select clean samples according to the loss values. The reason is that the loss values of clean samples among the samples with the same observed labels are smaller. It is worth noting that the theorem is under the assumption of the class-dependent noise type and requires the transition matrix to satisfy the diagonally-dominant condition. Additionally, the finite data may also make the conditions of the theorem difficult to hold because the model $g$ may be far away from $g^*$.

## B  Dataset Information

To evaluate the robustness of our proposed Scale-teaching and baselines on the time-series label-noise learning task, we selected three benchmark time-series datasets for experimental analysis.

### B.1  Four individual large datasets

The statistical information of the four individual time series datasets is shown in Table 5. And the specific dataset information is as follows:

**Human Activity Recognition (HAR)**

The HAR dataset [52, 61] is collected from 30 students performing six human actions (i.e., walking, walking upstairs, downstairs, standing, sitting, and lying down) by wearing sensors.

**University of Milano Bicocca Smartphone-based Human Activity Recognition (UniMiB SHAR)**

The UniMiB SHAR dataset [3, 62] is human activity information collected at a sampling rate of 50 Hz from volunteers with a smartphone with an accelerometer sensor in the front pocket of their pants. Specifically, each accelerometer entry is labeled by specifying the type of ADL (e.g., walking, sitting, or standing) or the type of fall (e.g., forward, fainting, or backward).

**Faulty Detection Condition A (FD-A)**

The FD-A dataset [52, 63] is generated by an electromechanical drive system that monitors the condition of rolling bearings and detects their failure. Each rolling bearing can be classified into three categories: undamaged, inner damaged, and externally damaged.

**Sleep Stage EEG Signal Classification (Sleep-EDF)**

The Sleep-EDF dataset [52, 64] includes the whole night PSG sleep recordings, which contain five EEG sleep signal recordings: Wake (W), Non-rapid eye movement (N1, N2, N3), and Rapid Eye Movement (REM).

Table 5: A summary of four individual large time series datasets used in the experiments.

| Dataset | # Train | # Test | Length | # Variables | # Classes |
|---------|---------|--------|--------|-------------|-----------|
| HAR | 7352 | 2947 | 128 | 9 | 6 |
| Sleep-EDF | 25612 | 8910 | 3000 | 1 | 5 |
| FD-A | 8184 | 2728 | 5120 | 1 | 3 |
| UniMiB-SHAR | 9416 | 2354 | 453 | 1 | 17 |

## B.2 UCR 128 Archive

The UCR time series archive [22] contains 128 univariate datasets and is widely used for classification in the time series mining community. Each UCR dataset includes a single training set and a single test set, and each time series sample has been z-normalized. In addition, we uniformly use the mean-imputation method to preprocess the datasets that contain missing values. For detailed information about UCR datasets, please refer to `https://www.cs.ucr.edu/~eamonn/time_series_data_2018/`.

## B.3 UEA 30 Archive

The UEA time series archive [54] contains 30 multivariate datasets, mainly derived from Human Activity Recognition, Motion classification, ECG classification, EEG/MEG classification, Audio Spectra Classification, and other realistic scenarios. Each dataset contains a partitioned training set and a test set. In addition, we use the mean-imputation method to deal with datasets with missing values. For detailed information about UEA datasets, please refer to `https://www.timeseriesclassification.com/dataset.php`.

## C  Baselines

To analyze the performance and effectiveness of Scale-teaching on time-series label-noise learning, we selected seven baselines for comparative analysis. The specific information is as follows.

- Standard directly employs all samples in the training set containing noisy labels and performs supervised classification training using cross-entropy loss. Then, the trained model is used to make predictions on the test set.

- Mixup [56] trains a neural network on convex combinations of pairs of time series samples and their labels (whatever is clean or noisy). For the specific open source code, please refer to `https://github.com/facebookresearch/mixup-cifar10`.

- Co-teaching [12] trains two deep neural networks simultaneously, and lets them teach each other given every mini-batch with selected clean labels based on a small-loss criterion. For the specific open source code, please refer to `https://github.com/bhanML/Co-teaching`.

- FINE [35] utilizes a novel detector for clean label selection. Especially, FINE focus on each data point's latent representation dynamics and measures the alignment between the latent distribution and each representation using the eigen decomposition of the data gram matrix. For the specific open source code, please refer to `https://github.com/Kthyeon/FINE_official`.

- SREA [4] employs a novel multi-task deep learning approach for time series noisy label correction that jointly trains a classifier and an autoencoder with a shared embedding representation. For the specific open source code, please refer to `https://github.com/Castel44/SREA`.

- SELC [37] utilizes a simple and effective method self-ensemble label correction (SELC) to progressively correct noisy labels and refine the model. For the specific open source code, please refer to `https://github.com/MacLLL/SELC`.

- CULCU [23] incorporates the uncertainty of losses by adopting interval estimation instead of point estimation of losses to select clean labels based on Co-teaching. CULCU has two

versions: CNLCU-S and CNLCU-H, where CNLCU-S uses soft labels for training and CNLCU-H uses hard labels for training. According to the original paper's [23] experimental results, CNLCU-S has a better performance. Hence, we use CNLCU-S as a baseline. For the specific open source code, please refer to `https://github.com/xiaoboxia/CNLCU`.

Finally, based on the source code of the above baselines, we provide the reproduction source code of all baselines, as well as the source code of our proposed Scale-teaching (refer to Algorithm 1). For the specific open-source code, please refer to our GitHub repository `https://github.com/qianlima-lab/Scale-teaching`.

Our experiment contains 162 datasets. It would be time-consuming to perform hyperparameter selection for each dataset. Therefore, the hyperparameters of Scale-teaching are not carefully tuned for each dataset, and most of the hyperparameters are set based on the default hyperparameters of related works. The learning rate and maximum epoch are set based on the parameters of existing noise-label learning methods, such as FINE and CULCU. $\alpha$ in Eq. 3, $\sigma$ in Eq. 4 and $\beta$ in Eq. 5 are set based on the default hyperparameters of related label propagation works. $e_{warm}$ is based on FINE settings. $e_{update}$, $\gamma$ and batch size are based on manual empirical settings without specific hyperparameter analysis. The largest neighbor $K$ is set based on human experience, and we had a simple test on several datasets, and found that a larger value of does not improve the classification performance, but instead increases the running time of the model.

For the implementation of small-loss criterion in Scale-teaching, we select small-loss samples within each class from the mini-batch data as clean labeled data. For stduies [12, 13], they use warm-up training to decrease $\lambda(e)$ from 1 to $1 - \eta$. $\lambda(e)$ denotes the selection ratio of small-loss samples within the mini-batch data without considering the difference of class, and $\eta$ is the ratio of noise labels in the training set. Based on the above criterion, the current work [29] uses the Jensen-Shannon divergence to calculate difference $d$ between the classification result $p_i$ of sample $\mathcal{X}_i^c$ and the observation label $\hat{y}_i$. Following [29], for each class $c$, we consider the observed label of $\mathcal{X}_i^c$ as a clean label when the $d$ of the training sample $\mathcal{X}_i^c$ is less than $d_{avg}^e$ after a $e_{warm}$ warm-up training. $d_{avg}^e$ denotes the average of $d$s of all the training samples when the epoch is $e$. We observed that using the Jensen-Shannon divergence method [29] and directly employing stduies [12, 13] for clean sample selection within each class have distinct strengths and weaknesses when applied to various time series datasets. In our study, we implemented the strategy of stduies [12, 13] for clean sample selection within each class on four individual large datasets and the UCR 128 archive. Meanwhile, the Jensen-Shannon divergence method [29] was applied to the UEA 30 archive for clean sample selection within each class.

# D   Details of Main Results

For the four individual large time series datasets, the specific classification results of our proposed Scale-teaching paradigm and baselines are shown in Table 6. For the UCR 128 archive, the specific classification results for all methods with different noise ratios are shown in Table 11 (Sym 20%), 12 (Sym 50%), 13 (Asym 40%), and 14 (Ins 40%). For the UEA 30 archive, the specific classification results for all methods at different noise ratios are shown in Tables 15 (Sym 20%), 16 (Sym 50%), 17 (Asym 40%), and 18 (Ins 40%). For layout and reading convenience, we only give the average classification accuracy for multiple runs of all methods without standard deviation on the UCR 128 archive and UEA 30 archive.

# E   Details of Multi-scale Results

To analyze the multi-scale mechanism in the Scale-teaching paradigm, we provide the classification performance of classifiers corresponding to fine, medium and coarse scales, as shown in Tables 19 and 21. And the classification results by ablation cross-scale fusion mechanism based on the Scale-teaching are shown in Tables 20 and 22. For the abbreviations in Tables 19, 20, 21 and 22, such as a_t_b_f, b_f_c_t, and c_t_a_f, where $a$ denotes fine classifier, $b$ denotes medium classifier, and $c$ denotes coarse classifier, and $t$ and $f$ represent correct and incorrect classification results, respectively. For example, a_t_b_f indicates the number of samples correctly predicted by the fine classifier and incorrectly predicted by the medium classifier. In addition, we provide t-SNE [59] visualization on the FD-A dataset with Sym 50% noisy labels (as in Figure 6) to explore the distribution of different classes of embeddings. Figure 6 shows that the cross-scale fusion mechanism in Scale-teaching for

**Algorithm 1** The proposed Scale-teaching paradigm.

---

**Input:** encoders $[w_A, w_B, w_C]$, classifiers $[c_A, c_B, c_C]$, fine-scale series $x_A$, medium-scale series $x_B$, and coarse-scale series $x_C$

**Output:** $[w_A, w_B, w_C]$ and $[c_A, c_B, c_C]$

   **Note:** For clarity, our analysis utilizes three distinct scales for training, but this approach can be extended to incorporate multiple scales.

 1: **Step one:** Obtain single-scale embeddings $r_A$, $r_B$, $r_C$;
   $r_A = w_A(x_A)$;
   $r_B = w_B(x_B)$;
   $r_C = w_C(x_C)$;
 2: **Step two:** Obtain cross-scale embeddings $v_A$, $v_B$, $v_C$;
   $v_A = r_A$;
   $v_B = $ Eq. 2$(r_B, v_A)$;
   $v_C = $ Eq. 2$(r_C, v_B)$;
 3: **Step three:** Obtain clean labels $y_A$, $y_B$, $y_C$ for cross-teaching training;
   $y_A = c_C(v_C)$ via small loss criterion;
   $y_B = c_A(v_A)$ via small loss criterion;
   $y_C = c_B(v_B)$ via small loss criterion;
 4: **Step four:** Obtain corrected labels $yc_A$, $yc_B$, $yc_C$ for classification training;
   $yc_A = $ Eq. 6$(v_A, y_A)$ via label propagation;
   $yc_B = $ Eq. 6$(v_B, y_B)$ via label propagation;
   $yc_C = $ Eq. 6$(v_C, y_C)$ via label propagation;
 5: **Step five:** Overall training;
   Update encoder $w_A$ and classifier $c_A$ via cross-entropy loss$(v_A, y_A$ & $yc_A)$;
   Update encoder $w_B$ and classifier $c_B$ via cross-entropy loss$(v_B, y_B$ & $yc_B)$;
   Update encoder $w_C$ and classifier $c_C$ via cross-entropy loss$(v_C, y_C$ & $yc_C)$.

---

Table 6: The detailed test classification accuracy (%) compared with baselines on four individual large datasets (values in parentheses are standard deviations). The best results are in **bold**.

| Dataset | Noise | Standard | Mixup | Co-teaching | FINE | SREA | SELC | CULCU | Scale-teaching |
|---------|-------|----------|-------|-------------|------|------|------|-------|----------------|
| HAR | Sym 20% | 92.13 (0.64) | 92.52 (1.05) | 92.28 (0.67) | 92.15 (0.55) | 92.53 (1.41) | 92.88 (0.82) | 92.66 (0.37) | **93.93 (0.66)** |
| | Sym 50% | 83.99 (2.89) | 76.75 (1.88) | 89.90 (1.63) | 88.42 (3.83) | **91.38 (0.59)** | 90.37 (0.73) | 89.91 (2.19) | 90.17 (0.67) |
| | Asym 40% | 75.59 (5.39) | 66.91 (2.61) | 87.67 (2.52) | 83.87 (5.98) | 88.98 (0.57) | 87.67 (2.39) | 87.22 (1.22) | **89.62 (0.73)** |
| | Ins 40% | 83.56 (2.82) | 73.86 (0.89) | 90.98 (0.96) | 90.77 (0.33) | 91.25 (1.11) | 91.02 (1.53) | 91.15 (1.43) | **91.58 (1.47)** |
| UniMiB-SHAR | Sym 20% | 87.07 (0.95) | 82.13 (1.08) | 80.54 (2.16) | 26.63 (3.07) | 51.48 (3.65) | 68.52 (2.86) | 82.80 (1.87) | **90.69 (1.02)** |
| | Sym 50% | 79.37 (0.41) | 77.77 (1.59) | 66.33 (2.85) | 18.92 (4.61) | 47.62 (3.33) | 67.65 (3.31) | 66.36 (3.91) | **81.31 (0.67)** |
| | Asym 40% | 63.59 (4.13) | 66.32 (1.93) | 60.25 (1.45) | 19.18 (4.37) | 51.16 (3.01) | 55.65 (1.59) | 60.45 (1.65) | **70.68 (2.15)** |
| | Ins 40% | 55.83 (8.14) | 56.97 (6.48) | 54.09 (3.79) | 11.18 (4.75) | 51.5 (1.98) | 54.62 (6.63) | 53.90 (4.75) | **71.14 (3.99)** |
| FD-A | Sym 20% | 98.89 (0.05) | 99.78 (0.06) | 99.83 (0.08) | 78.13 (21.47) | 89.92 (0.68) | 99.67 (0.09) | 99.85 (0.08) | **99.93 (0.04)** |
| | Sym 50% | 96.63 (1.16) | 98.73 (0.62) | 99.04 (0.32) | 70.65 (17.53) | 82.18 (0.01) | 98.59 (0.25) | 99.06 (0.29) | **99.38 (0.53)** |
| | Asym 40% | 96.12 (1.65) | 93.50 (1.85) | 97.06 (4.05) | 61.04 (14.24) | 90.23 (0.02) | 98.24 (0.58) | 98.91 (0.42) | **99.55 (0.36)** |
| | Ins 40% | 99.36 (0.47) | 99.55 (0.10) | 99.51 (0.19) | 67.81 (12.95) | 88.63 (0.02) | 99.36 (0.23) | 99.53 (0.22) | **99.82 (0.06)** |
| Sleep-EDF | Sym 20% | 85.01 (0.09) | 84.31 (0.36) | 84.81 (0.14) | 81.21 (0.28) | 72.79 (0.99) | 84.32 (0.33) | 85.23 (0.14) | **85.56 (0.35)** |
| | Sym 50% | 83.58 (0.74) | 83.61 (0.39) | 83.39 (0.25) | 78.17 (4.42) | 72.78 (1.30) | 83.06 (0.29) | 84.02 (0.53) | **84.59 (0.97)** |
| | Asym 40% | 79.62 (2.39) | 77.40 (1.92) | 82.87 (0.40) | 64.77 (2.10) | 72.23 (0.89) | 82.50 (1.07) | 83.05 (0.64) | **83.87 (0.38)** |
| | Ins 40% | 84.35 (0.38) | 84.25 (0.31) | 84.62 (0.28) | 79.68 (2.55) | 71.99 (1.24) | 83.78 (0.28) | 84.86 (0.22) | **85.03 (0.61)** |

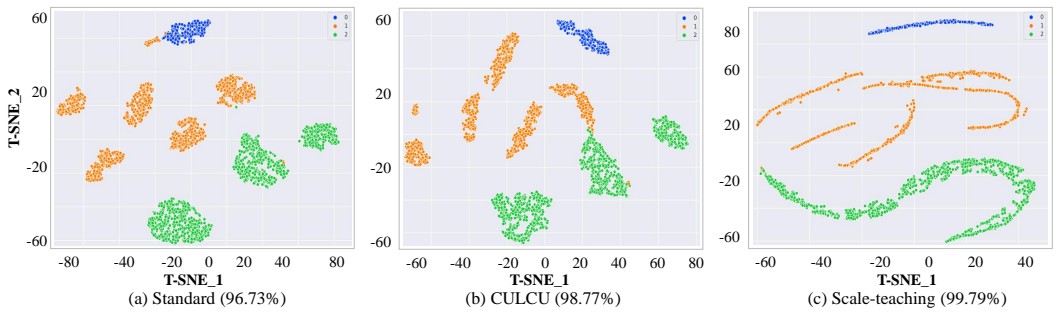

(a) Standard (96.73%)  (b) CULCU (98.77%)  (c) Scale-teaching (99.79%)

Figure 6: t-SNE visualization of the learned embeddings on the FD-A dataset with Sym 50% noisy labels (values in parentheses are the test classification accuracies).

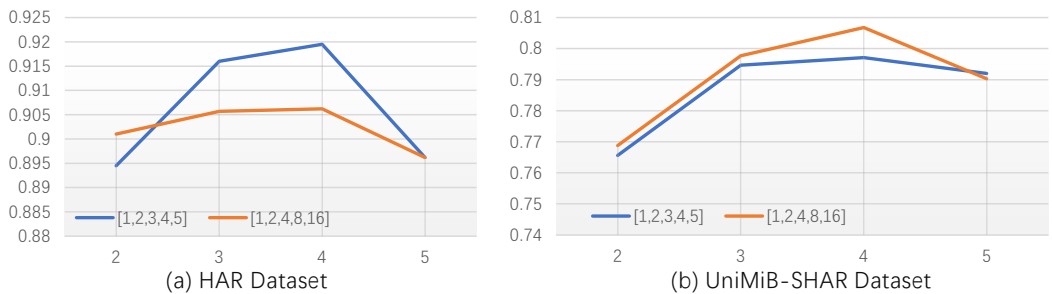

(a) HAR Dataset  (b) UniMiB-SHAR Dataset

Figure 7: Multi-scale sampling strategies analysis under Sym 50% noisy labels.

time-series label-noise learning can make the embeddings of different classes more discriminative, thus facilitating clean sample selection and noisy label correction.

**Impact of downsampling scale sequence list.**    Scale-teaching can be performed using a variety of different downsampling scales for label-noise learning. Based on the experience of [19, 44] on time series classification and prediction tasks, we utilize the downsampling scales of [1,2,4] for the experimental analyses of Scale-teaching. However, for real-world scenarios that actually contain noisy labels, it is generally not possible to perform hyperparametric analyses using a clean-labeled validation set. To facilitate the analysis, in this paper, we use the classification performance of the test set for multi-scale hyperparameter analyses. However, to avoid test set information leakage, we do not use the hyperparameter analysis result for Scale-teaching in our experiments. We use two multi-scale sampling strategies for analyses, which are (1) {[1,2], [1,2,3], [1,2,3,4], [1,2,3,4,5]}; (2) {[1,3], [1,2,4], [1,2,4,8], [1,2,4,8,16]}. From Figure 7, we find that Scale-teaching using four different scales for training has the highest classification accuracy, which indicates that more input scales do not necessarily make the classification performance better. In addition, using three or four scales of sequences can effectively improve the classification performance of Scale-teaching compared with using two different scales.

**Impact of input scales of sequences order.**    Scale-teaching employs a finer-to-coarser strategy for cross-scale embedding fusion. Intuitively, when a single scale is used for classification, the original single scale (finer) time series is better overall because it does not discard the original sequence information compared to coarser scale time series. Therefore, Scale-teaching is trained using the finer-to-coarser cross-scale fusion strategy. To analyze the difference in classification performance between different fusion directions, we subtract the classification accuracy using the finer-to-coarser and coarser-to-finer training approaches, and the specific results are shown in Figure 8. We can find that the classification performance of finer-to-coarser is better overall, which is due to its ability to use a single fine-scale sequence with an excellent classification performance from the beginning to gradually promote the classification performance of multiscale fusion embeddings.

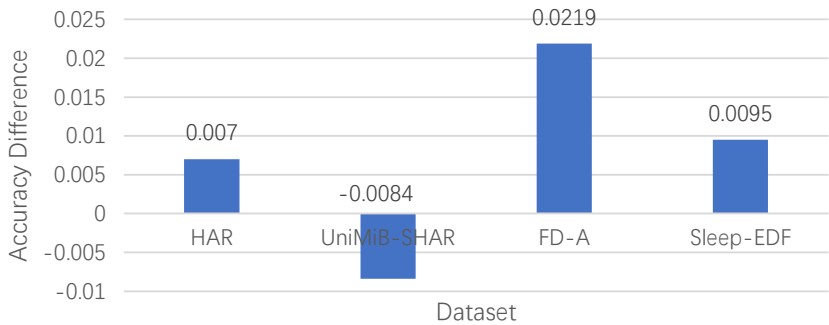

Figure 8: The cross fusion direction of input scale series analysis under Sym 50% noisy labels.

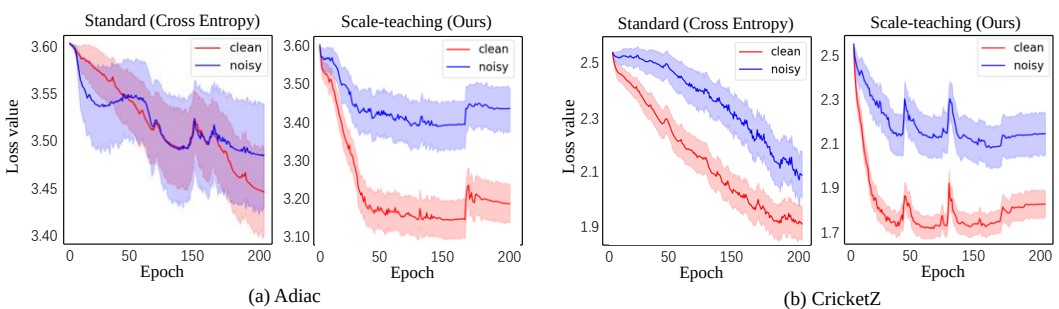

Figure 9: The change of loss values for clean and noisy time series samples under Aysm 40% noise labels. The solid line and shading indicate the mean and standard deviation loss values of all clean (or noisy) training samples within each epoch.

## F   Small-loss Visualization

The small-loss criterion has been extensively validated for clean label selection in label-noise learning for computer vision. To further analyze the application of the small-loss criterion in time series data, we provide the change of loss values of the models trained by the Standard method and Scale-teaching on Adiac and CricketZ UCR datasets (as in Figure 9). Also, we visualize the probability distributions of the ground-truth clean and noisy (corrupted) sample loss values on the test set with different training strategies. Specifically, Figures 10 and 11 show the loss probability distributions of the models trained by different strategies on the HAR dataset and UniMiB-SHAR with Aysm 40% noisy labels. Both red (clean) and blue (corrupted) in Figure 10 and Figure 11 contain two peaks, which indicate that some correctly labeled samples are still difficult to learn (large loss) and some incorrectly labeled samples are also easy to learn (small loss). Compared with the Standard method (Figure 10 (a) and Figure 11 (a)), Scale-teaching (Figure 10 (b) and Figure 11 (b)) can clearly distinguish clean and noisy samples by the loss value distribution, further validating the robustness of the multi-scale embeddings to cope with time-series noisy labels.

## G   Other Analysis

**The test F1-score results of ablation study.**    Following [4], we select the averaged F1-score on the test set as a new metric for ablation analysis in Section 4.5. Hence, we give the corresponding test classification F1-score (%) in Tables 7 and 8.

**Running time analysis.**    We select two datasets for running the time-consuming analysis, the FD-A dataset with the largest sequence length and the Sleep-EDF dataset with the largest samples. We performed the running time statistics on the NVIDIA GeForce RTX 3090 GPU using all baselines, and the results are shown in Table 9. On the FD-A dataset with the longest sequence length, Co-teaching and CULCU take essentially twice as long to run as the Stanard method because they use

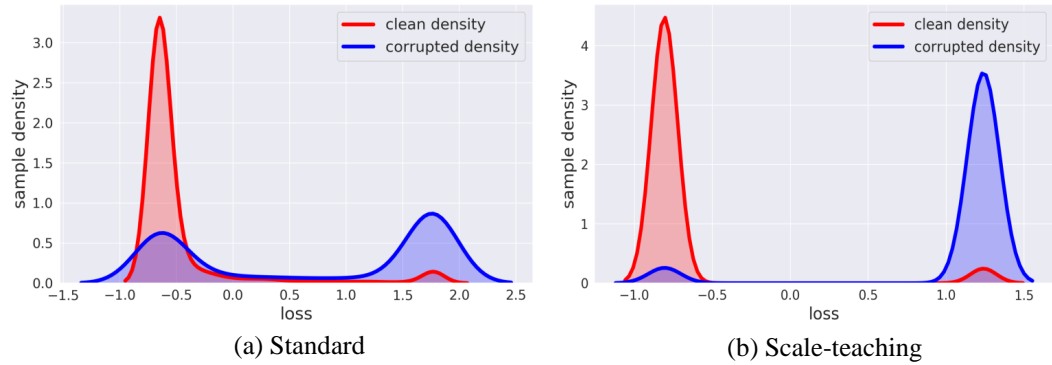

(a) Standard                (b) Scale-teaching

Figure 10: The loss value probability distributions visualization on HAR dataset with Asym 40% noisy labels.

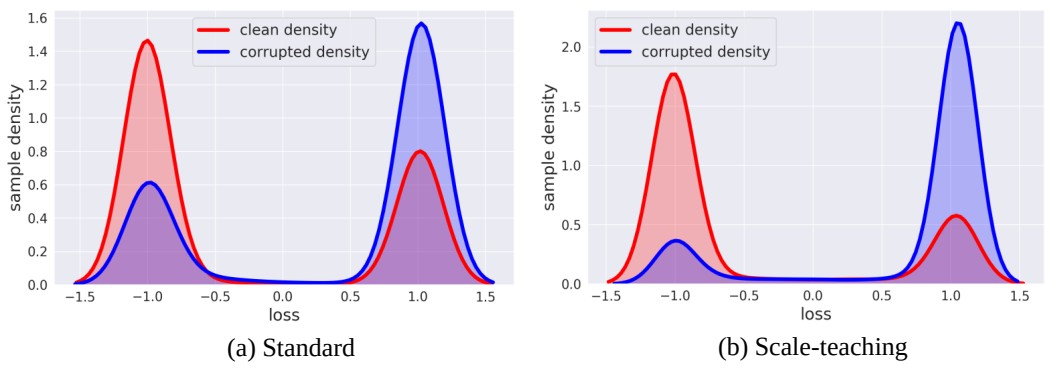

(a) Standard                (b) Scale-teaching

Figure 11: The loss value probability distributions visualization on UniMiB-SHAR dataset with Asym 40% noisy labels.

Table 7: The test classification F1-score (%) results of ablation study (values in parentheses denote drop F1-score).

| Method | HAR | | UniMiB-SHAR | |
|---|---|---|---|---|
| | Sym 50% | Asym 40% | Sym 50% | Asym 40% |
| Scale-teaching | **90.05** | **89.14** | **77.56** | **65.89** |
| w/o cross-scale fusion | 88.16 (-1.89) | 87.05 (-2.09) | 68.23 (-9.33) | 57.76 (-8.13) |
| only single scale | 87.56 (-2.49) | 86.75 (-2.39) | 66.87 (-10.69) | 54.12 (-11.77) |
| w/o graph learning | 87.79 (-2.26) | 87.41 (-1.73) | 74.62 (-2.94) | 63.15 (-2.74) |
| w/o moment | 89.34 (-0.71) | 88.27 (-0.87) | 76.67 (-0.89) | 64.92 (-0.97) |
| w/o dynamic threshold | 88.93 (-1.12) | 88.29 (-0.85) | 73.11 (-4.45) | 64.76 (-1.17) |

Table 8: The test classification F1-score (%) results on four individual large datasets without noisy labels. The best results are **bold**, and the second best results are underlined.

| Dataset | Standard | Mixup | Co-teaching | FINE | SREA | SELC | CULCU | Scale-teaching |
|---|---|---|---|---|---|---|---|---|
| HAR | 93.27 | **95.39** | 93.75 | 93.19 | 92.91 | 93.71 | 94.72 | 94.18 |
| UniMiB-SHAR | 86.37 | 80.17 | 84.43 | 84.03 | 66.54 | 89.19 | 86.45 | **93.62** |
| FD-A | 99.93 | 99.91 | **99.96** | 64.05 | 90.14 | 99.82 | 99.95 | **99.96** |
| Sleep-EDF | 81.99 | 82.11 | 82.52 | 83.07 | 77.67 | 82.17 | 83.26 | **84.76** |

Table 9: Training time (hours) analysis using the FD-A and Sleep-EDF datasets with Asym 40% noisy labels.

| Dataset | Standard | Mixup | Co-teaching | FINE | SREA | SELC | CULCU | Scale-teaching | | |
| --- | --- | --- | --- | --- | --- | --- | --- | --- | --- | --- |
| | | | | | | | | [1,2,4] | [1,4,16] | [1,8,32] |
| FD-A | **0.37** | 0.42 | 0.79 | 0.63 | 0.90 | 0.42 | 0.87 | 1.06 | 0.86 | 0.82 |
| Sleep-EDF | **0.54** | 0.83 | 1.09 | 2.04 | 1.64 | 0.73 | 1.47 | 2.02 | 1.76 | 1.60 |

Table 10: Comparison with classification methods without label noise learning strategy. The best test classification accuracy (%) results are **bold**, and the second best results are underlined.

| Dataset | HAR | | | | | FD-A | | | | |
| --- | --- | --- | --- | --- | --- | --- | --- | --- | --- | --- |
| Method | 0 | Sym 20% | Sym 50% | Asym 40% | Ins 40% | 0 | Sym 20% | Sym 50% | Asym 40% | Ins 40% |
| Boss [15] | 72.34 | 62.55 | 56.11 | 53.29 | 52.34 | 69.75 | 64.75 | 57.95 | 61.99 | 62.25 |
| Rocket [60] | **95.29** | 92.93 | 90.04 | 82.53 | 90.43 | **99.99** | 99.71 | 97.01 | 89.75 | 97.98 |
| FCN [57] | 93.74 | 92.13 | 83.99 | 75.59 | 83.56 | 99.56 | 98.89 | 96.63 | 96.12 | 99.36 |
| Scale-teaching | 94.72 | **93.93** | **90.17** | **89.62** | **91.58** | 99.98 | **99.93** | **99.38** | **99.55** | **99.82** |

two encoders. Furthermore, although SREA uses a single network training, it utilizes a decoder for the unsupervised reconstruction task of the original time series, which significantly increases training time on the FD-A dataset with longer sequences. running time is higher than Co-teaching. The Scale-teaching paradigm uses multiple encoders for training and has an additional noisy label correction module, which is expected to increase the training time. Nevertheless, the larger the sampling scale (coarse scale) of the training data used by the Scale-teaching paradigm, the lower the training elapsed time of its model. For example, with input scales of [1, 8, 32], the training time of Scale-teaching is lower than that of CULCU and SREA. On the SleepEEG dataset with the largest number of samples, we find that FINE with an encoder has a higher running time because FINE using all training samples to select clean labels is time-consuming when the sample size is large. In contrast, the runtime of Scale-teaching is lower than FINE. Also, when the input scales are set to [1,8,32], the runtime of Scale-teaching is lower than SREA.

It is worth noting that when Scale-teaching is trained using two scales, such as [1,2] or [1,16], its training run time decreases further. From the analysis in Appendix E, it is clear that Scale-teaching using three different scales generally performs better than two scales for classification with noisy labels. In addition, the classification performance of [1,2,4], [1,4,16], and [1,8,32] when Scale-teaching is trained using three different scales has less difference in classification performance on datasets with longer sequences (e.g., FD-A and Sleep-EDF). The above results indicate that the Scale-teaching paradigm has a greater advantage in runtime on time-series datasets with longer sequences.

**Robustness analysis.** Three time-series supervised classification methods (Boss [15], Rocket [60] and FCN [57]) and the Scale-teaching paradigm are chosen for robustness analysis against time-series noise labels. Boss [15] is a time series classification method based on similarity search, which can effectively mitigate the negative impact of noise (e.g., adding Gaussian noise) in time series values on classification. Rocket [60] uses a large number of randomly initialized convolution kernels to extract time series features, and employs the extracted features to classify time series using a machine learning classifier (e.g., Ridge classifier). FCN [57] is the encoder used by Scale-teaching, which is a time series classification method based on DNNs. As shown in Table 10, the classification performance of the Scale-teaching paradigm using FCN as encoders is better than that of Boss, Rocket and FCN in the presence of noisy labels. It is worth noting that both Boss and Rocket training processes are independent of the optimization of DNNs. However, their classification performance is still reduced due to the influence of noisy labels. In addition, the encoder of the Scale-teaching paradigm can be designed flexibly, such as using ResNet [1], InceptionTime [65] and OS-CNN [20] in the field of time series classification. In other words, using better robustness encoders, the classification performance of Scale-teaching can be further improved with time-series noise labels.

Table 11: The test classification accuracy (%) on UCR archive with Sym 20% noisy labels.

| ID | Dataset | Standard | MixUp | Co-teaching | FINE | SREA | SELC | CULCU | Scale-teaching |
|---|---|---|---|---|---|---|---|---|---|
| 1 | ACSF1 | 71.44 | 73.28 | 67.00 | 10.00 | 54.00 | 70.40 | **73.68** | 66.20 |
| 2 | Adiac | 14.63 | 12.04 | 11.51 | 2.30 | 3.07 | 19.18 | 33.06 | **52.43** |
| 3 | AllGestureWiimoteX | 49.68 | 45.01 | 50.67 | 10.00 | 46.43 | 44.25 | **56.66** |
| 4 | AllGestureWiimoteY | 61.37 | 53.29 | 57.01 | 14.03 | 16.91 | 61.14 | 50.07 | **64.94** |
| 5 | AllGestureWiimoteZ | 55.09 | 50.13 | 52.34 | 18.20 | 20.09 | 51.00 | 47.42 | **64.31** |
| 6 | ArrowHead | 65.53 | **67.77** | 60.72 | 32.11 | 61.14 | 66.74 | 62.31 | 61.71 |
| 7 | BME | 48.05 | 51.73 | 47.77 | 47.47 | 50.67 | 49.87 | 48.73 | **79.33** |
| 8 | Beef | 46.67 | 47.33 | **49.33** | 20.00 | 33.33 | 46.67 | 44.47 | 38.00 |
| 9 | BeetleFly | 75.00 | 72.20 | 78.00 | 50.00 | 81.00 | 75.00 | 68.00 | **85.00** |
| 10 | BirdChicken | 86.00 | 86.80 | 89.50 | 50.00 | 75.00 | 86.00 | 83.50 | **95.00** |
| 11 | CBF | 82.47 | 83.82 | **88.81** | 33.38 | 84.18 | 83.53 | 76.75 | 87.24 |
| 12 | Car | 65.33 | 66.40 | **68.67** | 25.00 | 23.33 | 65.00 | 53.90 | 67.33 |
| 13 | Chinatown | 70.13 | 81.55 | 83.28 | 36.52 | 72.46 | 82.43 | **83.54** | 73.91 |
| 14 | ChlorineConcentration | 60.87 | **61.19** | 59.64 | 47.34 | 55.08 | 58.51 | 57.82 | 61.17 |
| 15 | CinCECGTorso | 62.90 | 61.62 | 63.27 | 27.06 | 39.07 | **64.51** | 54.59 | 55.06 |
| 16 | Coffee | 83.43 | 87.43 | 87.43 | 49.29 | 69.29 | 91.43 | 88.57 | **100.00** |
| 17 | Computers | 72.06 | 73.49 | 71.44 | 57.04 | 70.00 | **74.40** | 71.24 | 72.00 |
| 18 | CricketX | 57.10 | 46.46 | 46.35 | 9.95 | 33.33 | 53.59 | 62.91 | **68.67** |
| 19 | CricketY | 56.80 | 44.05 | 31.64 | 8.62 | 13.08 | 49.74 | **59.88** | 59.23 |
| 20 | CricketZ | 51.53 | 37.46 | 37.82 | 8.46 | 26.41 | 45.13 | 53.71 | **69.95** |
| 21 | Crop | 72.84 | 73.24 | 73.29 | 69.97 | 65.65 | 72.07 | 72.25 | **74.44** |
| 22 | DiatomSizeReduction | 63.05 | 71.27 | 61.54 | 69.72 | 67.32 | 70.13 | 59.29 | **82.68** |
| 23 | DistalPhalanxOutlineAgeGroup | 70.13 | 70.39 | 71.23 | 50.50 | 69.06 | 72.09 | **72.83** | 66.62 |
| 24 | DistalPhalanxOutlineCorrect | 72.10 | 69.20 | 75.91 | 48.33 | 42.75 | 75.22 | 75.22 | **79.13** |
| 25 | DistalPhalanxTW | 66.78 | 66.16 | 64.72 | 29.78 | 58.27 | **67.34** | 65.74 | 62.59 |
| 26 | DodgerLoopDay | 24.40 | 28.50 | 33.55 | 15.50 | 15.00 | 31.25 | 35.10 | **36.25** |
| 27 | DodgerLoopGame | 57.14 | 62.03 | 65.71 | 50.43 | 52.17 | **68.84** | 66.16 | 50.00 |
| 28 | DodgerLoopWeekend | 85.07 | 80.29 | 85.61 | 45.22 | 33.33 | 85.80 | 86.16 | **86.96** |
| 29 | ECG200 | 79.00 | 78.32 | **82.60** | 58.40 | 79.20 | 79.80 | 82.10 | 77.00 |
| 30 | ECG5000 | 92.37 | 91.23 | 92.93 | 44.55 | 90.92 | 92.76 | 93.13 | **94.11** |
| 31 | ECGFiveDays | 77.11 | **77.19** | 74.38 | 49.71 | 71.17 | 76.54 | 64.21 | 70.40 |
| 32 | EOGHorizontalSignal | 49.08 | 42.81 | 47.42 | 41.22 | 10.36 | 40.61 | 41.91 | **52.49** |
| 33 | EOGVerticalSignal | 34.34 | 30.57 | 33.12 | 30.39 | 11.48 | 30.66 | 30.06 | **37.18** |
| 34 | Earthquakes | 70.65 | 68.78 | 74.53 | 45.04 | **74.82** | 72.81 | 74.62 | 74.82 |
| 35 | ElectricDevices | 72.17 | **72.97** | 72.72 | 63.62 | 64.35 | 72.44 | 72.48 | 70.35 |
| 36 | EthanolLevel | 47.78 | 37.52 | 46.61 | 29.00 | 25.20 | 33.20 | 46.66 | **57.12** |
| 37 | FaceAll | 81.32 | **85.84** | 82.44 | 10.62 | 71.98 | 85.63 | 82.97 | 75.57 |
| 38 | FaceFour | 71.23 | **78.68** | 65.39 | 24.09 | 54.55 | 75.68 | 65.00 | 51.59 |
| 39 | FacesUCR | 77.40 | 60.71 | 65.10 | 10.35 | 34.73 | 69.41 | 75.38 | **80.37** |
| 40 | FiftyWords | 34.92 | 28.03 | 38.80 | 15.63 | 21.41 | 32.09 | 39.39 | **50.95** |
| 41 | Fish | 68.94 | 69.71 | 69.77 | 13.60 | 16.00 | 61.71 | 60.65 | **72.11** |
| 42 | FordA | 89.74 | 90.00 | 90.02 | 64.18 | 86.20 | 90.76 | 90.34 | **92.35** |
| 43 | FordB | 75.20 | 75.89 | 76.21 | 62.91 | 59.63 | 77.90 | 75.57 | **80.00** |
| 44 | FreezerRegularTrain | 91.76 | **95.56** | 92.97 | 61.60 | 76.21 | 88.20 | 93.81 | 84.07 |
| 45 | FreezerSmallTrain | 69.36 | 71.58 | **76.13** | 64.18 | 75.79 | 69.52 | 75.23 | 59.81 |
| 46 | Fungi | 39.87 | 22.11 | 24.62 | 24.52 | 23.66 | 31.18 | **44.09** | 26.34 |
| 47 | GestureMidAirD1 | 31.31 | 28.62 | 25.31 | 24.00 | 19.23 | 33.85 | 32.20 | **43.23** |
| 48 | GestureMidAirD2 | 29.23 | 23.42 | 21.92 | 3.85 | 13.08 | 26.92 | 27.51 | **34.62** |
| 49 | GestureMidAirD3 | 16.31 | 14.49 | 14.52 | 14.00 | 8.46 | 15.38 | 20.08 | **23.85** |
| 50 | GesturePebbleZ1 | 54.88 | 52.07 | 44.12 | 29.90 | 57.21 | 59.88 | **78.35** | 72.56 |
| 51 | GesturePebbleZ2 | 73.39 | 73.11 | 74.44 | 32.78 | 57.72 | 75.95 | 75.09 | **81.27** |
| 52 | GunPoint | 71.79 | 76.53 | 69.47 | 49.87 | 57.33 | **81.47** | 70.24 | 76.00 |
| 53 | GunPointAgeSpan | 74.33 | 85.72 | 75.57 | 53.54 | 50.63 | 83.61 | **89.11** | 58.23 |
| 54 | GunPointMaleVersusFemale | 85.65 | 92.68 | 93.82 | 68.48 | 47.47 | 94.24 | **96.65** | 96.20 |
| 55 | GunPointOldVersusYoung | 95.96 | 99.62 | 98.90 | **100.00** | 47.62 | 99.56 | **100.00** | 100.00 |
| 56 | Ham | 63.62 | 65.18 | 67.24 | 49.71 | 57.90 | **68.76** | 65.62 | 63.81 |
| 57 | HandOutlines | 76.53 | 70.30 | 75.97 | 58.66 | 64.05 | 68.92 | 70.29 | **82.05** |
| 58 | Haptics | 38.61 | **39.27** | 38.06 | 19.87 | 26.62 | 37.27 | 36.81 | 38.70 |
| 59 | Herring | 54.94 | 62.69 | 59.69 | 55.63 | 59.38 | 62.81 | 59.38 | **66.56** |
| 60 | HouseTwenty | 79.52 | 83.13 | 83.12 | 48.40 | 57.98 | 84.03 | **84.81** | 67.39 |
| 61 | InlineSkate | 27.12 | 23.24 | 25.24 | 13.85 | 18.36 | 25.64 | 28.19 | **29.49** |
| 62 | InsectEPGRegularTrain | 99.37 | 95.98 | **100.00** | 96.63 | 100.00 | 96.63 | **100.00** | 100.00 |
| 63 | InsectEPGSmallTrain | 72.17 | 73.25 | 93.88 | 93.25 | 95.46 | 77.03 | 95.46 | **100.00** |
| 64 | InsectWingbeatSound | 31.69 | 28.82 | 30.47 | 9.09 | 11.10 | 32.07 | 29.18 | **42.66** |
| 65 | ItalyPowerDemand | 75.12 | 78.53 | 84.47 | 49.91 | 83.45 | 90.38 | **90.79** | 89.08 |
| 66 | LargeKitchenAppliances | 85.79 | 85.91 | 87.07 | 43.71 | 67.24 | **87.20** | 83.18 | 86.67 |
| 67 | Lightning2 | 56.28 | 63.21 | 62.62 | 52.46 | 59.61 | **64.26** | 63.77 | 62.30 |
| 68 | Lightning7 | 50.73 | 55.01 | 56.16 | 21.10 | 50.96 | **56.44** | 51.97 | 53.97 |
| 69 | Mallat | 49.13 | 43.14 | 47.91 | 12.48 | 13.76 | 59.28 | 50.63 | **73.17** |
| 70 | Meat | 73.40 | 72.00 | 66.67 | 35.33 | 33.33 | 71.67 | 61.73 | **82.33** |
| 71 | MedicalImages | 66.11 | 61.14 | 67.42 | 34.47 | 51.45 | 65.66 | 61.89 | **70.45** |
| 72 | MelbournePedestrian | 90.99 | 91.95 | 91.82 | 56.87 | 29.00 | 90.29 | 90.16 | **95.18** |
| 73 | MiddlePhalanxOutlineAgeGroup | 50.49 | 49.77 | 57.95 | 49.48 | **61.69** | 58.31 | 56.52 | 49.61 |
| 74 | MiddlePhalanxOutlineCorrect | 73.79 | 75.01 | 77.56 | 48.59 | 57.04 | **77.87** | 74.31 | 67.01 |
| 75 | MiddlePhalanxTW | 53.17 | 52.44 | 54.48 | 28.05 | 55.84 | **55.97** | 54.22 | 50.91 |
| 76 | MixedShapesRegularTrain | 93.09 | 93.43 | 93.10 | 21.87 | 54.62 | 92.54 | 92.11 | **94.25** |
| 77 | MixedShapesSmallTrain | **80.06** | 72.49 | 69.00 | 20.59 | 35.86 | 75.47 | 77.16 | 70.95 |
| 78 | MoteStrain | 75.20 | 78.42 | 80.15 | 50.78 | 79.85 | 80.38 | **82.33** | 53.91 |
| 79 | NonInvasiveFetalECGThorax1 | 36.94 | 24.37 | 11.79 | 2.44 | 4.99 | 38.12 | 40.73 | **87.54** |
| 80 | NonInvasiveFetalECGThorax2 | 39.71 | 24.79 | 13.56 | 2.24 | 10.52 | 31.65 | 42.42 | **81.01** |
| 81 | OSULeaf | **88.94** | 85.52 | 87.60 | 15.21 | 41.49 | 87.60 | 87.73 | 82.89 |
| 82 | OliveOil | 44.67 | 42.00 | 42.00 | 40.00 | 40.00 | 40.00 | 43.67 | **66.67** |
| 83 | PLAID | 36.09 | 38.13 | 38.73 | 25.88 | 25.88 | 37.09 | **39.61** | 22.72 |
| 84 | PhalangesOutlinesCorrect | 77.26 | 77.60 | 76.36 | 47.74 | 65.29 | 77.62 | **77.69** | 71.93 |
| 85 | Phoneme | 27.10 | 24.62 | 27.13 | 19.76 | 11.18 | 25.63 | **27.60** | 25.45 |
| 86 | PickupGestureWiimoteZ | 52.24 | 42.00 | 38.00 | 10.00 | 20.00 | 44.00 | 56.48 | **66.00** |
| 87 | PigAirwayPressure | 12.56 | 9.37 | 12.02 | 11.23 | 3.85 | 12.02 | **20.82** | 14.52 |
| 88 | PigArtPressure | 18.98 | 11.63 | 11.54 | 23.54 | 3.85 | 15.38 | 28.96 | **29.23** |
| 89 | PigCVP | 10.83 | 9.85 | 9.62 | **18.79** | 8.65 | 12.98 | 13.01 | 15.38 |
| 90 | Plane | 95.55 | 91.62 | 93.48 | 12.19 | 82.29 | 91.43 | 94.38 | **100.00** |
| 91 | PowerCons | 78.02 | 76.49 | 84.94 | 50.00 | **85.22** | 82.22 | 82.86 | 83.22 |
| 92 | ProximalPhalanxOutlineAgeGroup | 82.22 | 81.78 | 84.24 | 37.17 | **85.37** | 84.88 | 83.74 | 78.63 |
| 93 | ProximalPhalanxOutlineCorrect | 84.37 | 83.92 | 85.45 | 53.68 | 67.29 | 84.88 | **85.79** | 80.48 |
| 94 | ProximalPhalanxTW | 77.78 | 79.57 | 78.34 | 25.27 | 67.80 | **79.80** | 78.38 | 79.51 |
| 95 | RefrigerationDevices | 51.19 | 50.23 | 53.86 | 33.33 | **54.36** | 54.13 | 47.35 | 52.48 |
| 96 | Rock | 41.64 | 40.08 | 41.84 | 25.60 | 28.00 | 41.60 | **43.28** | 34.00 |
| 97 | ScreenType | 63.14 | 61.79 | **63.66** | 33.33 | 42.13 | 62.56 | 61.19 | 57.44 |
| 98 | SemgHandGenderCh2 | 73.73 | 74.07 | 71.61 | 67.82 | 65.00 | 73.07 | 72.84 | **74.10** |
| 99 | SemgHandMovementCh2 | 47.93 | 47.00 | 52.93 | 38.18 | 23.47 | 48.00 | 50.03 | **57.47** |
| 100 | SemgHandSubjectCh2 | 61.80 | 58.77 | 60.73 | 32.18 | 28.00 | 60.44 | 58.02 | **71.02** |
| 101 | ShakeGestureWiimoteZ | 63.32 | 60.24 | 51.60 | 20.40 | 44.00 | 68.00 | **73.72** | 63.20 |
| 102 | ShapeletSim | 61.11 | 71.76 | 79.67 | 50.00 | **85.20** | 67.33 | 60.01 | 80.00 |
| 103 | ShapesAll | 45.85 | 32.73 | 45.33 | 1.67 | 1.67 | 36.67 | 37.96 | **74.77** |
| 104 | SmallKitchenAppliances | 73.53 | 78.66 | 78.86 | 54.77 | 75.81 | **80.75** | 77.91 | 80.00 |
| 105 | SmoothSubspace | 76.00 | 85.20 | 92.13 | 33.33 | 88.53 | 90.00 | **92.67** | 90.00 |
| 106 | SonyAIBORobotSurface1 | 77.13 | 83.54 | **87.07** | 45.76 | 84.65 | 83.66 | 83.48 | 78.74 |
| 107 | SonyAIBORobotSurface2 | 76.63 | 82.92 | 87.18 | 47.66 | 81.94 | 83.88 | 87.56 | **92.24** |
| 108 | StarLightCurves | 96.87 | 97.08 | 97.14 | 57.32 | 85.39 | **97.19** | 96.95 | 95.82 |
| 109 | Strawberry | 91.37 | 92.52 | 91.19 | 52.86 | 70.65 | 93.03 | 92.37 | **93.41** |
| 110 | SwedishLeaf | 89.89 | 83.71 | 90.64 | 6.27 | 33.88 | 88.83 | 85.42 | **95.36** |
| 111 | Symbols | **76.37** | 63.00 | 67.16 | 22.44 | 70.37 | 60.16 | 69.00 | 60.80 |
| 112 | SyntheticControl | 91.47 | 93.19 | 97.80 | 24.00 | **98.48** | 96.20 | 97.90 | 96.33 |
| 113 | ToeSegmentation1 | 84.12 | 79.30 | 84.51 | 50.53 | 83.26 | 84.91 | **85.48** | 73.68 |
| 114 | ToeSegmentation2 | 77.23 | 74.43 | 83.23 | 43.69 | **84.74** | 77.69 | 82.92 | 83.85 |
| 115 | Trace | 95.16 | 92.24 | 91.20 | 26.00 | 84.20 | 90.00 | **98.60** | 92.00 |
| 116 | TwoLeadECG | 86.50 | 83.13 | 83.63 | 49.96 | 84.27 | **87.39** | 84.11 | 69.39 |
| 117 | TwoPatterns | 80.00 | 67.35 | 86.35 | 24.95 | 86.07 | 85.31 | 85.68 | **90.63** |
| 118 | UMD | 83.08 | 83.19 | 81.60 | 53.47 | 50.00 | **84.17** | 77.58 | 78.06 |
| 119 | UWaveGestureLibraryAll | 74.12 | 68.99 | 72.08 | 12.38 | 12.62 | 61.99 | 67.12 | **80.18** |
| 120 | UWaveGestureLibraryX | 70.30 | 70.79 | **74.35** | 12.65 | 30.35 | 71.26 | 70.92 | 72.26 |
| 121 | UWaveGestureLibraryY | 62.03 | 61.03 | 61.43 | 12.53 | 42.08 | 60.30 | 60.20 | **63.98** |
| 122 | UWaveGestureLibraryZ | 68.63 | 67.50 | 68.23 | 12.42 | 48.92 | 65.36 | 65.78 | **71.31** |
| 123 | Wafer | 96.96 | **97.69** | 96.68 | 73.53 | 89.21 | 97.45 | 96.57 | 88.07 |
| 124 | Wine | 59.70 | 54.89 | **61.11** | 50.00 | 50.00 | 50.74 | 53.04 | 54.07 |
| 125 | WordSynonyms | 34.04 | 29.93 | 34.48 | 14.42 | 23.04 | 30.44 | **39.42** | 34.95 |
| 126 | Worms | 62.42 | 67.58 | 66.83 | 22.08 | 59.74 | 67.27 | 62.73 | **75.32** |
| 127 | WormsTwoClass | 72.83 | 73.25 | 76.84 | 42.86 | 57.14 | 76.10 | 77.27 | **77.92** |
| 128 | Yoga | 73.48 | **73.56** | 71.73 | 47.86 | 53.57 | 71.04 | 69.52 | 65.40 |
| | **Avg Acc** | 65.02 | 63.80 | 65.06 | 36.01 | 50.39 | 65.67 | 66.23 | **68.70** |
| | **Avg Rank** | 4.15 | 4.33 | 3.61 | 7.50 | 6.16 | 3.48 | 3.54 | **3.02** |
| | **P-value** | 1.90E-04 | 4.06E-05 | 1.90E-03 | 1.49E-34 | 1.70E-17 | 3.04E-03 | 8.57E-03 | - |

Table 12: The test classification accuracy (%) on UCR archive with Sym 50% noisy labels.

| ID | Dataset | Standard | MixUp | Co-teaching | FINE | SREA | SELC | CULCU | Scale-teaching |
|---|---|---|---|---|---|---|---|---|---|
| 1 | ACSF1 | 41.00 | 45.00 | 36.96 | 10.00 | 29.60 | 47.00 | **47.20** | 42.00 |
| 2 | Adiac | 12.48 | 8.56 | 6.39 | 2.20 | 4.09 | 9.97 | 23.55 | **31.65** |
| 3 | AllGestureWiimoteX | 33.14 | 22.89 | 29.32 | 31.29 | 11.29 | 32.17 | 26.40 | **37.91** |
| 4 | AllGestureWiimoteY | 36.51 | 31.00 | 32.46 | 19.05 | 16.97 | 35.00 | 31.64 | **39.57** |
| 5 | AllGestureWiimoteZ | 38.14 | 28.69 | 32.02 | 11.29 | 14.57 | 37.49 | 33.63 | **42.95** |
| 6 | ArrowHead | 34.29 | 31.31 | 31.43 | 33.94 | 30.29 | 38.40 | 37.83 | **39.25** |
| 7 | BME | 33.23 | 13.73 | 34.00 | 35.73 | 35.87 | 33.33 | 33.33 | **36.80** |
| 8 | Beef | 23.33 | 21.33 | 29.67 | 20.00 | 20.00 | 30.33 | **31.00** | 30.00 |
| 9 | BeetleFly | 51.00 | 40.00 | 49.00 | 50.00 | 46.00 | **53.00** | 52.00 | 50.00 |
| 10 | BirdChicken | 59.00 | 45.00 | 60.50 | 50.00 | 55.00 | 61.00 | 52.50 | **71.00** |
| 11 | CBF | 61.35 | 44.33 | 58.24 | 33.29 | 63.22 | 60.93 | 53.36 | **64.30** |
| 12 | Car | 49.33 | 38.33 | 42.50 | 23.33 | 23.33 | **50.00** | 38.17 | 30.67 |
| 13 | Chinatown | 60.79 | 58.71 | 62.15 | **65.51** | 36.52 | 59.54 | 55.30 | 57.22 |
| 14 | ChlorineConcentration | 43.88 | 41.59 | 46.69 | 29.47 | **50.27** | 41.63 | 44.85 | 49.17 |
| 15 | CinCECGTorso | 42.43 | 44.10 | 34.84 | 24.91 | 31.44 | **45.55** | 31.86 | 40.47 |
| 16 | Coffee | 60.00 | 54.29 | 55.36 | 47.86 | 46.43 | 55.71 | 52.50 | **65.71** |
| 17 | Computers | 49.74 | 51.47 | **54.24** | 47.76 | 50.48 | 51.76 | 47.60 | 54.16 |
| 18 | CricketX | 31.32 | 25.20 | 24.77 | 8.36 | 13.08 | 28.05 | **43.39** | 39.73 |
| 19 | CricketY | 33.67 | 29.13 | 23.85 | 8.46 | 8.46 | 31.69 | 36.77 | **41.73** |
| 20 | CricketZ | 29.19 | 24.71 | 26.90 | 7.49 | 20.77 | 26.92 | **35.78** | 33.10 |
| 21 | Crop | 67.58 | 68.22 | 68.04 | 44.52 | 64.11 | 67.81 | 68.23 | **70.12** |
| 22 | DiatomSizeReduction | 54.34 | **63.23** | 47.78 | 33.53 | 50.65 | 61.50 | 47.68 | 57.49 |
| 23 | DistalPhalanxOutlineAgeGroup | 50.85 | 53.96 | 64.26 | 38.71 | 62.30 | 62.16 | **66.62** | 63.57 |
| 24 | DistalPhalanxOutlineCorrect | 51.52 | **52.32** | 51.59 | 48.33 | 48.33 | 48.04 | 50.93 | 50.87 |
| 25 | DistalPhalanxTW | 55.65 | 55.57 | 56.97 | 29.78 | 51.65 | 59.14 | 58.27 | **59.17** |
| 26 | DodgerLoopDay | 19.15 | 21.40 | 26.37 | 14.50 | 15.00 | 23.00 | 21.90 | **28.25** |
| 27 | DodgerLoopGame | 49.71 | 48.70 | 50.88 | 49.57 | 50.72 | 50.29 | **53.33** | 50.06 |
| 28 | DodgerLoopWeekend | 62.46 | 64.46 | 71.59 | 64.35 | 37.83 | 62.17 | 54.71 | **77.45** |
| 29 | ECG200 | 49.80 | 49.48 | 52.20 | 52.80 | 40.80 | 50.20 | 53.60 | **58.40** |
| 30 | ECG5000 | 69.14 | 76.27 | 90.83 | 39.94 | 90.15 | 88.90 | 90.50 | **91.58** |
| 31 | ECGFiveDays | 50.46 | 51.34 | **53.93** | 49.71 | 51.86 | 50.31 | 50.71 | 44.87 |
| 32 | EOGHorizontalSignal | 38.43 | 34.56 | 37.76 | 35.30 | 13.65 | 31.82 | 30.34 | **41.30** |
| 33 | EOGVerticalSignal | 26.25 | 22.59 | 23.72 | 20.32 | 11.49 | 23.31 | 19.25 | **29.73** |
| 34 | Earthquakes | 53.32 | 54.10 | 66.98 | **74.82** | 64.75 | 51.94 | 61.37 | 60.78 |
| 35 | ElectricDevices | 67.65 | 66.58 | 68.36 | 64.03 | 57.10 | 67.89 | **68.50** | 64.91 |
| 36 | EthanolLevel | 32.62 | 28.57 | 31.75 | 25.38 | 25.20 | 27.76 | 32.60 | **35.02** |
| 37 | FaceAll | 56.62 | 46.70 | 50.99 | 10.62 | 10.12 | **56.96** | 52.02 | 53.53 |
| 38 | FaceFour | 40.68 | 50.45 | 42.16 | 23.18 | 32.00 | **50.68** | 38.98 | 35.05 |
| 39 | FacesUCR | **44.98** | 42.32 | 42.96 | 11.29 | 18.86 | 44.81 | 43.64 | 41.57 |
| 40 | FiftyWords | 31.64 | 29.46 | 33.14 | 28.54 | 12.97 | 30.99 | 30.30 | **37.51** |
| 41 | Fish | 48.69 | **51.20** | 35.93 | 13.71 | 16.00 | 47.77 | 35.41 | 32.55 |
| 42 | FordA | 49.46 | 49.33 | **56.38** | 52.47 | 51.59 | 54.58 | 49.19 | 54.83 |
| 43 | FordB | 50.39 | 49.81 | 47.09 | 46.53 | 50.10 | **50.52** | 45.95 | 48.54 |
| 44 | FreezerRegularTrain | 38.15 | 53.98 | 48.56 | 54.20 | **55.21** | 54.46 | 45.08 | 47.16 |
| 45 | FreezerSmallTrain | 32.25 | 47.09 | 50.30 | 51.39 | 50.47 | 43.63 | 50.17 | **51.81** |
| 46 | Fungi | 15.59 | 17.20 | 16.32 | 18.34 | 10.75 | 15.59 | 17.92 | **19.18** |
| 47 | GestureMidAirD1 | 19.48 | 17.32 | 13.32 | 6.92 | 15.38 | 18.46 | 22.69 | **24.18** |
| 48 | GestureMidAirD2 | 17.08 | 12.31 | 9.29 | 5.08 | 10.46 | 13.08 | 17.57 | **25.29** |
| 49 | GestureMidAirD3 | 10.92 | 10.80 | 7.65 | 9.02 | 6.92 | 9.38 | **13.42** | 12.12 |
| 50 | GesturePebbleZ1 | 41.67 | 37.16 | 49.58 | 23.02 | 35.37 | 41.30 | **57.20** | 47.81 |
| 51 | GesturePebbleZ2 | 48.25 | 49.97 | 50.61 | 19.87 | 41.57 | 51.04 | 44.49 | **55.39** |
| 52 | GunPoint | 37.68 | 35.44 | 46.00 | 49.87 | 47.87 | 36.27 | 47.60 | **52.80** |
| 53 | GunPointAgeSpan | 50.09 | 50.30 | 48.34 | 49.05 | 50.13 | 50.00 | 46.30 | **55.44** |
| 54 | GunPointMaleVersusFemale | 46.68 | 44.89 | 55.09 | 49.90 | 49.49 | 48.61 | 51.14 | **56.75** |
| 55 | GunPointOldVersusYoung | 53.69 | 56.10 | 61.82 | 50.79 | 50.48 | 61.46 | 54.79 | **70.54** |
| 56 | Ham | 46.29 | 47.20 | **54.19** | 50.29 | 49.33 | 49.14 | 53.05 | 50.10 |
| 57 | HandOutlines | 38.01 | 35.18 | **47.62** | 46.97 | 47.19 | 35.51 | 35.25 | 47.14 |
| 58 | Haptics | 24.61 | **29.77** | 25.95 | 20.39 | 21.75 | 28.83 | 24.35 | 28.16 |
| 59 | Herring | 51.06 | 56.00 | 59.38 | 51.88 | 55.63 | 52.81 | 59.38 | **59.69** |
| 60 | HouseTwenty | 35.29 | 55.16 | 48.07 | 51.60 | 48.40 | 55.46 | **55.50** | 53.78 |
| 61 | InlineSkate | 20.43 | 19.03 | 19.79 | 15.64 | 18.36 | 19.24 | **21.40** | 18.78 |
| 62 | InsectEPGRegularTrain | 71.20 | 72.48 | 82.21 | 85.14 | **93.25** | 78.23 | 69.52 | 76.87 |
| 63 | InsectEPGSmallTrain | 58.68 | 55.71 | **75.78** | 51.97 | **75.78** | 75.50 | 64.90 | **75.78** |
| 64 | InsectWingbeatSound | 20.01 | 19.37 | 18.73 | 9.09 | 9.09 | 20.53 | 18.96 | **23.04** |
| 65 | ItalyPowerDemand | 48.54 | 47.88 | 50.46 | 50.00 | **51.74** | 48.92 | 49.49 | 49.18 |
| 66 | LargeKitchenAppliances | 57.01 | 62.35 | 64.25 | 36.75 | 40.27 | **70.88** | 57.23 | 60.23 |
| 67 | Lightning2 | 55.08 | 52.00 | 52.62 | 50.82 | 47.61 | 52.46 | 51.15 | **58.03** |
| 68 | Lightning7 | 38.41 | 38.41 | 40.19 | 13.70 | 32.33 | 41.37 | **43.67** | 41.81 |
| 69 | Mallat | 29.34 | **39.77** | 29.91 | 12.43 | 12.54 | 38.02 | 26.15 | 35.48 |
| 70 | Meat | 65.80 | **66.07** | 45.33 | 36.00 | 33.33 | 56.00 | 42.87 | 45.67 |
| 71 | MedicalImages | 53.02 | 52.82 | 57.34 | 42.74 | 51.45 | 56.39 | 54.61 | **58.01** |
| 72 | MelbournePedestrian | 84.49 | 84.78 | **87.37** | 48.33 | 29.05 | 86.96 | 86.85 | 87.17 |
| 73 | MiddlePhalanxOutlineAgeGroup | 43.79 | 40.70 | 35.79 | 34.16 | 37.92 | 42.60 | **54.29** | 46.42 |
| 74 | MiddlePhalanxOutlineCorrect | 46.21 | 47.64 | 46.63 | 42.96 | 45.77 | 47.63 | 45.99 | **48.59** |
| 75 | MiddlePhalanxTW | 48.73 | 48.60 | 51.36 | 23.38 | **55.84** | 52.34 | 54.25 | 53.38 |
| 76 | MixedShapesRegularTrain | 80.82 | 84.57 | 84.04 | 20.86 | 46.24 | **86.16** | 78.97 | 69.96 |
| 77 | MixedShapesSmallTrain | 58.86 | 57.25 | 41.97 | 21.19 | 23.96 | 57.84 | **59.24** | 46.12 |
| 78 | MoteStrain | 43.45 | 41.48 | **50.14** | 47.65 | 44.47 | 43.59 | 43.57 | 47.77 |
| 79 | NonInvasiveFetalECGThorax1 | 20.65 | 13.78 | 8.20 | 2.50 | 2.95 | 15.37 | 27.85 | **60.91** |
| 80 | NonInvasiveFetalECGThorax2 | 21.97 | 14.82 | 8.65 | 2.40 | 2.90 | 15.93 | 32.00 | **61.47** |
| 81 | OSULeaf | 66.60 | 57.31 | 47.21 | 19.09 | 40.17 | 56.12 | **68.47** | 48.46 |
| 82 | OliveOil | 36.53 | 35.60 | 34.67 | 38.00 | 34.67 | 35.33 | 32.67 | **40.40** |
| 83 | PLAID | 33.62 | 33.77 | 30.50 | 24.45 | 20.48 | 32.40 | **34.63** | 32.62 |
| 84 | PhalangesOutlinesCorrect | 52.63 | 53.25 | 52.19 | 50.73 | 51.86 | **56.22** | 47.93 | 49.03 |
| 85 | Phoneme | 20.13 | 18.93 | 19.15 | 17.54 | 9.44 | 20.22 | **21.72** | 19.73 |
| 86 | PickupGestureWiimoteZ | 26.72 | 25.28 | 30.80 | 10.00 | 16.80 | 29.60 | 31.76 | **33.12** |
| 87 | PigAirwayPressure | 9.52 | 7.00 | 7.69 | 3.42 | 3.85 | 7.69 | **11.90** | 9.90 |
| 88 | PigArtPressure | 9.90 | 8.27 | 9.38 | 4.52 | 2.88 | 8.17 | 17.21 | **18.42** |
| 89 | PigCVP | 7.21 | 6.00 | 7.93 | 2.67 | 5.58 | 5.96 | 8.94 | **10.33** |
| 90 | Plane | 78.13 | 81.75 | 67.57 | 14.10 | 73.14 | **82.86** | 81.52 | 68.76 |
| 91 | PowerCons | 50.56 | **51.73** | 50.06 | 48.11 | 50.31 | 50.56 | 49.17 | 37.33 |
| 92 | ProximalPhalanxOutlineAgeGroup | 65.13 | 67.98 | 69.06 | 29.07 | 71.71 | 73.56 | **83.86** | 72.70 |
| 93 | ProximalPhalanxOutlineCorrect | 53.92 | 49.73 | 59.49 | **61.03** | 46.32 | 48.73 | 45.60 | **61.03** |
| 94 | ProximalPhalanxTW | 73.01 | 73.89 | 69.22 | 22.34 | 67.80 | 75.51 | **77.76** | 72.49 |
| 95 | RefrigerationDevices | 42.30 | 41.14 | 44.64 | 37.33 | 45.55 | 43.36 | **45.62** | 43.77 |
| 96 | Rock | 29.60 | 27.20 | 25.40 | 19.20 | 28.40 | 26.80 | **31.00** | 26.40 |
| 97 | ScreenType | 47.13 | **47.84** | 43.20 | 36.11 | 39.73 | 46.45 | 43.76 | 41.29 |
| 98 | SemgHandGenderCh2 | 47.51 | 49.70 | 46.14 | **51.72** | 41.00 | 49.93 | 48.03 | 46.95 |
| 99 | SemgHandMovementCh2 | 42.62 | 38.52 | **42.91** | 35.11 | 27.16 | 40.40 | 38.93 | 39.84 |
| 100 | SemgHandSubjectCh2 | 44.76 | **49.64** | 43.21 | 24.37 | 26.00 | 48.36 | 44.08 | 46.44 |
| 101 | ShakeGestureWiimoteZ | 39.12 | 32.16 | 29.84 | 15.60 | 16.40 | 34.00 | **47.80** | 28.48 |
| 102 | ShapeletSim | 51.78 | 52.18 | 49.94 | 50.00 | 57.02 | 52.00 | 48.09 | **59.98** |
| 103 | ShapesAll | 26.78 | 19.34 | 28.77 | 6.43 | 1.67 | 20.83 | 22.48 | **43.24** |
| 104 | SmallKitchenAppliances | 55.62 | 57.22 | 63.45 | 53.89 | 50.14 | **63.47** | 57.20 | 57.65 |
| 105 | SmoothSubspace | 51.60 | 51.01 | 68.33 | 33.33 | 50.64 | 51.47 | **71.88** | 56.40 |
| 106 | SonyAIBORobotSurface1 | 56.49 | 54.22 | **71.80** | 51.41 | 52.45 | 56.84 | 54.74 | 48.42 |
| 107 | SonyAIBORobotSurface2 | 48.29 | 48.87 | 46.92 | 47.66 | 47.94 | 47.97 | 48.67 | **61.36** |
| 108 | StarLightCurves | 86.01 | 89.49 | 87.59 | 63.60 | 85.41 | **91.02** | 89.15 | 86.81 |
| 109 | Strawberry | 48.92 | 47.99 | 45.65 | **49.86** | 41.41 | 47.46 | 43.03 | 45.33 |
| 110 | SwedishLeaf | 66.72 | 66.50 | 68.52 | 6.21 | 19.40 | **69.28** | 61.14 | 67.74 |
| 111 | Symbols | 44.22 | 50.24 | 43.06 | 28.52 | **50.98** | 47.20 | 44.68 | **50.98** |
| 112 | SyntheticControl | 62.23 | 64.69 | **92.60** | 16.67 | 80.07 | 72.93 | 89.60 | 88.32 |
| 113 | ToeSegmentation1 | 41.23 | 42.93 | 46.04 | **50.53** | 47.58 | 43.07 | 41.32 | 49.47 |
| 114 | ToeSegmentation2 | 51.54 | 56.89 | 52.38 | 43.69 | 55.91 | 54.92 | **60.92** | 41.78 |
| 115 | Trace | 73.64 | **77.32** | 53.42 | 43.80 | 46.60 | 76.20 | 62.96 | 63.60 |
| 116 | TwoLeadECG | 42.84 | **54.34** | 47.87 | 49.99 | 50.48 | 52.61 | 50.59 | 44.02 |
| 117 | TwoPatterns | 46.82 | 41.01 | 79.68 | 24.85 | 63.84 | 61.87 | 78.81 | **85.17** |
| 118 | UMD | 43.64 | 45.47 | 40.89 | 35.28 | 45.28 | **47.08** | 41.04 | 44.03 |
| 119 | UWaveGestureLibraryAll | 49.63 | 53.72 | 48.36 | 12.35 | 12.62 | **54.17** | 47.41 | 48.81 |
| 120 | UWaveGestureLibraryX | 55.67 | 59.71 | 55.62 | 12.65 | 20.70 | **62.67** | 55.72 | 53.63 |
| 121 | UWaveGestureLibraryY | 49.49 | 50.82 | 50.42 | 12.53 | 40.89 | **51.35** | 49.24 | 48.21 |
| 122 | UWaveGestureLibraryZ | 54.49 | 56.63 | 55.03 | 12.47 | 12.62 | **57.65** | 55.47 | 51.20 |
| 123 | Wafer | 50.91 | 48.13 | 43.45 | **59.17** | 32.14 | 46.39 | 50.25 | 46.52 |
| 124 | Wine | 46.30 | 48.15 | **50.00** | **50.00** | **50.00** | 47.78 | **50.00** | **50.00** |
| 125 | WordSynonyms | 30.15 | 13.82 | 18.33 | 9.31 | 14.80 | 20.91 | **31.62** | 30.57 |
| 126 | Worms | 58.70 | 56.36 | 57.01 | 26.23 | 48.57 | **61.38** | 60.91 | 57.71 |
| 127 | WormsTwoClass | 53.51 | 49.09 | **61.56** | 47.27 | 47.27 | 57.14 | 57.40 | 59.48 |
| 128 | Yoga | 52.81 | 49.67 | 49.59 | **54.12** | 49.29 | 53.21 | 50.22 | 49.13 |
| | **Avg Acc** | 45.27 | 44.59 | 46.03 | 32.83 | 37.80 | 46.91 | 46.86 | **48.78** |
| | **Avg Rank** | 4.31 | 4.57 | 4.05 | 6.43 | 5.89 | 3.56 | 3.86 | **3.11** |
| | **P-value** | 3.15E-05 | 1.70E-05 | 4.02E-04 | 7.48E-19 | 1.22E-15 | 1.40E-02 | 4.93E-03 | - |

Table 13: The test classification accuracy (%) results on UCR archive with Asym 40% noisy labels.

| ID | Dataset | Standard | MixUp | Co-teaching | FINE | SREA | SELC | CULCU | Scale-teaching |
|---|---|---|---|---|---|---|---|---|---|
| 1 | ACSF1 | 39.00 | 40.56 | 41.12 | 10.00 | 26.08 | 40.22 | 41.60 | **42.00** |
| 2 | Adiac | 4.35 | 8.85 | 7.80 | 2.76 | 2.30 | 9.46 | 24.37 | **25.17** |
| 3 | AllGestureWiimoteX | 34.60 | 39.66 | 35.47 | 28.15 | 12.32 | 38.83 | 38.33 | **46.83** |
| 4 | AllGestureWiimoteY | 35.14 | 39.87 | 36.20 | 17.48 | 12.81 | 37.17 | 36.79 | **43.03** |
| 5 | AllGestureWiimoteZ | 38.86 | 38.35 | 38.76 | 11.49 | 16.75 | 36.74 | 37.92 | **42.29** |
| 6 | ArrowHead | 47.80 | 52.89 | 48.01 | 32.11 | 45.42 | **58.97** | 49.95 | 39.20 |
| 7 | BME | 39.00 | 48.03 | 42.40 | 40.53 | 47.07 | 45.60 | 41.60 | **49.33** |
| 8 | Beef | 26.67 | **35.47** | 31.93 | 25.43 | 20.67 | 31.33 | 25.67 | 23.33 |
| 9 | BeetleFly | 43.00 | 51.40 | **56.50** | 50.00 | 55.00 | 53.00 | 55.50 | 40.00 |
| 10 | BirdChicken | 53.00 | 57.00 | 51.00 | 50.00 | 50.00 | 60.02 | 50.00 | **75.00** |
| 11 | CBF | 56.20 | 70.29 | 73.80 | 33.38 | 68.84 | 71.91 | 66.21 | **74.22** |
| 12 | Car | 57.12 | **60.47** | 54.83 | 24.67 | 23.00 | 54.33 | 53.47 | 44.00 |
| 13 | Chinatown | 59.47 | 55.11 | 57.26 | 45.51 | 63.48 | 55.42 | 58.41 | **67.25** |
| 14 | ChlorineConcentration | **55.85** | 49.69 | 49.52 | 41.18 | 47.22 | 49.48 | 48.58 | 51.10 |
| 15 | CinCECGTorso | 50.72 | 46.88 | 43.29 | 24.87 | 31.17 | **52.93** | 45.78 | 32.90 |
| 16 | Coffee | 47.84 | 52.00 | **54.36** | 52.14 | 52.14 | 51.43 | 52.50 | 53.57 |
| 17 | Computers | 53.14 | 59.36 | 60.72 | 55.84 | **63.81** | 60.32 | 59.38 | 48.80 |
| 18 | CricketX | **58.64** | 36.59 | 30.97 | 7.79 | 10.77 | 36.21 | 40.05 | 35.85 |
| 19 | CricketY | **41.39** | 28.52 | 25.95 | 11.03 | 9.20 | 29.44 | 37.49 | 34.72 |
| 20 | CricketZ | 37.46 | 30.15 | 20.82 | 9.69 | 11.38 | 30.62 | **37.82** | 35.49 |
| 21 | Crop | 37.17 | 48.29 | 49.58 | 42.13 | **60.29** | 51.29 | 50.04 | 55.57 |
| 22 | DiatomSizeReduction | 50.16 | 36.08 | 42.32 | 38.45 | 34.58 | 36.54 | 40.65 | **54.50** |
| 23 | DistalPhalanxOutlineAgeGroup | 37.44 | 63.65 | 62.79 | 36.98 | 64.03 | 65.04 | **65.14** | 46.76 |
| 24 | DistalPhalanxOutlineCorrect | 59.68 | 58.04 | 65.69 | 45.00 | 48.99 | 63.04 | **66.45** | 64.39 |
| 25 | DistalPhalanxTW | 55.32 | 56.83 | 57.84 | 25.18 | 47.05 | 58.71 | 58.56 | **64.03** |
| 26 | DodgerLoopDay | 21.75 | 21.25 | 28.07 | 14.50 | 14.75 | 33.75 | 31.87 | **33.85** |
| 27 | DodgerLoopGame | 50.22 | 41.30 | 52.97 | 49.57 | 51.30 | 54.20 | 49.13 | **64.78** |
| 28 | DodgerLoopWeekend | 62.61 | 70.64 | **80.87** | 64.35 | 35.94 | 73.48 | 70.43 | 70.61 |
| 29 | ECG200 | 46.80 | 40.80 | 65.80 | 41.60 | **68.80** | 46.20 | 60.60 | 61.00 |
| 30 | ECG5000 | 60.72 | 65.88 | 87.36 | 33.26 | 84.25 | 68.84 | 87.44 | **89.18** |
| 31 | ECGFiveDays | 63.06 | 54.75 | 61.29 | 50.06 | 57.21 | 63.85 | 53.39 | **66.52** |
| 32 | EOGHorizontalSignal | 40.13 | 35.14 | 36.88 | 34.48 | 8.57 | 33.87 | 30.62 | **42.32** |
| 33 | EOGVerticalSignal | 31.88 | 23.48 | 30.46 | 23.65 | 10.72 | 26.24 | 26.51 | **34.81** |
| 34 | Earthquakes | 57.37 | 54.82 | 74.32 | 35.11 | **74.82** | 59.86 | 72.59 | **74.82** |
| 35 | ElectricDevices | 56.76 | 54.86 | 61.18 | 53.42 | 55.35 | **62.86** | 62.20 | 61.98 |
| 36 | EthanolLevel | 40.46 | 25.08 | 31.04 | 27.26 | 25.04 | 27.88 | 27.26 | **40.76** |
| 37 | FaceAll | 47.52 | 40.04 | 46.85 | 4.76 | 15.21 | 49.74 | **50.91** | 45.74 |
| 38 | FaceFour | 48.09 | 50.50 | 41.14 | 49.32 | 41.36 | 50.23 | 43.86 | **57.73** |
| 39 | FacesUCR | 50.18 | 41.29 | 44.42 | 8.71 | 20.75 | 50.51 | 50.36 | **51.02** |
| 40 | FiftyWords | 22.20 | 13.71 | 24.20 | 27.43 | 9.32 | 23.38 | 20.59 | **30.90** |
| 41 | Fish | 46.40 | **53.49** | 49.04 | 13.49 | 13.14 | 51.66 | 48.67 | 44.57 |
| 42 | FordA | 67.73 | 73.05 | 73.37 | 72.43 | 68.46 | 82.33 | 82.69 | **86.59** |
| 43 | FordB | 53.11 | 55.56 | 62.88 | 55.45 | 57.99 | **63.47** | 62.81 | 60.20 |
| 44 | FreezerRegularTrain | 65.85 | 72.20 | 60.40 | 62.23 | 62.31 | 71.78 | 61.32 | **75.44** |
| 45 | FreezerSmallTrain | 53.80 | 41.11 | 46.34 | 46.24 | 44.85 | 49.98 | 53.62 | **54.60** |
| 46 | Fungi | **36.88** | 20.43 | 23.60 | 21.54 | 11.40 | 24.30 | 22.53 | 29.03 |
| 47 | GestureMidAirD1 | 22.31 | 16.15 | 17.31 | 9.85 | 7.69 | 20.15 | 23.77 | **43.23** |
| 48 | GestureMidAirD2 | 20.46 | 14.31 | 21.15 | 4.00 | 6.92 | 17.85 | 24.26 | **28.46** |
| 49 | GestureMidAirD3 | 11.54 | 7.69 | 12.69 | 6.92 | 5.54 | 12.92 | 12.51 | **15.38** |
| 50 | GesturePebbleZ1 | 40.91 | 35.58 | 32.48 | 28.60 | 31.05 | 46.63 | 50.26 | **51.07** |
| 51 | GesturePebbleZ2 | 41.04 | 36.46 | 40.19 | 30.00 | 25.82 | 44.56 | 44.16 | **48.99** |
| 52 | GunPoint | 52.43 | 50.13 | 52.13 | 50.93 | 46.53 | **62.33** | 60.67 | 56.67 |
| 53 | GunPointAgeSpan | 50.81 | 47.15 | 54.03 | 51.65 | 49.87 | 58.13 | 55.22 | **59.71** |
| 54 | GunPointMaleVersusFemale | 61.90 | 70.76 | 70.43 | 58.42 | 47.47 | 70.06 | 71.65 | **81.65** |
| 55 | GunPointOldVersusYoung | 68.90 | 66.81 | 77.03 | **85.46** | 47.62 | 72.44 | 82.76 | 82.79 |
| 56 | Ham | 47.39 | 44.19 | 48.46 | 50.29 | 47.81 | 50.29 | 50.53 | **51.43** |
| 57 | HandOutlines | 64.49 | 61.41 | **65.95** | 59.03 | 64.05 | 63.08 | 65.40 | 58.65 |
| 58 | Haptics | 32.53 | **32.95** | 30.08 | 20.65 | 21.36 | 30.84 | 29.51 | 31.62 |
| 59 | Herring | 51.38 | 52.94 | 49.22 | 51.88 | 48.13 | **55.00** | 49.69 | 43.75 |
| 60 | HouseTwenty | 55.21 | 58.79 | 65.80 | 57.98 | 51.60 | 61.34 | 62.99 | **90.76** |
| 61 | InlineSkate | 25.35 | 23.04 | 23.24 | 17.02 | 17.13 | 21.67 | **28.08** | 25.09 |
| 62 | InsectEPGRegularTrain | 92.37 | 95.31 | 99.80 | 96.63 | 99.86 | 98.80 | 95.86 | **100.00** |
| 63 | InsectEPGSmallTrain | 67.72 | 60.59 | 75.74 | 65.86 | 76.35 | 73.01 | **79.00** | 35.74 |
| 64 | InsectWingbeatSound | 18.96 | 18.37 | 23.88 | 9.09 | 9.09 | 22.58 | 22.57 | **26.31** |
| 65 | ItalyPowerDemand | 57.46 | 46.03 | 71.85 | 51.74 | 68.64 | 63.97 | 67.80 | **86.98** |
| 66 | LargeKitchenAppliances | 48.89 | 51.36 | 61.29 | 42.45 | 49.75 | **63.68** | 55.92 | 55.52 |
| 67 | Lightning2 | 60.66 | 55.41 | 61.31 | 50.82 | 60.66 | 61.64 | 52.46 | **62.66** |
| 68 | Lightning7 | 34.25 | 38.93 | 35.59 | 23.56 | 30.41 | 40.27 | 35.32 | **41.64** |
| 69 | Mallat | 36.98 | 41.82 | 36.69 | 12.40 | 12.45 | 39.82 | 34.62 | **44.43** |
| 70 | Meat | 61.47 | **62.47** | 56.43 | 40.33 | 33.33 | 59.67 | 49.77 | 58.33 |
| 71 | MedicalImages | 50.07 | 47.26 | 53.24 | 34.37 | 51.45 | 53.03 | 47.31 | **59.66** |
| 72 | MelbournePedestrian | 62.07 | 67.72 | 69.20 | 34.30 | 23.03 | **71.67** | 68.21 | 67.41 |
| 73 | MiddlePhalanxOutlineAgeGroup | 37.40 | 36.36 | 50.73 | 28.57 | 47.92 | 51.71 | **60.78** | 17.92 |
| 74 | MiddlePhalanxOutlineCorrect | 60.78 | 53.54 | 58.49 | 54.23 | 57.04 | **63.44** | 57.56 | 60.14 |
| 75 | MiddlePhalanxTW | 46.29 | 38.83 | 44.17 | 22.86 | 50.39 | 50.13 | **51.36** | 47.40 |
| 76 | MixedShapesRegularTrain | 72.19 | 69.94 | 71.95 | 20.86 | 34.27 | **78.54** | 70.65 | 66.09 |
| 77 | MixedShapesSmallTrain | 57.22 | **58.54** | 43.23 | 19.14 | 38.38 | 54.99 | 56.72 | 40.49 |
| 78 | MoteStrain | 54.76 | **54.80** | 51.66 | 50.78 | 52.50 | 54.06 | 48.68 | 46.09 |
| 79 | NonInvasiveFetalECGThorax1 | 23.77 | 17.39 | 7.20 | 2.25 | 3.34 | 17.69 | 27.93 | **55.65** |
| 80 | NonInvasiveFetalECGThorax2 | 29.06 | 18.38 | 8.22 | 2.33 | 3.84 | 22.69 | 32.04 | **55.02** |
| 81 | OSULeaf | 59.69 | 64.18 | 51.38 | 16.86 | 27.02 | 62.31 | 53.23 | **72.89** |
| 82 | OliveOil | 37.33 | 36.00 | 36.00 | 30.67 | 35.33 | 35.33 | 35.33 | **40.00** |
| 83 | PLAID | 29.13 | 28.23 | 23.36 | 21.38 | 12.51 | 29.24 | **33.02** | 18.44 |
| 84 | PhalangesOutlinesCorrect | 61.56 | 59.63 | 59.69 | 62.43 | 56.78 | 65.22 | 64.63 | **66.43** |
| 85 | Phoneme | 21.21 | 17.78 | 18.80 | 19.76 | 6.12 | 19.51 | **22.14** | 19.63 |
| 86 | PickupGestureWiimoteZ | 35.44 | 23.20 | 22.16 | 10.00 | 11.60 | 34.40 | 35.48 | **49.20** |
| 87 | PigAirwayPressure | 12.71 | 5.87 | 7.86 | 6.51 | 4.23 | 9.71 | **18.04** | 12.98 |
| 88 | PigArtPressure | 15.31 | 9.62 | 8.64 | 7.87 | 1.92 | 10.96 | 22.12 | **30.38** |
| 89 | PigCVP | 11.63 | 4.33 | 8.17 | 6.57 | 6.15 | 9.52 | 13.01 | **14.33** |
| 90 | Plane | 70.29 | 65.87 | 61.14 | 13.52 | 69.71 | **70.81** | 69.52 | 49.52 |
| 91 | PowerCons | 60.04 | 58.60 | 77.33 | 50.00 | **80.42** | 63.33 | 74.67 | 63.56 |
| 92 | ProximalPhalanxOutlineAgeGroup | 58.65 | 59.43 | 45.37 | 38.34 | 67.22 | 65.17 | **73.59** | 71.71 |
| 93 | ProximalPhalanxOutlineCorrect | 63.31 | 64.52 | **74.13** | 60.41 | 68.38 | 70.58 | 71.90 | 72.03 |
| 94 | ProximalPhalanxTW | 60.04 | 61.83 | 56.26 | 28.59 | 50.73 | 66.34 | **70.82** | 68.39 |
| 95 | RefrigerationDevices | 36.92 | 38.93 | 42.13 | 33.33 | **44.58** | 39.73 | 41.52 | 39.47 |
| 96 | Rock | 45.80 | 40.00 | 47.40 | 40.80 | 37.60 | **48.00** | 46.40 | 38.00 |
| 97 | ScreenType | 48.79 | **49.29** | 42.51 | 34.67 | 38.24 | 47.25 | 38.57 | 34.24 |
| 98 | SemgHandGenderCh2 | 57.31 | 57.87 | 53.92 | 49.59 | 53.00 | 58.30 | 53.72 | **62.27** |
| 99 | SemgHandMovementCh2 | 40.74 | 38.34 | 40.29 | 29.73 | 23.94 | 37.91 | 39.29 | **40.89** |
| 100 | SemgHandSubjectCh2 | 42.55 | **45.12** | 42.98 | 26.62 | 21.76 | 41.69 | 41.03 | 41.07 |
| 101 | ShakeGestureWiimoteZ | 39.92 | 42.72 | 31.08 | 15.84 | 24.80 | 40.00 | 40.40 | **52.00** |
| 102 | ShapeletSim | 53.22 | 50.98 | 50.89 | 50.00 | **53.36** | 52.78 | 51.80 | 50.00 |
| 103 | ShapesAll | 30.23 | 19.95 | 25.54 | 2.13 | 1.67 | 20.60 | 24.84 | **49.00** |
| 104 | SmallKitchenAppliances | 58.25 | 61.06 | 67.06 | 47.93 | 60.85 | 64.48 | 68.20 | **69.01** |
| 105 | SmoothSubspace | 61.87 | 60.29 | 72.67 | 33.33 | 59.17 | 61.07 | 71.89 | **73.87** |
| 106 | SonyAIBORobotSurface1 | 69.88 | 70.49 | 70.67 | 51.41 | 75.13 | 70.18 | 69.55 | **78.37** |
| 107 | SonyAIBORobotSurface2 | 60.08 | 60.90 | **63.00** | 57.02 | 57.31 | 59.62 | 62.61 | 46.90 |
| 108 | StarLightCurves | 75.55 | 76.10 | 85.10 | 45.38 | 56.47 | 80.35 | 82.89 | **95.26** |
| 109 | Strawberry | 71.12 | 74.83 | 74.65 | 41.41 | 64.32 | **76.03** | 69.16 | 66.92 |
| 110 | SwedishLeaf | 67.77 | 64.08 | 68.43 | 6.82 | 21.98 | 69.94 | 63.31 | **70.29** |
| 111 | Symbols | 55.45 | 48.60 | 49.18 | 52.31 | **56.72** | 50.05 | 50.50 | 45.29 |
| 112 | SyntheticControl | 67.81 | 68.20 | 78.87 | 16.67 | 72.91 | 70.73 | **78.95** | 75.13 |
| 113 | ToeSegmentation1 | 51.96 | 52.39 | **62.72** | 49.47 | 59.42 | 53.07 | 47.96 | 41.67 |
| 114 | ToeSegmentation2 | 54.15 | 53.69 | 64.69 | 56.31 | 66.89 | 54.62 | 65.92 | **78.46** |
| 115 | Trace | 51.52 | 53.36 | 46.32 | 29.20 | 50.00 | 59.60 | **60.34** | 42.40 |
| 116 | TwoLeadECG | 72.26 | 70.47 | 71.84 | 49.97 | 74.96 | 72.96 | 71.98 | **86.11** |
| 117 | TwoPatterns | 63.84 | 46.01 | **78.65** | 24.54 | 67.94 | 68.04 | 77.95 | 78.33 |
| 118 | UMD | 60.44 | 63.39 | 47.43 | 40.56 | 49.44 | **64.28** | 54.24 | 60.42 |
| 119 | UWaveGestureLibraryAll | 50.56 | 52.06 | 47.51 | 12.46 | 12.28 | **53.14** | 47.51 | 52.60 |
| 120 | UWaveGestureLibraryX | 50.72 | 52.96 | 52.39 | 12.48 | 16.35 | 52.88 | 52.41 | **56.58** |
| 121 | UWaveGestureLibraryY | 47.23 | 49.15 | 47.58 | 12.59 | 30.97 | **50.76** | 47.93 | 48.40 |
| 122 | UWaveGestureLibraryZ | 52.53 | **52.82** | 51.28 | 12.56 | 28.83 | 52.70 | 51.08 | 52.29 |
| 123 | Wafer | 79.93 | 84.06 | 88.69 | 70.43 | 89.21 | 90.19 | 88.26 | **90.37** |
| 124 | Wine | 51.41 | 45.19 | **52.19** | 50.00 | 50.00 | 51.48 | 50.44 | 50.00 |
| 125 | WordSynonyms | 29.20 | 26.53 | 19.82 | 14.58 | 21.36 | 25.30 | **29.80** | 27.59 |
| 126 | Worms | 52.31 | 55.01 | 53.32 | 25.19 | 40.52 | 56.48 | 49.64 | **58.96** |
| 127 | WormsTwoClass | 56.83 | 50.91 | 58.05 | 56.44 | 49.09 | 58.70 | **59.04** | 51.17 |
| 128 | Yoga | 55.17 | 55.77 | 55.66 | 47.86 | 50.71 | **57.82** | 50.58 | 51.89 |
| | **Avg Acc** | 48.60 | 47.31 | 49.36 | 34.11 | 40.37 | 50.75 | 50.98 | **52.87** |
| | **Avg Rank** | 4.38 | 4.80 | 3.93 | 6.91 | 5.91 | 3.30 | 3.67 | **2.95** |
| | **P-value** | 1.62E-05 | 3.53E-07 | 6.10E-04 | 1.93E-23 | 9.82E-14 | 1.89E-02 | 2.24E-02 | - |

Table 14: The test classification accuracy (%) results on UCR archive with Ins 40% noisy labels.

| ID | Dataset | Standard | MixUp | Co-teaching | FINE | SREA | SELC | CULCU | Scale-teaching |
|----|---------|----------|-------|-------------|------|------|------|-------|----------------|
| 1 | ACSF1 | 35.40 | 39.08 | 43.00 | 14.80 | 29.20 | 41.40 | 42.70 | **45.00** |
| 2 | Adiac | 8.33 | 9.73 | 7.29 | 4.57 | 2.35 | 9.77 | 19.93 | **27.16** |
| 3 | AllGestureWiimoteX | 36.80 | 38.15 | 32.91 | 24.82 | 10.00 | 36.92 | 32.78 | **43.78** |
| 4 | AllGestureWiimoteY | 39.82 | 39.10 | 37.02 | 23.39 | 10.03 | 38.80 | 37.83 | **46.49** |
| 5 | AllGestureWiimoteZ | 31.95 | 31.15 | 33.87 | 15.43 | 11.51 | 33.91 | 32.04 | **38.78** |
| 6 | ArrowHead | 42.27 | 45.97 | **48.32** | 32.11 | 38.51 | 41.26 | 34.06 | 42.03 |
| 7 | BME | 36.93 | 40.81 | 41.49 | 40.27 | 36.13 | 41.73 | 38.13 | **42.21** |
| 8 | Beef | 31.33 | 33.00 | 35.67 | 20.00 | 22.67 | **36.33** | 33.93 | 29.33 |
| 9 | BeetleFly | 55.40 | 55.00 | 60.50 | 50.00 | 50.00 | 59.00 | 68.00 | **70.00** |
| 10 | BirdChicken | 64.00 | 70.80 | 55.00 | 50.00 | 50.00 | **71.00** | 67.00 | 68.00 |
| 11 | CBF | 61.00 | 59.97 | 58.34 | 33.20 | 64.45 | **65.31** | 58.66 | 51.31 |
| 12 | Car | 49.80 | 49.80 | 50.50 | 24.67 | 26.33 | **50.67** | 40.03 | 45.60 |
| 13 | Chinatown | 62.96 | 62.11 | 58.87 | 54.49 | 45.51 | 62.94 | 59.10 | **64.53** |
| 14 | ChlorineConcentration | 41.58 | 43.47 | 43.05 | 35.27 | 35.96 | 44.63 | 43.55 | **45.55** |
| 15 | CinCECGTorso | 37.14 | 49.42 | 41.05 | 35.70 | 30.52 | **50.39** | 45.08 | 46.80 |
| 16 | Coffee | 61.43 | 66.57 | 62.86 | 49.29 | 53.57 | 66.43 | 65.71 | **75.00** |
| 17 | Computers | 41.92 | 54.16 | 51.12 | 47.28 | 51.60 | 54.00 | 54.28 | **54.48** |
| 18 | CricketX | 33.79 | 35.04 | 23.45 | 7.95 | 10.62 | 38.26 | 44.06 | **46.39** |
| 19 | CricketY | 44.24 | 36.55 | 31.28 | 8.77 | 9.33 | 35.38 | 42.40 | **47.48** |
| 20 | CricketZ | 41.57 | 32.33 | 28.08 | 8.92 | 11.33 | 33.38 | 44.99 | **45.27** |
| 21 | Crop | 52.52 | 53.74 | 56.48 | 44.00 | **63.37** | 54.45 | 56.94 | 63.25 |
| 22 | DiatomSizeReduction | 63.35 | 71.07 | 55.82 | 30.00 | 30.20 | **72.12** | 45.77 | 67.36 |
| 23 | DistalPhalanxOutlineAgeGroup | 47.25 | 49.29 | 59.12 | 45.04 | **62.30** | 49.35 | 57.99 | 51.83 |
| 24 | DistalPhalanxOutlineCorrect | 53.12 | 56.22 | **66.17** | 51.67 | 55.87 | 64.42 | 59.57 | 63.12 |
| 25 | DistalPhalanxTW | 50.76 | 52.23 | 56.63 | 25.47 | 49.35 | 52.23 | 57.77 | **59.42** |
| 26 | DodgerLoopDay | 26.20 | 29.05 | 30.07 | 13.75 | 15.00 | 30.25 | 26.00 | **31.25** |
| 27 | DodgerLoopGame | 55.22 | 56.87 | 53.64 | 49.57 | 52.17 | 56.23 | 52.01 | **56.93** |
| 28 | DodgerLoopWeekend | 58.84 | 56.29 | 68.84 | 45.22 | 54.78 | 58.99 | **69.90** | 57.19 |
| 29 | ECG200 | 61.40 | 63.20 | 61.40 | 52.80 | 64.52 | 61.40 | 67.30 | **67.40** |
| 30 | ECG5000 | 73.57 | 71.83 | 89.10 | 49.16 | 90.76 | 82.03 | 90.46 | **91.12** |
| 31 | ECGFiveDays | 54.18 | 52.99 | 56.27 | 50.06 | 51.91 | 55.08 | 50.88 | **59.77** |
| 32 | EOGHorizontalSignal | 37.19 | 29.45 | 35.52 | 31.96 | 10.70 | 28.73 | 28.86 | **39.51** |
| 33 | EOGVerticalSignal | 26.15 | 26.60 | 25.62 | 23.43 | 9.83 | 23.59 | 21.59 | **27.16** |
| 34 | Earthquakes | 60.12 | 58.76 | 71.24 | **74.82** | **74.82** | 60.29 | 72.50 | 72.09 |
| 35 | ElectricDevices | 68.39 | **69.51** | 68.90 | 67.86 | 60.71 | 68.57 | 68.82 | 64.59 |
| 36 | EthanolLevel | 25.15 | 31.18 | **32.74** | 24.88 | 25.12 | 27.76 | 32.14 | 32.40 |
| 37 | FaceAll | 56.89 | 57.38 | 52.60 | 7.87 | 17.49 | **59.91** | 53.41 | 57.25 |
| 38 | FaceFour | 57.36 | **60.00** | 38.07 | 21.36 | 31.50 | 58.41 | 38.75 | 47.68 |
| 39 | FacesUCR | 48.81 | 42.41 | 48.23 | 7.53 | 14.25 | **50.26** | 46.51 | 48.29 |
| 40 | FiftyWords | 32.49 | 27.16 | 35.93 | 28.56 | 10.73 | 29.80 | 29.78 | **40.44** |
| 41 | Fish | 37.99 | 40.66 | 37.71 | 19.43 | 13.94 | 37.49 | 37.70 | **40.94** |
| 42 | FordA | 68.92 | 71.77 | 80.55 | 59.04 | 69.17 | 82.76 | **83.39** | 81.52 |
| 43 | FordB | 60.70 | 62.61 | **70.78** | 61.02 | 57.93 | 68.99 | 67.27 | 66.54 |
| 44 | FreezerRegularTrain | 71.31 | **74.21** | 54.41 | 59.09 | 52.78 | 71.61 | 61.98 | 62.43 |
| 45 | FreezerSmallTrain | 55.52 | 55.64 | 57.34 | 44.18 | 44.88 | 52.60 | **58.16** | 52.53 |
| 46 | Fungi | 23.28 | 14.41 | **24.48** | 20.51 | 11.08 | 14.52 | 19.78 | 18.09 |
| 47 | GestureMidAirD1 | 28.15 | 16.92 | 19.23 | 19.14 | 12.12 | 23.54 | 25.17 | **39.17** |
| 48 | GestureMidAirD2 | 13.51 | 6.46 | 13.46 | 4.92 | 7.78 | 11.85 | 15.82 | **22.58** |
| 49 | GestureMidAirD3 | 12.77 | 9.08 | 11.15 | 11.23 | 5.08 | 11.85 | 12.58 | **13.91** |
| 50 | GesturePebbleZ1 | 50.65 | 44.05 | 47.09 | 26.77 | 38.63 | 44.77 | 51.41 | **52.12** |
| 51 | GesturePebbleZ2 | 56.73 | 50.56 | 53.35 | 36.20 | 33.80 | 56.48 | 54.16 | **59.27** |
| 52 | GunPoint | 60.05 | **66.69** | 55.13 | 48.40 | 47.73 | 60.00 | 56.40 | 52.40 |
| 53 | GunPointAgeSpan | 59.49 | **60.59** | 57.06 | 51.58 | 49.62 | 60.57 | 57.51 | 53.11 |
| 54 | GunPointMaleVersusFemale | 66.18 | 63.11 | 67.59 | 49.87 | 47.47 | 65.63 | 71.22 | **76.27** |
| 55 | GunPointOldVersusYoung | 87.24 | 90.18 | 93.52 | 78.29 | 48.57 | 90.79 | 96.10 | **99.56** |
| 56 | Ham | 59.81 | 57.45 | 56.23 | 48.57 | 49.71 | **59.90** | 56.57 | 51.24 |
| 57 | HandOutlines | 65.57 | **66.36** | 65.38 | 64.32 | 64.05 | 66.16 | 66.35 | 64.28 |
| 58 | Haptics | 27.40 | **28.70** | 26.19 | 19.81 | 20.06 | 28.53 | 26.92 | 27.97 |
| 59 | Herring | 53.50 | 55.88 | 54.06 | 51.88 | 51.88 | 56.56 | 52.53 | **58.56** |
| 60 | HouseTwenty | 70.39 | 57.14 | 63.24 | 48.40 | 54.79 | **71.26** | 48.35 | 69.41 |
| 61 | InlineSkate | 19.67 | 15.75 | 19.49 | 16.00 | 14.18 | 18.62 | 20.16 | **21.11** |
| 62 | InsectEPGRegularTrain | 69.43 | 71.18 | 83.90 | 85.38 | **87.15** | 72.45 | 71.53 | 65.06 |
| 63 | InsectEPGSmallTrain | 67.34 | 71.52 | 89.88 | 56.63 | 72.57 | 74.70 | **93.10** | 86.84 |
| 64 | InsectWingbeatSound | 20.47 | 14.26 | 22.87 | 9.09 | 9.32 | 18.90 | 19.40 | **25.99** |
| 65 | ItalyPowerDemand | 67.35 | 65.55 | 61.10 | 49.91 | 69.69 | 69.23 | **84.53** | 63.76 |
| 66 | LargeKitchenAppliances | 72.12 | 69.65 | 77.86 | 46.08 | 46.59 | **78.36** | 72.84 | 73.00 |
| 67 | Lightning2 | 56.72 | 54.10 | 56.89 | 54.43 | 55.74 | **60.98** | 58.69 | 55.08 |
| 68 | Lightning7 | 36.15 | 35.62 | 33.15 | 19.18 | 26.03 | 35.62 | 38.77 | **39.12** |
| 69 | Mallat | 35.91 | 26.42 | 33.05 | 15.43 | 12.47 | 33.39 | 31.37 | **40.87** |
| 70 | Meat | 46.60 | **49.73** | 45.63 | 33.33 | 33.33 | 47.67 | 43.17 | 46.20 |
| 71 | MedicalImages | 52.91 | 44.42 | 53.83 | 51.45 | 49.08 | 52.29 | 50.81 | **55.41** |
| 72 | MelbournePedestrian | 68.53 | 62.94 | 69.50 | 39.53 | 22.90 | 71.26 | 70.95 | **71.68** |
| 73 | MiddlePhalanxOutlineAgeGroup | 46.52 | 36.88 | 51.09 | 27.53 | **61.69** | 49.09 | 44.71 | 47.53 |
| 74 | MiddlePhalanxOutlineCorrect | 59.27 | 52.51 | 55.14 | 48.59 | 51.41 | **60.10** | 53.33 | 51.07 |
| 75 | MiddlePhalanxTW | 43.71 | 39.64 | 40.52 | 21.30 | 34.16 | 45.97 | **50.82** | 45.06 |
| 76 | MixedShapesRegularTrain | 79.62 | 78.52 | 71.27 | 22.80 | 38.91 | **83.27** | 71.35 | 67.12 |
| 77 | MixedShapesSmallTrain | 59.86 | 51.69 | 49.41 | 21.01 | 21.18 | 62.99 | **63.29** | 51.54 |
| 78 | MoteStrain | 73.52 | 73.10 | 70.78 | 49.22 | 72.73 | 72.83 | **76.43** | 67.14 |
| 79 | NonInvasiveFetalECGThorax1 | 16.24 | 8.66 | 9.92 | 2.39 | 2.92 | 12.31 | 18.02 | **44.45** |
| 80 | NonInvasiveFetalECGThorax2 | 20.81 | 8.60 | 11.22 | 2.15 | 3.21 | 13.57 | 18.03 | **41.88** |
| 81 | OSULeaf | 63.10 | 62.63 | 59.86 | 16.03 | 27.01 | 60.25 | **65.83** | 61.97 |
| 82 | OliveOil | 44.13 | 40.00 | 40.00 | 40.00 | 40.00 | 38.00 | 38.00 | **48.67** |
| 83 | PLAID | 29.83 | 25.33 | 27.60 | 15.12 | 13.61 | 29.24 | **34.59** | 28.27 |
| 84 | PhalangesOutlinesCorrect | 61.60 | 62.49 | 59.80 | 63.26 | 57.18 | 63.62 | 59.61 | **64.61** |
| 85 | Phoneme | 21.55 | 16.32 | 19.59 | 17.21 | 7.01 | 19.76 | **22.79** | 20.79 |
| 86 | PickupGestureWiimoteZ | 31.68 | 19.20 | 27.16 | 10.53 | 11.20 | 31.20 | 30.08 | **33.92** |
| 87 | PigAirwayPressure | 11.88 | 7.46 | 7.88 | 8.21 | 3.08 | 6.92 | 11.13 | **12.04** |
| 88 | PigArtPressure | 15.13 | 6.15 | 11.54 | 6.98 | 2.31 | 9.13 | 17.62 | **21.46** |
| 89 | PigCVP | 9.79 | 4.33 | 10.14 | 9.14 | 5.62 | 7.69 | 10.26 | **13.73** |
| 90 | Plane | 62.48 | 62.48 | 64.48 | 16.19 | 55.62 | 63.81 | **65.33** | 60.95 |
| 91 | PowerCons | 63.16 | 59.78 | 82.72 | 57.78 | 83.13 | 66.44 | 78.78 | **83.36** |
| 92 | ProximalPhalanxOutlineAgeGroup | 56.55 | 56.20 | 65.85 | 40.68 | **71.61** | 63.41 | 70.54 | 52.37 |
| 93 | ProximalPhalanxOutlineCorrect | 63.20 | 57.94 | 70.27 | 38.97 | 67.29 | 71.75 | 71.62 | **72.01** |
| 94 | ProximalPhalanxTW | 65.05 | 65.37 | 65.71 | 28.39 | 59.90 | 71.41 | **78.99** | 67.61 |
| 95 | RefrigerationDevices | 45.06 | 44.21 | 48.03 | 33.33 | **48.37** | 46.61 | 45.90 | 48.17 |
| 96 | Rock | 27.60 | 22.00 | 30.20 | 30.45 | 28.40 | 25.60 | **31.40** | 24.00 |
| 97 | ScreenType | 52.44 | **52.54** | 46.09 | 33.33 | 36.76 | 51.57 | 41.09 | 44.11 |
| 98 | SemgHandGenderCh2 | 56.55 | 57.53 | 56.38 | 51.73 | 53.00 | **58.67** | 53.10 | 53.46 |
| 99 | SemgHandMovementCh2 | 37.29 | 38.23 | 39.48 | 31.37 | 19.76 | 39.29 | 38.61 | **41.26** |
| 100 | SemgHandSubjectCh2 | 45.22 | **46.83** | 42.37 | 32.09 | 21.56 | 44.58 | 39.74 | 45.29 |
| 101 | ShakeGestureWiimoteZ | 40.72 | 36.32 | 34.76 | 29.87 | 26.40 | 46.80 | **47.20** | 35.20 |
| 102 | ShapeletSim | 50.64 | 48.11 | 45.98 | 50.00 | 50.67 | 49.89 | 49.83 | **54.40** |
| 103 | ShapesAll | 21.13 | 8.83 | 23.58 | 5.17 | 1.67 | 17.93 | 21.32 | **46.39** |
| 104 | SmallKitchenAppliances | 64.45 | 63.04 | **75.88** | 42.99 | 68.84 | 71.41 | 71.08 | 71.25 |
| 105 | SmoothSubspace | 59.87 | 59.84 | 66.68 | 33.33 | 59.33 | 59.07 | **67.87** | 55.17 |
| 106 | SonyAIBORobotSurface1 | 52.88 | 41.55 | 47.93 | 48.59 | 58.26 | 51.81 | **61.59** | 51.20 |
| 107 | SonyAIBORobotSurface2 | 55.56 | 50.47 | 56.68 | 52.07 | 58.35 | 55.91 | 60.81 | **61.90** |
| 108 | StarLightCurves | 75.50 | 83.51 | 88.16 | 62.79 | 76.77 | 86.72 | 89.08 | **89.53** |
| 109 | Strawberry | 66.81 | 74.49 | 73.24 | 58.71 | 52.86 | **76.27** | 60.43 | 56.30 |
| 110 | SwedishLeaf | 63.31 | 52.42 | 62.11 | 6.50 | 19.81 | 65.02 | 56.70 | **65.67** |
| 111 | Symbols | 43.34 | **47.40** | 41.84 | 41.31 | 42.72 | 46.63 | 39.23 | 44.29 |
| 112 | SyntheticControl | 70.73 | 73.53 | 86.83 | 16.67 | 82.20 | 78.73 | **88.07** | 82.87 |
| 113 | ToeSegmentation1 | 66.02 | 64.84 | 71.32 | 52.63 | **71.75** | 64.82 | 67.96 | 65.09 |
| 114 | ToeSegmentation2 | 69.38 | 72.49 | 71.54 | 43.69 | **73.88** | 71.38 | 68.15 | 69.35 |
| 115 | Trace | 63.48 | **73.64** | 51.00 | 20.00 | 46.80 | 69.80 | 65.16 | 52.16 |
| 116 | TwoLeadECG | 61.86 | 57.53 | 50.69 | 49.99 | **62.05** | 59.74 | 53.03 | 51.71 |
| 117 | TwoPatterns | 58.48 | 49.17 | 84.69 | 24.87 | 85.27 | 73.59 | 83.81 | **87.37** |
| 118 | UMD | 45.67 | 57.58 | 42.14 | 50.69 | 40.00 | **58.06** | 53.75 | 47.50 |
| 119 | UWaveGestureLibraryAll | 52.41 | 49.83 | 50.04 | 12.42 | 12.38 | 49.94 | 45.61 | **53.34** |
| 120 | UWaveGestureLibraryX | 57.60 | 55.68 | 57.21 | 12.49 | 14.75 | **58.80** | 57.05 | 56.88 |
| 121 | UWaveGestureLibraryY | 46.65 | 45.77 | 43.90 | 12.39 | 32.02 | 45.31 | 43.71 | **46.92** |
| 122 | UWaveGestureLibraryZ | 49.46 | 49.37 | 48.47 | 12.56 | 21.39 | 50.09 | 47.72 | **50.23** |
| 123 | Wafer | 81.09 | 85.23 | 90.20 | 38.77 | 89.21 | 90.08 | 88.03 | **91.03** |
| 124 | Wine | 51.63 | 50.89 | 50.00 | 50.00 | 50.00 | **51.85** | 48.89 | 50.22 |
| 125 | WordSynonyms | 34.70 | 30.36 | 31.46 | 18.56 | 22.19 | 30.66 | 35.56 | **36.46** |
| 126 | Worms | 61.36 | 63.58 | 62.26 | 27.27 | 48.83 | **64.68** | 58.21 | 59.01 |
| 127 | WormsTwoClass | 53.38 | 55.16 | 54.94 | 48.57 | 50.91 | **59.48** | 53.90 | 50.39 |
| 128 | Yoga | 54.77 | 55.85 | 54.87 | 53.57 | 50.71 | **56.03** | 53.09 | 54.14 |
| | **Avg Acc** | 49.57 | 48.39 | 50.11 | 34.24 | 39.80 | 51.21 | 51.12 | **52.57** |
| | **Avg Rank** | 4.05 | 4.52 | 4.02 | 7.04 | 6.18 | 3.30 | 3.77 | **2.95** |
| | **P-value** | 1.43E-05 | 1.81E-06 | 2.43E-04 | 9.81E-26 | 2.36E-17 | 3.27E-02 | 1.54E-02 | - |

Table 15: The detailed test classification accuracy (%) results on UEA 30 archive with Sym 20% noisy labels.

| ID | Dataset | Standard | MixUp | Co-teaching | FINE | SREA | SELC | CULCU | Scale-teaching |
|---|---|---|---|---|---|---|---|---|---|
| 1 | ArticularyWordRecognition | 89.33 | 81.04 | 81.77 | 41.07 | 90.13 | 87.87 | 84.33 | **92.73** |
| 2 | AtrialFibrillation | 26.67 | 25.67 | 26.67 | **33.33** | 26.67 | 26.67 | 22.67 | 28.00 |
| 3 | BasicMotions | 97.10 | 97.40 | **99.25** | 91.00 | 97.50 | 98.00 | 97.50 | 95.50 |
| 4 | CharacterTrajectories | 98.16 | 98.49 | 92.59 | 7.45 | 96.54 | **98.93** | 98.66 | **98.93** |
| 5 | Cricket | 94.72 | 94.17 | 91.53 | 41.11 | **97.50** | 96.67 | 93.89 | 97.28 |
| 6 | DuckDuckGeese | 48.40 | **52.48** | 48.60 | 24.40 | 46.80 | 50.00 | 49.60 | 45.60 |
| 7 | EigenWorms | 53.56 | 55.42 | 62.88 | 34.66 | 61.80 | **64.58** | 64.10 | 61.25 |
| 8 | Epilepsy | 84.96 | 80.28 | **93.33** | 82.46 | 88.70 | 86.38 | **93.33** | 87.59 |
| 9 | EthanolConcentration | 24.15 | 25.48 | 26.43 | 24.71 | 25.11 | 24.87 | 24.39 | **27.42** |
| 10 | ERing | 73.11 | **73.70** | 70.04 | 16.67 | 64.59 | 72.59 | 61.70 | 72.77 |
| 11 | FaceDetection | 52.90 | 51.93 | 52.36 | 51.96 | 50.93 | 52.30 | 52.69 | **53.09** |
| 12 | FingerMovements | 50.80 | 50.44 | 52.76 | 52.20 | 50.20 | 52.80 | 52.30 | **54.68** |
| 13 | HandMovementDirection | 28.92 | 32.22 | 35.14 | 24.05 | 19.46 | 28.11 | 29.16 | **35.35** |
| 14 | Handwriting | 35.71 | 28.55 | 30.71 | 25.31 | 16.63 | 28.19 | 34.12 | **41.35** |
| 15 | Heartbeat | 52.00 | 52.93 | 57.46 | **72.10** | 62.75 | 50.17 | 66.39 | 51.76 |
| 16 | InsectWingbeat | 62.31 | 53.90 | 64.74 | 63.92 | 51.56 | 64.63 | **64.75** | 63.85 |
| 17 | JapaneseVowels | 88.99 | 87.61 | 94.70 | 49.73 | **97.41** | 93.51 | 97.19 | 95.62 |
| 18 | Libras | 74.60 | 70.07 | 77.00 | 10.78 | 73.22 | 76.78 | 70.83 | **79.67** |
| 19 | LSST | 49.68 | 48.91 | 50.23 | 51.61 | 35.53 | 50.48 | 51.91 | **55.16** |
| 20 | MotorImagery | 51.00 | 50.88 | 51.60 | 50.80 | **55.92** | 52.80 | 50.10 | 50.84 |
| 21 | NATOPS | 80.78 | 80.40 | 90.56 | 25.78 | 89.33 | 82.67 | 87.17 | **91.11** |
| 22 | PenDigits | 96.04 | 97.58 | 97.33 | **98.30** | 97.90 | 98.27 | 98.16 | 98.12 |
| 23 | PEMS-SF | 62.08 | 62.87 | 60.06 | 14.91 | **64.97** | 62.08 | 61.10 | 63.05 |
| 24 | PhonemeSpectra | 21.89 | 23.46 | 22.25 | 18.00 | 6.23 | 22.82 | 23.74 | **26.50** |
| 25 | RacketSports | 76.84 | 73.50 | 78.64 | 25.00 | 75.74 | 77.24 | **79.41** | 76.87 |
| 26 | SelfRegulationSCP1 | 76.81 | 77.49 | 78.91 | 56.86 | 49.83 | 78.43 | 77.55 | **81.28** |
| 27 | SelfRegulationSCP2 | 47.47 | 49.24 | 48.32 | 50.89 | 50.00 | 47.44 | 50.66 | **51.78** |
| 28 | SpokenArabicDigits | 96.17 | 96.21 | 98.83 | 27.79 | 98.52 | 98.67 | 98.71 | **99.18** |
| 29 | StandWalkJump | 44.00 | 40.00 | 36.00 | 33.33 | **45.33** | 42.67 | 33.40 | 34.13 |
| 30 | UWaveGestureLibrary | 76.90 | 76.97 | 74.61 | 12.50 | 74.29 | 77.75 | 66.97 | **78.12** |
| | **Avg Acc** | 63.87 | 62.98 | 64.84 | 40.42 | 62.04 | 64.81 | 64.55 | **66.29** |
| | **Avg Rank** | 5.03 | 5.20 | 3.83 | 6.37 | 4.77 | 3.73 | 4.00 | **2.73** |
| | **P-value** | 6.61E-04 | 3.33E-04 | 2.69E-02 | 2.37E-05 | 1.14E-02 | 2.63E-02 | 3.93E-02 | - |

Table 16: The detailed test classification accuracy (%) results on UEA 30 archive with Sym 50% noisy labels.

| ID | Dataset | Standard | MixUp | Co-teaching | FINE | SREA | SELC | CULCU | Scale-teaching |
|---|---|---|---|---|---|---|---|---|---|
| 1 | ArticularyWordRecognition | 66.64 | 55.57 | 52.12 | 7.53 | 67.33 | 57.40 | 49.63 | **68.68** |
| 2 | AtrialFibrillation | 28.00 | 26.67 | 29.33 | 33.33 | 29.33 | 30.67 | 32.00 | **36.00** |
| 3 | BasicMotions | 56.00 | 58.20 | 70.75 | 54.50 | **81.00** | 59.00 | 71.25 | 60.20 |
| 4 | CharacterTrajectories | 90.62 | 87.56 | 67.75 | 6.73 | 95.28 | 93.82 | 97.03 | **97.23** |
| 5 | Cricket | 67.44 | 60.00 | 68.89 | 8.33 | 84.28 | 66.67 | 68.06 | **87.27** |
| 6 | DuckDuckGeese | 32.00 | **36.64** | 29.60 | 26.00 | 34.00 | 33.20 | 33.20 | 33.84 |
| 7 | EigenWorms | 49.62 | 45.37 | 55.65 | 32.98 | 38.78 | **56.79** | 56.09 | 49.28 |
| 8 | Epilepsy | 57.39 | 57.25 | 64.16 | 36.38 | 62.62 | 60.00 | 63.55 | **64.36** |
| 9 | EthanolConcentration | 25.20 | 25.32 | 26.05 | 25.02 | **27.56** | 25.62 | 24.58 | 26.27 |
| 10 | ERing | 44.30 | 44.39 | 40.63 | 16.67 | 39.35 | 44.52 | 38.54 | **45.10** |
| 11 | FaceDetection | 48.54 | 49.81 | 49.30 | 49.51 | 48.94 | 49.11 | 48.58 | **50.72** |
| 12 | FingerMovements | 50.40 | 50.20 | 51.80 | 51.40 | 50.60 | 50.80 | 51.30 | **54.40** |
| 13 | HandMovementDirection | 26.22 | 25.51 | 25.41 | 26.11 | 23.51 | 26.57 | 24.35 | **27.08** |
| 14 | Handwriting | 19.54 | 19.53 | 19.99 | 13.41 | 7.53 | 19.18 | 18.28 | **21.06** |
| 15 | Heartbeat | 55.32 | 53.52 | 52.20 | 54.44 | **55.40** | 54.63 | 53.27 | 48.62 |
| 16 | InsectWingbeat | 49.30 | 32.34 | 58.02 | 52.25 | 31.07 | 53.97 | **58.13** | 52.01 |
| 17 | JapaneseVowels | 60.14 | 59.28 | 73.97 | 15.03 | 70.65 | 66.43 | 78.11 | **79.23** |
| 18 | Libras | 47.09 | 43.98 | **51.39** | 6.67 | 44.78 | 50.11 | 40.06 | 49.78 |
| 19 | LSST | 47.29 | 44.58 | 46.21 | 47.89 | 34.35 | 46.33 | 47.92 | **48.75** |
| 20 | MotorImagery | 50.84 | 51.48 | 50.10 | 51.40 | **52.00** | 50.60 | 49.70 | 49.80 |
| 21 | NATOPS | 54.33 | 52.53 | **60.17** | 16.67 | 59.80 | 53.44 | 58.06 | 59.56 |
| 22 | PenDigits | 93.38 | 85.29 | 95.92 | 92.85 | 92.74 | **96.80** | 96.53 | 93.89 |
| 23 | PEMS-SF | 41.20 | 40.55 | 32.96 | 14.45 | 42.43 | **43.24** | 37.57 | 41.27 |
| 24 | PhonemeSpectra | 19.08 | 19.11 | 19.94 | 11.48 | 3.92 | 19.69 | 19.23 | **20.09** |
| 25 | RacketSports | 52.50 | 51.03 | 52.80 | 24.08 | 53.29 | 52.89 | **56.58** | 54.21 |
| 26 | SelfRegulationSCP1 | 47.41 | 42.68 | 48.86 | 48.60 | 49.97 | 48.33 | 57.93 | **58.08** |
| 27 | SelfRegulationSCP2 | 48.13 | 48.22 | 47.61 | 48.78 | **50.00** | 49.22 | **50.00** | 48.73 |
| 28 | SpokenArabicDigits | 85.64 | 69.95 | 96.55 | 96.16 | **99.23** | 95.66 | 97.59 | 97.69 |
| 29 | StandWalkJump | 38.67 | 37.87 | 40.67 | 33.33 | 42.67 | 42.67 | 37.33 | **44.00** |
| 30 | UWaveGestureLibrary | 50.41 | 48.60 | 45.45 | 12.50 | **53.91** | 49.00 | 37.94 | 52.61 |
| | **Avg Acc** | 50.09 | 47.43 | 50.81 | 33.82 | 50.88 | 51.55 | 51.75 | **53.99** |
| | **Avg Rank** | 5.17 | 5.73 | 4.23 | 6.23 | 3.93 | 3.83 | 4.30 | **2.43** |
| | **P-value** | 2.98E-04 | 7.40E-05 | 1.59E-02 | 9.35E-05 | 1.67E-02 | 1.08E-02 | 3.75E-02 | - |

Table 17: The detailed test classification accuracy (%) results on UEA 30 archive with Asym 40% noisy labels.

| ID | Dataset | Standard | MixUp | Co-teaching | FINE | SREA | SELC | CULCU | Scale-teaching |
|---|---|---|---|---|---|---|---|---|---|
| 1 | ArticularyWordRecognition | 66.33 | 62.40 | 55.87 | 17.27 | 69.40 | 63.67 | 53.73 | **70.44** |
| 2 | AtrialFibrillation | 21.67 | 33.33 | 33.33 | 33.33 | 11.33 | 33.67 | 32.67 | **34.67** |
| 3 | BasicMotions | 66.00 | 62.30 | 67.25 | 49.50 | **69.00** | 64.00 | 61.75 | 65.10 |
| 4 | CharacterTrajectories | 61.08 | 60.01 | 57.42 | 19.05 | 87.78 | 64.29 | 61.35 | **88.34** |
| 5 | Cricket | 72.22 | 71.56 | 70.97 | 50.00 | 73.44 | 72.78 | 68.61 | **80.56** |
| 6 | DuckDuckGeese | 43.20 | 42.96 | 44.80 | 24.00 | 43.68 | **45.20** | 44.60 | 38.24 |
| 7 | EigenWorms | 41.75 | 34.75 | **51.34** | 37.86 | 43.56 | 41.68 | 50.38 | 42.47 |
| 8 | Epilepsy | 62.32 | 63.01 | 63.48 | 47.25 | 61.01 | 61.45 | **64.71** | 58.70 |
| 9 | EthanolConcentration | 23.04 | 23.85 | 23.61 | 25.02 | 24.78 | 24.33 | 25.57 | **27.70** |
| 10 | ERing | 60.30 | 60.37 | 42.74 | 39.47 | 45.56 | 59.11 | 43.96 | **61.74** |
| 11 | FaceDetection | 49.88 | 50.64 | 51.12 | 51.06 | 50.32 | 51.07 | 50.12 | **51.61** |
| 12 | FingerMovements | 47.76 | 49.92 | 48.50 | 49.80 | 49.00 | 48.20 | 50.19 | **50.96** |
| 13 | HandMovementDirection | 28.97 | 31.24 | 30.41 | 28.38 | **31.46** | 29.19 | 29.73 | 29.03 |
| 14 | Handwriting | 21.61 | 23.92 | 24.74 | 21.03 | 10.67 | 22.99 | 25.69 | **26.98** |
| 15 | Heartbeat | 55.22 | 57.46 | 55.61 | **72.20** | 61.52 | 55.12 | 55.51 | 56.68 |
| 16 | InsectWingbeat | 43.40 | 38.07 | 45.34 | 46.32 | 48.78 | 47.81 | 50.34 | **51.87** |
| 17 | JapaneseVowels | 61.62 | 58.63 | 62.46 | 36.81 | 65.97 | 64.27 | **73.76** | 70.02 |
| 18 | Libras | 57.47 | 57.00 | 53.39 | 8.67 | 54.33 | 59.44 | 45.72 | **63.22** |
| 19 | LSST | 42.11 | 42.77 | 41.70 | 43.67 | 32.79 | 43.67 | **43.74** | 29.10 |
| 20 | MotorImagery | 48.80 | 50.24 | 51.32 | 53.00 | 52.60 | 49.60 | 49.70 | **53.20** |
| 21 | NATOPS | 57.00 | 55.29 | 58.65 | 16.67 | 64.89 | 55.89 | 63.22 | **65.13** |
| 22 | PenDigits | 78.76 | 67.36 | 92.78 | 84.05 | 91.07 | 89.18 | 92.23 | **93.57** |
| 23 | PEMS-SF | 50.20 | 51.38 | 42.60 | 14.45 | 50.87 | 50.76 | 47.86 | **51.45** |
| 24 | PhonemeSpectra | 17.71 | 19.05 | 18.70 | 14.41 | 5.11 | 18.02 | 18.65 | **19.52** |
| 25 | RacketSports | 57.50 | **59.13** | 54.30 | 27.50 | 55.26 | 58.16 | 56.32 | 54.21 |
| 26 | SelfRegulationSCP1 | 63.47 | 66.21 | 64.94 | 60.96 | 49.83 | 66.42 | **68.10** | 66.30 |
| 27 | SelfRegulationSCP2 | 49.04 | 51.24 | 51.20 | **52.16** | 50.00 | 52.11 | 51.26 | 51.22 |
| 28 | SpokenArabicDigits | 64.19 | 60.64 | 79.13 | 72.11 | **99.04** | 79.42 | 88.10 | 93.85 |
| 29 | StandWalkJump | 38.67 | 34.67 | 39.33 | 33.33 | 33.33 | **40.00** | 39.33 | 36.27 |
| 30 | UWaveGestureLibrary | 53.56 | 53.36 | 55.38 | 12.50 | **57.81** | 53.69 | 45.53 | 55.84 |
| | **Avg Acc** | 50.16 | 49.76 | 51.08 | 38.06 | 51.47 | 52.17 | 51.75 | **54.60** |
| | **Avg Rank** | 5.60 | 4.77 | 4.40 | 6.13 | 4.20 | 4.00 | 3.97 | **2.73** |
| | **P-value** | 3.81E-03 | 6.17E-03 | 1.63E-02 | 9.33E-05 | 1.36E-02 | 2.62E-02 | 3.88E-02 | - |

Table 18: The detailed test classification accuracy (%) results on UEA 30 archive with Ins 40% noisy labels.

| ID | Dataset | Standard | MixUp | Co-teaching | FINE | SREA | SELC | CULCU | Scale-teaching |
|---|---|---|---|---|---|---|---|---|---|
| 1 | ArticularyWordRecognition | 67.27 | 57.39 | 60.93 | 9.20 | 68.67 | 61.73 | 57.10 | **75.40** |
| 2 | AtrialFibrillation | 28.00 | 29.33 | 30.00 | 33.33 | 32.00 | 28.00 | **34.67** | 32.00 |
| 3 | BasicMotions | 77.00 | 73.00 | **81.75** | 39.00 | 80.90 | 77.50 | 74.75 | 78.50 |
| 4 | CharacterTrajectories | 82.52 | 69.46 | 66.33 | 5.22 | 85.38 | 81.92 | 83.48 | **87.47** |
| 5 | Cricket | 80.28 | 78.28 | 79.31 | 26.11 | 92.50 | 81.11 | 76.81 | **92.78** |
| 6 | DuckDuckGeese | 38.80 | 41.60 | 36.40 | 20.00 | 39.20 | 39.60 | **42.00** | 37.20 |
| 7 | EigenWorms | 33.44 | 54.63 | **60.38** | 32.67 | 43.66 | 43.21 | 60.31 | 54.81 |
| 8 | Epilepsy | 65.80 | 64.55 | 72.03 | 67.10 | 73.04 | 71.74 | 78.64 | **79.36** |
| 9 | EthanolConcentration | 24.82 | 26.40 | 27.75 | 24.71 | 25.17 | 26.69 | 26.81 | **28.37** |
| 10 | ERing | 53.69 | 52.48 | 48.48 | 36.79 | 49.82 | **56.44** | 44.59 | 54.15 |
| 11 | FaceDetection | 50.05 | 50.04 | 50.80 | 50.57 | 49.81 | 50.22 | 50.93 | **51.00** |
| 12 | FingerMovements | 51.40 | **51.60** | 51.10 | 51.20 | 49.60 | 51.40 | 48.60 | 49.60 |
| 13 | HandMovementDirection | 25.31 | 30.11 | 26.49 | 19.73 | 19.73 | 28.92 | 25.95 | **31.62** |
| 14 | Handwriting | 21.35 | 22.54 | 22.92 | 17.93 | 6.17 | 21.51 | **23.19** | 23.01 |
| 15 | Heartbeat | 56.39 | 58.01 | **66.63** | 57.69 | 58.50 | 56.20 | 59.02 | 51.00 |
| 16 | InsectWingbeat | 47.94 | 39.18 | 57.07 | 55.93 | 36.95 | 57.12 | **59.40** | 58.32 |
| 17 | JapaneseVowels | 68.05 | 66.46 | 65.36 | 27.89 | 77.03 | 73.35 | 78.97 | **81.54** |
| 18 | Libras | 46.04 | **51.56** | 47.00 | 9.33 | 48.67 | 49.44 | 41.11 | 50.18 |
| 19 | LSST | 48.16 | 47.72 | 46.43 | 49.04 | 33.58 | 48.78 | 49.11 | **50.47** |
| 20 | MotorImagery | 49.20 | 51.00 | 49.90 | 50.00 | 51.00 | **52.00** | 51.90 | 47.80 |
| 21 | NATOPS | 57.78 | 56.80 | 58.78 | 26.67 | **67.24** | 57.67 | 58.34 | 59.98 |
| 22 | PenDigits | 81.99 | 70.68 | 93.43 | 91.81 | 92.59 | 91.18 | 93.29 | **96.69** |
| 23 | PEMS-SF | 42.89 | 43.86 | 35.14 | 16.42 | **47.86** | 43.82 | 41.49 | 44.35 |
| 24 | PhonemeSpectra | 16.50 | 15.82 | 17.37 | 8.40 | 3.22 | 17.15 | **17.48** | 17.42 |
| 25 | RacketSports | 59.08 | 59.55 | **64.68** | 27.50 | 60.11 | 58.55 | 58.62 | 59.89 |
| 26 | SelfRegulationSCP1 | 60.63 | 60.38 | 56.02 | 48.12 | 49.90 | 65.80 | **66.00** | 54.54 |
| 27 | SelfRegulationSCP2 | 51.69 | 50.87 | 50.63 | 51.11 | 50.00 | 51.89 | 50.22 | **52.11** |
| 28 | SpokenArabicDigits | 74.85 | 67.24 | 89.70 | 83.37 | **98.76** | 86.68 | 87.87 | 97.53 |
| 29 | StandWalkJump | 40.00 | **42.67** | 39.33 | 32.00 | **42.67** | 38.67 | 42.00 | 40.00 |
| 30 | UWaveGestureLibrary | 67.19 | 67.90 | 60.56 | 12.50 | 67.74 | 67.69 | 55.69 | **69.96** |
| | **Avg Acc** | 52.27 | 51.70 | 53.76 | 36.04 | 53.38 | 54.53 | 54.61 | **56.90** |
| | **Avg Rank** | 5.20 | 4.77 | 4.33 | 6.60 | 4.27 | 4.20 | 3.77 | **2.60** |
| | **P-value** | 6.08E-04 | 2.92E-03 | 1.20E-02 | 2.55E-05 | 5.52E-03 | 1.08E-02 | 3.47E-02 | - |

Table 19: Multi-scale analysis of Scale-teaching using UCR 128 archive with Sym 20% noisy labels.

| ID | Dataset | fine | medium | coarse | a_t_b_f | a_f_b_t | a_t_b_t | b_t_c_f | b_f_c_t | b_t_c_t | c_t_a_f | c_f_a_t | c_t_a_t |
|----|---------|------|--------|--------|---------|---------|---------|---------|---------|---------|---------|---------|---------|
| 1 | ACSF1 | 0.6080 | 0.6640 | 0.6620 | 8 | 14 | 52 | 4 | 4 | 63 | 16 | 11 | 50 |
| 2 | Adiac | 0.0281 | 0.4890 | 0.5243 | 11 | 191 | 0 | 18 | 32 | 173 | 205 | 11 | 0 |
| 3 | AllGestureWiimoteX | 0.4089 | 0.5526 | 0.5666 | 34 | 135 | 252 | 22 | 31 | 365 | 149 | 39 | 247 |
| 4 | AllGestureWiimoteY | 0.4526 | 0.6643 | 0.6494 | 47 | 196 | 269 | 27 | 17 | 438 | 188 | 50 | 266 |
| 5 | AllGestureWiimoteZ | 0.4863 | 0.6449 | 0.6431 | 39 | 150 | 301 | 31 | 30 | 420 | 164 | 55 | 286 |
| 6 | ArrowHead | 0.4194 | 0.5886 | 0.6171 | 11 | 41 | 62 | 2 | 7 | 101 | 46 | 11 | 62 |
| 7 | BME | 0.5347 | 0.7333 | 0.7933 | 6 | 36 | 74 | 0 | 9 | 110 | 45 | 6 | 74 |
| 8 | Beef | 0.2667 | 0.4000 | 0.3800 | 0 | 4 | 8 | 1 | 0 | 11 | 3 | 0 | 8 |
| 9 | BeetleFly | 0.6000 | 0.8500 | 0.8500 | 0 | 5 | 12 | 0 | 0 | 17 | 5 | 0 | 12 |
| 10 | BirdChicken | 0.9600 | 0.8500 | 0.9500 | 3 | 1 | 16 | 1 | 3 | 16 | 1 | 1 | 18 |
| 11 | CBF | 0.7611 | 0.8311 | 0.8724 | 23 | 86 | 662 | 2 | 40 | 746 | 109 | 8 | 677 |
| 12 | Car | 0.4033 | 0.6633 | 0.6733 | 3 | 18 | 22 | 1 | 2 | 39 | 19 | 3 | 22 |
| 13 | Chinatown | 0.7304 | 0.7391 | 0.7391 | 5 | 8 | 247 | 8 | 8 | 247 | 3 | 0 | 252 |
| 14 | ChlorineConcentration | 0.5724 | 0.6120 | 0.6117 | 265 | 417 | 1933 | 92 | 91 | 2258 | 484 | 333 | 1865 |
| 15 | CinCECGTorso | 0.5097 | 0.5007 | 0.5506 | 148 | 136 | 555 | 34 | 102 | 657 | 145 | 89 | 614 |
| 16 | Coffee | 1.0000 | 1.0000 | 1.0000 | 0 | 0 | 28 | 0 | 0 | 28 | 0 | 0 | 28 |
| 17 | Computers | 0.6824 | 0.7280 | 0.7200 | 2 | 14 | 168 | 4 | 2 | 178 | 12 | 3 | 168 |
| 18 | CricketX | 0.3923 | 0.6733 | 0.6867 | 16 | 126 | 137 | 9 | 14 | 254 | 132 | 18 | 135 |
| 19 | CricketY | 0.4549 | 0.5990 | 0.5923 | 25 | 81 | 152 | 16 | 13 | 218 | 85 | 31 | 146 |
| 20 | CricketZ | 0.4764 | 0.6949 | 0.6995 | 23 | 108 | 163 | 11 | 12 | 260 | 118 | 31 | 155 |
| 21 | Crop | 0.7035 | 0.7443 | 0.7444 | 545 | 1231 | 11273 | 215 | 217 | 12289 | 1292 | 604 | 11214 |
| 22 | DiatomSizeReduction | 0.8105 | 0.8333 | 0.8268 | 0 | 7 | 248 | 2 | 0 | 253 | 5 | 0 | 248 |
| 23 | DistalPhalanxOutlineAgeGroup | 0.6619 | 0.6647 | 0.6662 | 13 | 14 | 79 | 5 | 5 | 88 | 18 | 17 | 75 |
| 24 | DistalPhalanxOutlineCorrect | 0.7949 | 0.7986 | 0.7913 | 12 | 13 | 208 | 3 | 1 | 217 | 14 | 15 | 205 |
| 25 | DistalPhalanxTW | 0.6806 | 0.6806 | 0.6259 | 8 | 8 | 87 | 11 | 3 | 84 | 9 | 16 | 78 |
| 26 | DodgerLoopDay | 0.3500 | 0.3750 | 0.3625 | 12 | 14 | 16 | 3 | 2 | 27 | 14 | 13 | 15 |
| 27 | DodgerLoopGame | 0.5942 | 0.5087 | 0.5000 | 20 | 8 | 62 | 19 | 18 | 51 | 22 | 35 | 47 |
| 28 | DodgerLoopWeekend | 0.8826 | 0.8957 | 0.8696 | 1 | 3 | 120 | 5 | 1 | 119 | 0 | 2 | 120 |
| 29 | ECG200 | 0.7500 | 0.7600 | 0.7700 | 1 | 2 | 74 | 0 | 1 | 76 | 2 | 0 | 75 |
| 30 | ECG5000 | 0.9313 | 0.9415 | 0.9411 | 50 | 96 | 4141 | 15 | 13 | 4221 | 103 | 59 | 4132 |
| 31 | ECGFiveDays | 0.6039 | 0.6139 | 0.6049 | 14 | 23 | 506 | 15 | 7 | 513 | 11 | 10 | 510 |
| 32 | EOGHorizontalSignal | 0.3994 | 0.4608 | 0.5249 | 28 | 50 | 117 | 8 | 32 | 158 | 76 | 30 | 114 |
| 33 | EOGVerticalSignal | 0.2790 | 0.3315 | 0.3718 | 13 | 32 | 88 | 5 | 20 | 115 | 44 | 10 | 91 |
| 34 | Earthquakes | 0.7482 | 0.7468 | 0.7482 | 3 | 3 | 101 | 4 | 5 | 99 | 7 | 7 | 97 |
| 35 | ElectricDevices | 0.6540 | 0.6774 | 0.7035 | 289 | 470 | 4754 | 136 | 337 | 5087 | 718 | 336 | 4707 |
| 36 | EthanolLevel | 0.2520 | 0.5628 | 0.5712 | 46 | 202 | 80 | 15 | 19 | 267 | 208 | 48 | 78 |
| 37 | FaceAll | 0.6820 | 0.7729 | 0.7557 | 92 | 246 | 1060 | 40 | 11 | 1266 | 225 | 101 | 1052 |
| 38 | FaceFour | 0.3182 | 0.5023 | 0.5159 | 5 | 21 | 23 | 2 | 3 | 42 | 19 | 2 | 26 |
| 39 | FacesUCR | 0.5544 | 0.8270 | 0.8037 | 36 | 594 | 1101 | 111 | 63 | 1584 | 575 | 64 | 1073 |
| 40 | FiftyWords | 0.2725 | 0.4295 | 0.5095 | 2 | 73 | 122 | 21 | 58 | 174 | 112 | 4 | 120 |
| 41 | Fish | 0.4114 | 0.7074 | 0.7211 | 2 | 53 | 70 | 7 | 9 | 117 | 58 | 4 | 68 |
| 42 | FordA | 0.9124 | 0.9189 | 0.9235 | 25 | 34 | 1179 | 4 | 10 | 1209 | 42 | 27 | 1177 |
| 43 | FordB | 0.7719 | 0.7968 | 0.8000 | 37 | 57 | 588 | 8 | 11 | 637 | 65 | 42 | 583 |
| 44 | FreezerRegularTrain | 0.9062 | 0.8516 | 0.8407 | 220 | 65 | 2363 | 43 | 12 | 2384 | 72 | 258 | 2324 |
| 45 | FreezerSmallTrain | 0.6724 | 0.7077 | 0.5981 | 87 | 188 | 1829 | 434 | 122 | 1583 | 72 | 283 | 1633 |
| 46 | Fungi | 0.0591 | 0.3430 | 0.2634 | 0 | 53 | 11 | 24 | 9 | 40 | 38 | 0 | 11 |
| 47 | GestureMidAirD1 | 0.2492 | 0.4108 | 0.4323 | 5 | 26 | 27 | 6 | 9 | 47 | 31 | 7 | 25 |
| 48 | GestureMidAirD2 | 0.2154 | 0.3215 | 0.3462 | 7 | 21 | 21 | 5 | 8 | 37 | 24 | 7 | 21 |
| 49 | GestureMidAirD3 | 0.1308 | 0.1631 | 0.2385 | 1 | 5 | 16 | 2 | 12 | 19 | 16 | 2 | 15 |
| 50 | GesturePebbleZ1 | 0.8233 | 0.7116 | 0.7256 | 25 | 6 | 117 | 3 | 6 | 119 | 8 | 24 | 117 |
| 51 | GesturePebbleZ2 | 0.8025 | 0.8228 | 0.8127 | 8 | 11 | 119 | 4 | 2 | 126 | 13 | 11 | 116 |
| 52 | GunPoint | 0.8293 | 0.7840 | 0.7600 | 25 | 19 | 99 | 5 | 1 | 113 | 20 | 30 | 94 |
| 53 | GunPointAgeSpan | 0.4937 | 0.5127 | 0.5823 | 1 | 7 | 155 | 3 | 25 | 159 | 32 | 4 | 152 |
| 54 | GunPointMaleVersusFemale | 0.9968 | 0.9671 | 0.9620 | 9 | 0 | 306 | 2 | 0 | 304 | 0 | 11 | 304 |
| 55 | GunPointOldVersusYoung | 1.0000 | 1.0000 | 1.0000 | 0 | 0 | 315 | 0 | 0 | 315 | 0 | 0 | 315 |
| 56 | Ham | 0.5905 | 0.6286 | 0.6381 | 3 | 7 | 59 | 3 | 4 | 63 | 9 | 4 | 58 |
| 57 | HandOutlines | 0.6405 | 0.7870 | 0.8205 | 34 | 88 | 203 | 7 | 20 | 284 | 99 | 33 | 204 |
| 58 | Haptics | 0.2851 | 0.3890 | 0.3870 | 35 | 67 | 53 | 11 | 10 | 109 | 72 | 41 | 47 |
| 59 | Herring | 0.5938 | 0.6719 | 0.6656 | 7 | 12 | 31 | 2 | 2 | 41 | 13 | 8 | 30 |
| 60 | HouseTwenty | 0.6202 | 0.6773 | 0.6739 | 1 | 8 | 73 | 2 | 2 | 78 | 7 | 0 | 73 |
| 61 | InlineSkate | 0.1585 | 0.2691 | 0.2949 | 30 | 91 | 57 | 20 | 34 | 128 | 119 | 44 | 43 |
| 62 | InsectEPGRegularTrain | 1.0000 | 1.0000 | 1.0000 | 0 | 0 | 249 | 0 | 0 | 249 | 0 | 0 | 249 |
| 63 | InsectEPGSmallTrain | 1.0000 | 1.0000 | 1.0000 | 0 | 0 | 249 | 0 | 0 | 249 | 0 | 0 | 249 |
| 64 | InsectWingbeatSound | 0.2799 | 0.4133 | 0.4266 | 131 | 395 | 423 | 109 | 135 | 710 | 467 | 176 | 378 |
| 65 | ItalyPowerDemand | 0.8871 | 0.8892 | 0.8908 | 4 | 6 | 909 | 2 | 4 | 913 | 8 | 4 | 909 |
| 66 | LargeKitchenAppliances | 0.8747 | 0.8645 | 0.8667 | 13 | 9 | 315 | 6 | 7 | 318 | 10 | 13 | 315 |
| 67 | Lightning2 | 0.6393 | 0.6066 | 0.6230 | 3 | 1 | 36 | 1 | 2 | 36 | 0 | 1 | 38 |
| 68 | Lightning7 | 0.3562 | 0.5068 | 0.5397 | 6 | 17 | 20 | 2 | 4 | 35 | 20 | 7 | 19 |
| 69 | Mallat | 0.2473 | 0.7108 | 0.7317 | 3 | 1090 | 577 | 109 | 158 | 1558 | 1204 | 69 | 511 |
| 70 | Meat | 0.3333 | 0.7100 | 0.8233 | 1 | 24 | 19 | 2 | 8 | 41 | 31 | 2 | 18 |
| 71 | MedicalImages | 0.5797 | 0.7153 | 0.7045 | 56 | 159 | 384 | 33 | 25 | 511 | 172 | 77 | 364 |
| 72 | MelbournePedestrian | 0.8996 | 0.9343 | 0.9518 | 11 | 96 | 2193 | 11 | 53 | 2278 | 146 | 18 | 2186 |
| 73 | MiddlePhalanxOutlineAgeGroup | 0.6104 | 0.5260 | 0.4961 | 26 | 13 | 68 | 10 | 5 | 71 | 16 | 34 | 60 |
| 74 | MiddlePhalanxOutlineCorrect | 0.5704 | 0.6227 | 0.6701 | 8 | 23 | 158 | 16 | 30 | 165 | 53 | 24 | 142 |
| 75 | MiddlePhalanxTW | 0.5195 | 0.5351 | 0.5091 | 12 | 14 | 68 | 8 | 4 | 75 | 17 | 19 | 61 |
| 76 | MixedShapesRegularTrain | 0.9326 | 0.9485 | 0.9425 | 47 | 85 | 2215 | 17 | 3 | 2283 | 81 | 57 | 2205 |
| 77 | MixedShapesSmallTrain | 0.4317 | 0.7070 | 0.7095 | 151 | 819 | 896 | 106 | 112 | 1609 | 874 | 200 | 847 |
| 78 | MoteStrain | 0.5391 | 0.6193 | 0.5391 | 102 | 202 | 573 | 202 | 102 | 573 | 0 | 0 | 675 |
| 79 | NonInvasiveFetalECGThorax1 | 0.0249 | 0.7906 | 0.8754 | 1 | 1506 | 48 | 44 | 211 | 1509 | 1680 | 9 | 40 |
| 80 | NonInvasiveFetalECGThorax2 | 0.0249 | 0.7719 | 0.8101 | 1 | 1469 | 48 | 140 | 215 | 1377 | 1544 | 1 | 48 |
| 81 | OSULeaf | 0.8372 | 0.8132 | 0.8289 | 12 | 6 | 190 | 2 | 6 | 195 | 8 | 10 | 192 |
| 82 | OliveOil | 0.4000 | 0.6000 | 0.6667 | 1 | 7 | 11 | 1 | 3 | 17 | 9 | 1 | 11 |
| 83 | PLAID | 0.3289 | 0.2205 | 0.2272 | 61 | 3 | 116 | 5 | 9 | 113 | 7 | 62 | 115 |
| 84 | PhalangesOutlinesCorrect | 0.6720 | 0.6911 | 0.7193 | 27 | 43 | 550 | 14 | 38 | 579 | 80 | 39 | 537 |
| 85 | Phoneme | 0.2346 | 0.2684 | 0.2545 | 114 | 178 | 330 | 86 | 60 | 423 | 179 | 141 | 304 |
| 86 | PickupGestureWiimoteZ | 0.3480 | 0.6600 | 0.6600 | 3 | 19 | 14 | 7 | 7 | 26 | 23 | 7 | 10 |
| 87 | PigAirwayPressure | 0.0654 | 0.1933 | 0.1452 | 4 | 31 | 10 | 17 | 7 | 23 | 23 | 7 | 7 |
| 88 | PigArtPressure | 0.0673 | 0.2663 | 0.2923 | 5 | 47 | 9 | 12 | 17 | 44 | 50 | 3 | 11 |
| 89 | PigCVP | 0.0721 | 0.1885 | 0.1538 | 8 | 32 | 7 | 15 | 8 | 24 | 22 | 5 | 10 |
| 90 | Plane | 1.0000 | 1.0000 | 1.0000 | 0 | 0 | 105 | 0 | 0 | 105 | 0 | 0 | 105 |
| 91 | PowerCons | 0.8422 | 0.8367 | 0.8322 | 5 | 4 | 147 | 3 | 2 | 148 | 5 | 6 | 145 |
| 92 | ProximalPhalanxOutlineAgeGroup | 0.8537 | 0.8224 | 0.7863 | 10 | 4 | 165 | 7 | 0 | 161 | 3 | 17 | 158 |
| 93 | ProximalPhalanxOutlineCorrect | 0.7216 | 0.8082 | 0.8048 | 8 | 33 | 202 | 5 | 4 | 230 | 37 | 13 | 197 |
| 94 | ProximalPhalanxTW | 0.8098 | 0.7951 | 0.7951 | 5 | 2 | 161 | 0 | 0 | 163 | 2 | 5 | 161 |
| 95 | RefrigerationDevices | 0.5061 | 0.5429 | 0.5248 | 9 | 23 | 181 | 15 | 8 | 188 | 26 | 19 | 171 |
| 96 | Rock | 0.4400 | 0.3880 | 0.3400 | 4 | 1 | 18 | 3 | 1 | 16 | 0 | 5 | 17 |
| 97 | ScreenType | 0.6325 | 0.5888 | 0.5744 | 41 | 24 | 197 | 17 | 11 | 204 | 25 | 47 | 190 |
| 98 | SemgHandGenderCh2 | 0.6970 | 0.7277 | 0.7410 | 25 | 43 | 394 | 7 | 15 | 429 | 52 | 26 | 392 |
| 99 | SemgHandMovementCh2 | 0.4622 | 0.5933 | 0.5747 | 28 | 87 | 180 | 24 | 16 | 243 | 87 | 37 | 171 |
| 100 | SemgHandSubjectCh2 | 0.5556 | 0.6929 | 0.7102 | 29 | 91 | 221 | 18 | 25 | 294 | 112 | 42 | 208 |
| 101 | ShakeGestureWiimoteZ | 0.3400 | 0.6400 | 0.6320 | 1 | 16 | 16 | 2 | 2 | 30 | 17 | 2 | 15 |
| 102 | ShapeletSim | 0.9367 | 0.8833 | 0.8000 | 12 | 2 | 157 | 25 | 10 | 134 | 7 | 32 | 137 |
| 103 | ShapesAll | 0.1863 | 0.7257 | 0.7477 | 7 | 331 | 105 | 26 | 39 | 409 | 343 | 6 | 106 |
| 104 | SmallKitchenAppliances | 0.8091 | 0.8000 | 0.8000 | 22 | 19 | 281 | 2 | 2 | 298 | 21 | 24 | 279 |
| 105 | SmoothSubspace | 0.9067 | 0.8933 | 0.9000 | 5 | 3 | 131 | 0 | 1 | 134 | 3 | 4 | 132 |
| 106 | SonyAIBORobotSurface1 | 0.9621 | 0.8133 | 0.7874 | 97 | 8 | 481 | 20 | 4 | 469 | 8 | 113 | 465 |
| 107 | SonyAIBORobotSurface2 | 0.9318 | 0.9328 | 0.9224 | 16 | 17 | 872 | 19 | 9 | 870 | 15 | 24 | 864 |
| 108 | StarLightCurves | 0.8559 | 0.9372 | 0.9582 | 96 | 765 | 6953 | 68 | 241 | 7651 | 998 | 155 | 6894 |
| 109 | Strawberry | 0.9216 | 0.9351 | 0.9341 | 3 | 8 | 338 | 2 | 2 | 344 | 8 | 3 | 338 |
| 110 | SwedishLeaf | 0.8416 | 0.9600 | 0.9536 | 6 | 80 | 520 | 7 | 3 | 593 | 78 | 8 | 518 |
| 111 | Symbols | 0.5809 | 0.6036 | 0.6080 | 1 | 24 | 577 | 1 | 5 | 600 | 28 | 1 | 577 |
| 112 | SyntheticControl | 0.9667 | 0.9667 | 0.9633 | 2 | 2 | 288 | 1 | 0 | 289 | 2 | 3 | 287 |
| 113 | ToeSegmentation1 | 0.8596 | 0.7851 | 0.7368 | 25 | 8 | 171 | 12 | 1 | 167 | 3 | 31 | 165 |
| 114 | ToeSegmentation2 | 0.8692 | 0.8215 | 0.8385 | 8 | 2 | 105 | 0 | 2 | 107 | 3 | 7 | 106 |
| 115 | Trace | 0.9600 | 0.8900 | 0.9200 | 9 | 2 | 87 | 0 | 3 | 89 | 2 | 6 | 90 |
| 116 | TwoLeadECG | 0.6673 | 0.6939 | 0.6939 | 18 | 48 | 742 | 11 | 11 | 780 | 54 | 24 | 736 |
| 117 | TwoPatterns | 0.8599 | 0.8959 | 0.9063 | 66 | 210 | 3374 | 11 | 53 | 3572 | 239 | 53 | 3387 |
| 118 | UMD | 0.5653 | 0.6875 | 0.7806 | 3 | 21 | 78 | 0 | 13 | 99 | 33 | 2 | 79 |
| 119 | UWaveGestureLibraryAll | 0.5499 | 0.7699 | 0.8018 | 112 | 900 | 1858 | 87 | 201 | 2671 | 1058 | 156 | 1814 |
| 120 | UWaveGestureLibraryX | 0.7054 | 0.7208 | 0.7226 | 229 | 284 | 2297 | 109 | 115 | 2473 | 363 | 302 | 2225 |
| 121 | UWaveGestureLibraryY | 0.5842 | 0.6321 | 0.6398 | 212 | 384 | 1880 | 111 | 138 | 2154 | 468 | 269 | 1823 |
| 122 | UWaveGestureLibraryZ | 0.6245 | 0.6887 | 0.7131 | 197 | 427 | 2040 | 139 | 227 | 2328 | 605 | 287 | 1950 |
| 123 | Wafer | 0.8921 | 0.8834 | 0.8807 | 95 | 41 | 5404 | 68 | 51 | 5377 | 81 | 151 | 5348 |
| 124 | Wine | 0.5000 | 0.5556 | 0.5407 | 6 | 9 | 21 | 3 | 2 | 27 | 11 | 9 | 18 |
| 125 | WordSynonyms | 0.2429 | 0.3210 | 0.3495 | 31 | 80 | 124 | 10 | 28 | 195 | 98 | 30 | 125 |
| 126 | Worms | 0.6623 | 0.7714 | 0.7532 | 4 | 12 | 47 | 1 | 0 | 58 | 12 | 5 | 46 |
| 127 | WormsTwoClass | 0.8052 | 0.7792 | 0.7792 | 3 | 1 | 59 | 4 | 4 | 56 | 5 | 7 | 55 |
| 128 | Yoga | 0.6559 | 0.6507 | 0.6540 | 163 | 147 | 1805 | 18 | 28 | 1934 | 150 | 156 | 1812 |
| | **Avg Value** | 0.5967 | 0.6817 | 0.6870 | 38 | 126 | 629 | 26 | 32 | 729 | 141 | 47 | 620 |

Table 20: Multi-scale analysis of w/o cross-scale fusion based on Scale-teaching using UCR 128 archive with Sym 20% noisy labels.

| ID | Dataset | fine | medium | coarse | a_t_b_f | a_f_b_t | a_t_b_t | b_t_c_f | b_f_c_t | b_t_c_t | c_t_a_f | c_f_a_t | c_t_a_t |
|----|---------|------|--------|--------|---------|---------|---------|---------|---------|---------|---------|---------|---------|
| 1 | ACSF1 | 0.5500 | 0.0100 | 0.2500 | 54 | 0 | 1 | 1 | 25 | 0 | 7 | 37 | 18 |
| 2 | Adiac | 0.1074 | 0.0327 | 0.0077 | 34 | 5 | 8 | 13 | 3 | 0 | 1 | 40 | 2 |
| 3 | AllGestureWiimoteX | 0.4697 | 0.1337 | 0.0980 | 280 | 45 | 49 | 43 | 18 | 51 | 32 | 293 | 36 |
| 4 | AllGestureWiimoteY | 0.5503 | 0.0543 | 0.0994 | 360 | 13 | 25 | 38 | 70 | 0 | 5 | 321 | 65 |
| 5 | AllGestureWiimoteZ | 0.5723 | 0.1109 | 0.0977 | 348 | 25 | 53 | 77 | 68 | 1 | 68 | 400 | 0 |
| 6 | ArrowHead | 0.6366 | 0.3029 | 0.4114 | 93 | 34 | 19 | 34 | 53 | 19 | 8 | 48 | 64 |
| 7 | BME | 0.4747 | 0.3467 | 0.5653 | 21 | 2 | 50 | 8 | 41 | 44 | 41 | 27 | 44 |
| 8 | Beef | 0.3667 | 0.2000 | 0.2000 | 10 | 5 | 1 | 6 | 6 | 0 | 3 | 8 | 3 |
| 9 | BeetleFly | 0.8000 | 0.5000 | 0.5000 | 10 | 4 | 6 | 0 | 0 | 10 | 4 | 10 | 6 |
| 10 | BirdChicken | 0.8500 | 0.5000 | 0.5000 | 9 | 2 | 8 | 0 | 0 | 10 | 2 | 9 | 8 |
| 11 | CBF | 0.8978 | 0.3311 | 0.4153 | 510 | 0 | 298 | 0 | 76 | 298 | 21 | 455 | 353 |
| 12 | Car | 0.6667 | 0.0833 | 0.2167 | 35 | 0 | 5 | 5 | 13 | 0 | 1 | 28 | 12 |
| 13 | Chinatown | 0.7438 | 0.7391 | 0.7246 | 8 | 6 | 249 | 5 | 0 | 250 | 4 | 11 | 246 |
| 14 | ChlorineConcentration | 0.5973 | 0.2367 | 0.5326 | 2091 | 707 | 202 | 909 | 2045 | 0 | 377 | 625 | 1668 |
| 15 | CinCECGTorso | 0.4768 | 0.0845 | 0.1759 | 588 | 46 | 70 | 115 | 241 | 2 | 38 | 453 | 205 |
| 16 | Coffee | 1.0000 | 0.4643 | 0.4643 | 15 | 0 | 13 | 0 | 0 | 13 | 0 | 15 | 13 |
| 17 | Computers | 0.6960 | 0.5984 | 0.5000 | 54 | 30 | 120 | 43 | 18 | 107 | 46 | 95 | 79 |
| 18 | CricketX | 0.4769 | 0.1723 | 0.0949 | 125 | 6 | 61 | 67 | 37 | 0 | 33 | 182 | 4 |
| 19 | CricketY | 0.4785 | 0.0846 | 0.1149 | 155 | 1 | 32 | 33 | 45 | 0 | 22 | 163 | 23 |
| 20 | CricketZ | 0.4769 | 0.1564 | 0.1631 | 142 | 17 | 44 | 45 | 48 | 16 | 8 | 131 | 55 |
| 21 | Crop | 0.6986 | 0.1058 | 0.0225 | 10215 | 256 | 1522 | 1767 | 368 | 11 | 193 | 11551 | 186 |
| 22 | DiatomSizeReduction | 0.7320 | 0.2915 | 0.2039 | 194 | 59 | 30 | 81 | 54 | 8 | 17 | 179 | 45 |
| 23 | DistalPhalanxOutlineAgeGroup | 0.7626 | 0.5122 | 0.4604 | 46 | 11 | 60 | 58 | 51 | 13 | 18 | 60 | 46 |
| 24 | DistalPhalanxOutlineCorrect | 0.7826 | 0.5971 | 0.5833 | 68 | 17 | 148 | 8 | 4 | 157 | 21 | 76 | 140 |
| 25 | DistalPhalanxTW | 0.7065 | 0.2806 | 0.2806 | 63 | 4 | 35 | 0 | 0 | 39 | 4 | 63 | 35 |
| 26 | DodgerLoopDay | 0.2450 | 0.1125 | 0.2000 | 19 | 8 | 1 | 8 | 15 | 1 | 16 | 20 | 0 |
| 27 | DodgerLoopGame | 0.7261 | 0.4783 | 0.4783 | 51 | 16 | 50 | 0 | 0 | 66 | 16 | 51 | 50 |
| 28 | DodgerLoopWeekend | 0.9029 | 0.2609 | 0.2609 | 99 | 10 | 26 | 0 | 0 | 36 | 10 | 99 | 26 |
| 29 | ECG200 | 0.6940 | 0.6440 | 0.6400 | 6 | 1 | 63 | 0 | 0 | 64 | 1 | 6 | 63 |
| 30 | ECG5000 | 0.9396 | 0.2929 | 0.0191 | 2940 | 30 | 1288 | 1314 | 82 | 4 | 63 | 4205 | 23 |
| 31 | ECGFiveDays | 0.6156 | 0.5029 | 0.5029 | 375 | 278 | 155 | 0 | 0 | 433 | 278 | 375 | 155 |
| 32 | EOGHorizontalSignal | 0.4779 | 0.1492 | 0.0304 | 134 | 15 | 39 | 53 | 10 | 1 | 8 | 170 | 3 |
| 33 | EOGVerticalSignal | 0.3265 | 0.0657 | 0.0801 | 100 | 6 | 18 | 11 | 16 | 13 | 15 | 104 | 14 |
| 34 | Earthquakes | 0.7511 | 0.2518 | 0.2518 | 104 | 35 | 0 | 0 | 0 | 35 | 35 | 104 | 0 |
| 35 | ElectricDevices | 0.7407 | 0.0929 | 0.0929 | 5458 | 463 | 254 | 642 | 641 | 75 | 187 | 5182 | 530 |
| 36 | EthanolLevel | 0.4480 | 0.2392 | 0.2408 | 121 | 17 | 103 | 118 | 118 | 2 | 120 | 223 | 1 |
| 37 | FaceAll | 0.9155 | 0.0643 | 0.0095 | 1447 | 8 | 100 | 109 | 16 | 0 | 5 | 1536 | 11 |
| 38 | FaceFour | 0.4545 | 0.0455 | 0.1591 | 40 | 4 | 0 | 3 | 13 | 1 | 5 | 31 | 9 |
| 39 | FacesUCR | 0.8332 | 0.1257 | 0.0474 | 1456 | 6 | 252 | 258 | 97 | 0 | 10 | 1621 | 87 |
| 40 | FiftyWords | 0.3143 | 0.0088 | 0.0400 | 142 | 3 | 1 | 4 | 18 | 0 | 4 | 128 | 15 |
| 41 | Fish | 0.6629 | 0.0457 | 0.2571 | 111 | 3 | 5 | 7 | 44 | 1 | 2 | 73 | 43 |
| 42 | FordA | 0.9098 | 0.6244 | 0.4841 | 425 | 48 | 776 | 200 | 15 | 624 | 60 | 622 | 579 |
| 43 | FordB | 0.7788 | 0.5521 | 0.5049 | 275 | 91 | 356 | 47 | 9 | 400 | 100 | 322 | 309 |
| 44 | FreezerRegularTrain | 0.9882 | 0.5004 | 0.5000 | 1409 | 19 | 1407 | 1 | 0 | 1425 | 19 | 1410 | 1406 |
| 45 | FreezerSmallTrain | 0.6216 | 0.4220 | 0.5000 | 1167 | 598 | 605 | 164 | 387 | 1038 | 705 | 1052 | 720 |
| 46 | Fungi | 0.1882 | 0.0710 | 0.0806 | 35 | 13 | 0 | 13 | 15 | 0 | 15 | 35 | 0 |
| 47 | GestureMidAirD1 | 0.2631 | 0.0308 | 0.0385 | 31 | 1 | 3 | 4 | 5 | 0 | 3 | 32 | 2 |
| 48 | GestureMidAirD2 | 0.2385 | 0.0231 | 0.0185 | 28 | 0 | 3 | 3 | 2 | 0 | 0 | 29 | 2 |
| 49 | GestureMidAirD3 | 0.1538 | 0.0308 | 0.0323 | 19 | 3 | 1 | 3 | 3 | 1 | 2 | 18 | 2 |
| 50 | GesturePebbleZ1 | 0.7395 | 0.1919 | 0.3198 | 114 | 20 | 13 | 1 | 23 | 32 | 21 | 93 | 34 |
| 51 | GesturePebbleZ2 | 0.8139 | 0.1329 | 0.2595 | 109 | 1 | 20 | 18 | 38 | 3 | 19 | 106 | 22 |
| 52 | GunPoint | 1.0000 | 0.9533 | 0.4933 | 7 | 0 | 143 | 69 | 0 | 74 | 0 | 76 | 74 |
| 53 | GunPointAgeSpan | 0.7506 | 0.5544 | 0.4937 | 67 | 5 | 170 | 84 | 65 | 91 | 70 | 151 | 86 |
| 54 | GunPointMaleVersusFemale | 0.9968 | 0.4399 | 0.4747 | 176 | 0 | 139 | 4 | 15 | 135 | 0 | 165 | 150 |
| 55 | GunPointOldVersusYoung | 1.0000 | 0.9873 | 0.5238 | 4 | 0 | 311 | 146 | 0 | 165 | 0 | 150 | 165 |
| 56 | Ham | 0.6381 | 0.5143 | 0.5143 | 35 | 22 | 32 | 0 | 0 | 54 | 22 | 35 | 32 |
| 57 | HandOutlines | 0.7741 | 0.6405 | 0.6405 | 81 | 31 | 206 | 0 | 0 | 237 | 31 | 81 | 206 |
| 58 | Haptics | 0.3662 | 0.1883 | 0.2370 | 77 | 22 | 36 | 58 | 73 | 0 | 27 | 67 | 46 |
| 59 | Herring | 0.7188 | 0.4062 | 0.4062 | 31 | 11 | 15 | 0 | 0 | 26 | 11 | 31 | 15 |
| 60 | HouseTwenty | 0.9328 | 0.4202 | 0.4202 | 62 | 1 | 49 | 0 | 0 | 50 | 1 | 62 | 49 |
| 61 | InlineSkate | 0.1880 | 0.1509 | 0.1793 | 99 | 79 | 4 | 23 | 39 | 60 | 85 | 90 | 14 |
| 62 | InsectEPGRegularTrain | 0.8313 | 0.6426 | 0.0008 | 89 | 42 | 118 | 160 | 0 | 0 | 0 | 207 | 0 |
| 63 | InsectEPGSmallTrain | 1.0000 | 0.4739 | 0.0161 | 131 | 0 | 118 | 114 | 0 | 4 | 0 | 245 | 4 |
| 64 | InsectWingbeatSound | 0.3357 | 0.1285 | 0.0943 | 519 | 109 | 146 | 240 | 173 | 14 | 70 | 548 | 116 |
| 65 | ItalyPowerDemand | 0.9499 | 0.5015 | 0.5015 | 481 | 20 | 496 | 0 | 0 | 516 | 20 | 481 | 496 |
| 66 | LargeKitchenAppliances | 0.8240 | 0.3333 | 0.2880 | 205 | 21 | 104 | 125 | 108 | 0 | 22 | 223 | 86 |
| 67 | Lightning2 | 0.6066 | 0.6393 | 0.5410 | 0 | 2 | 37 | 9 | 3 | 30 | 5 | 9 | 28 |
| 68 | Lightning7 | 0.4110 | 0.3178 | 0.0849 | 11 | 4 | 19 | 23 | 6 | 0 | 6 | 30 | 0 |
| 69 | Mallat | 0.5092 | 0.2370 | 0.1454 | 674 | 36 | 520 | 509 | 294 | 47 | 0 | 853 | 341 |
| 70 | Meat | 0.6633 | 0.3333 | 0.1500 | 22 | 2 | 18 | 15 | 4 | 5 | 0 | 31 | 9 |
| 71 | MedicalImages | 0.6405 | 0.4261 | 0.1800 | 206 | 43 | 281 | 261 | 74 | 62 | 51 | 401 | 85 |
| 72 | MelbournePedestrian | 0.9030 | 0.0079 | 0.1012 | 2198 | 5 | 14 | 17 | 246 | 2 | 7 | 1972 | 241 |
| 73 | MiddlePhalanxOutlineAgeGroup | 0.5870 | 0.5714 | 0.1883 | 21 | 19 | 69 | 88 | 29 | 0 | 20 | 81 | 9 |
| 74 | MiddlePhalanxOutlineCorrect | 0.7993 | 0.6852 | 0.5704 | 51 | 18 | 182 | 40 | 7 | 159 | 23 | 90 | 143 |
| 75 | MiddlePhalanxTW | 0.5506 | 0.2506 | 0.1883 | 69 | 22 | 16 | 19 | 9 | 20 | 19 | 75 | 10 |
| 76 | MixedShapesRegularTrain | 0.9509 | 0.0134 | 0.1340 | 2277 | 3 | 29 | 32 | 325 | 0 | 7 | 1988 | 318 |
| 77 | MixedShapesSmallTrain | 0.6786 | 0.3753 | 0.1295 | 914 | 178 | 732 | 910 | 314 | 0 | 18 | 1350 | 296 |
| 78 | MoteStrain | 0.8997 | 0.4617 | 0.4609 | 632 | 84 | 494 | 9 | 8 | 569 | 92 | 641 | 485 |
| 79 | NonInvasiveFetalECGThorax1 | 0.2532 | 0.0193 | 0.0165 | 462 | 2 | 36 | 38 | 32 | 0 | 4 | 469 | 28 |
| 80 | NonInvasiveFetalECGThorax2 | 0.1859 | 0.0220 | 0.0025 | 324 | 2 | 41 | 43 | 5 | 0 | 5 | 365 | 0 |
| 81 | OSULeaf | 0.9628 | 0.1860 | 0.1322 | 188 | 0 | 45 | 45 | 32 | 0 | 1 | 202 | 31 |
| 82 | OliveOil | 0.4000 | 0.4000 | 0.4000 | 0 | 0 | 12 | 0 | 0 | 12 | 0 | 0 | 12 |
| 83 | PLAID | 0.3158 | 0.1713 | 0.1456 | 83 | 5 | 87 | 14 | 0 | 78 | 0 | 91 | 78 |
| 84 | PhalangesOutlinesCorrect | 0.7683 | 0.7536 | 0.6131 | 38 | 25 | 622 | 150 | 29 | 497 | 32 | 165 | 494 |
| 85 | Phoneme | 0.2429 | 0.0216 | 0.0195 | 448 | 29 | 12 | 40 | 36 | 1 | 23 | 447 | 14 |
| 86 | PickupGestureWiimoteZ | 0.3760 | 0.1000 | 0.1000 | 14 | 0 | 5 | 2 | 2 | 3 | 0 | 14 | 5 |
| 87 | PigAirwayPressure | 0.1058 | 0.0000 | 0.0385 | 22 | 0 | 0 | 0 | 8 | 0 | 4 | 18 | 4 |
| 88 | PigArtPressure | 0.1250 | 0.0058 | 0.0192 | 26 | 1 | 0 | 1 | 4 | 0 | 4 | 26 | 0 |
| 89 | PigCVP | 0.0577 | 0.0202 | 0.0192 | 12 | 4 | 0 | 4 | 4 | 0 | 4 | 12 | 0 |
| 90 | Plane | 1.0000 | 0.1238 | 0.2000 | 92 | 0 | 13 | 13 | 21 | 0 | 0 | 84 | 21 |
| 91 | PowerCons | 0.8389 | 0.5167 | 0.5000 | 67 | 9 | 84 | 3 | 0 | 90 | 9 | 70 | 81 |
| 92 | ProximalPhalanxOutlineAgeGroup | 0.8263 | 0.4293 | 0.1883 | 105 | 24 | 64 | 49 | 0 | 39 | 6 | 137 | 33 |
| 93 | ProximalPhalanxOutlineCorrect | 0.8165 | 0.7113 | 0.6838 | 32 | 1 | 206 | 9 | 1 | 198 | 2 | 41 | 197 |
| 94 | ProximalPhalanxTW | 0.8049 | 0.1951 | 0.1951 | 136 | 11 | 29 | 0 | 0 | 40 | 11 | 136 | 29 |
| 95 | RefrigerationDevices | 0.5296 | 0.3413 | 0.3120 | 142 | 71 | 57 | 114 | 103 | 14 | 85 | 167 | 32 |
| 96 | Rock | 0.3600 | 0.1920 | 0.3000 | 12 | 4 | 6 | 8 | 13 | 2 | 9 | 12 | 6 |
| 97 | ScreenType | 0.6240 | 0.3333 | 0.3237 | 144 | 35 | 90 | 90 | 86 | 35 | 47 | 159 | 75 |
| 98 | SemgHandGenderCh2 | 0.7017 | 0.3777 | 0.3500 | 267 | 73 | 154 | 17 | 0 | 210 | 72 | 283 | 138 |
| 99 | SemgHandMovementCh2 | 0.4862 | 0.1169 | 0.2138 | 215 | 49 | 3 | 44 | 88 | 9 | 23 | 146 | 73 |
| 100 | SemgHandSubjectCh2 | 0.5711 | 0.2116 | 0.2000 | 184 | 23 | 73 | 88 | 83 | 7 | 42 | 209 | 48 |
| 101 | ShakeGestureWiimoteZ | 0.4800 | 0.0840 | 0.1000 | 21 | 1 | 3 | 4 | 5 | 0 | 0 | 19 | 5 |
| 102 | ShapeletSim | 0.8944 | 0.5000 | 0.5000 | 74 | 3 | 87 | 0 | 0 | 90 | 3 | 74 | 87 |
| 103 | ShapesAll | 0.3500 | 0.0263 | 0.0000 | 205 | 11 | 5 | 16 | 0 | 0 | 0 | 210 | 0 |
| 104 | SmallKitchenAppliances | 0.7787 | 0.3621 | 0.2629 | 195 | 39 | 97 | 63 | 26 | 72 | 39 | 233 | 59 |
| 105 | SmoothSubspace | 0.9400 | 0.3333 | 0.3800 | 93 | 2 | 48 | 23 | 30 | 27 | 2 | 86 | 55 |
| 106 | SonyAIBORobotSurface1 | 0.8985 | 0.4725 | 0.4293 | 259 | 3 | 281 | 26 | 0 | 258 | 3 | 285 | 255 |
| 107 | SonyAIBORobotSurface2 | 0.9003 | 0.6252 | 0.6170 | 294 | 32 | 564 | 8 | 0 | 588 | 32 | 302 | 556 |
| 108 | StarLightCurves | 0.9740 | 0.2799 | 0.5517 | 5739 | 22 | 2283 | 2305 | 4544 | 0 | 45 | 3524 | 4499 |
| 109 | Strawberry | 0.9405 | 0.6432 | 0.6432 | 121 | 11 | 227 | 0 | 0 | 238 | 11 | 121 | 227 |
| 110 | SwedishLeaf | 0.9072 | 0.0640 | 0.0822 | 527 | 0 | 40 | 38 | 49 | 2 | 0 | 516 | 51 |
| 111 | Symbols | 0.5648 | 0.0000 | 0.1749 | 562 | 0 | 0 | 0 | 174 | 0 | 0 | 388 | 174 |
| 112 | SyntheticControl | 0.9733 | 0.1867 | 0.1827 | 236 | 0 | 56 | 49 | 48 | 7 | 1 | 238 | 54 |
| 113 | ToeSegmentation1 | 0.9342 | 0.4737 | 0.4737 | 110 | 5 | 103 | 0 | 0 | 108 | 5 | 110 | 103 |
| 114 | ToeSegmentation2 | 0.8123 | 0.1846 | 0.1846 | 88 | 6 | 18 | 0 | 0 | 24 | 6 | 88 | 18 |
| 115 | Trace | 0.9500 | 0.0800 | 0.3000 | 87 | 0 | 8 | 8 | 30 | 0 | 5 | 70 | 25 |
| 116 | TwoLeadECG | 0.7489 | 0.5004 | 0.5004 | 568 | 285 | 285 | 0 | 0 | 570 | 285 | 568 | 285 |
| 117 | TwoPatterns | 0.8628 | 0.0251 | 0.2523 | 3382 | 32 | 69 | 96 | 1005 | 4 | 214 | 2656 | 795 |
| 118 | UMD | 0.5861 | 0.3333 | 0.3681 | 36 | 0 | 48 | 0 | 5 | 48 | 4 | 35 | 49 |
| 119 | UWaveGestureLibraryAll | 0.6949 | 0.1428 | 0.0367 | 2037 | 59 | 452 | 491 | 111 | 20 | 37 | 2395 | 94 |
| 120 | UWaveGestureLibraryX | 0.7059 | 0.1940 | 0.1095 | 1944 | 110 | 584 | 683 | 380 | 12 | 144 | 2280 | 248 |
| 121 | UWaveGestureLibraryY | 0.5853 | 0.1704 | 0.0959 | 1660 | 174 | 437 | 609 | 342 | 1 | 88 | 1841 | 256 |
| 122 | UWaveGestureLibraryZ | 0.6559 | 0.1788 | 0.1128 | 1835 | 126 | 514 | 640 | 404 | 0 | 66 | 2012 | 338 |
| 123 | Wafer | 0.9639 | 0.8921 | 0.8921 | 559 | 117 | 5382 | 0 | 0 | 5499 | 117 | 559 | 5382 |
| 124 | Wine | 0.6000 | 0.5000 | 0.5000 | 25 | 20 | 7 | 0 | 0 | 27 | 20 | 25 | 7 |
| 125 | WordSynonyms | 0.3323 | 0.2223 | 0.1589 | 77 | 7 | 135 | 120 | 79 | 22 | 42 | 153 | 59 |
| 126 | Worms | 0.6701 | 0.1688 | 0.1818 | 43 | 4 | 9 | 13 | 14 | 0 | 8 | 46 | 6 |
| 127 | WormsTwoClass | 0.8052 | 0.5714 | 0.5714 | 22 | 4 | 40 | 0 | 0 | 44 | 4 | 22 | 40 |
| 128 | Yoga | 0.6902 | 0.5357 | 0.5357 | 867 | 403 | 1204 | 0 | 0 | 1607 | 403 | 867 | 1204 |
| | **Avg Value** | 0.6513 | 0.3011 | 0.2817 | 512 | 44 | 217 | 119 | 113 | 142 | 41 | 516 | 213 |

Table 21: Multi-scale analysis of Scale-teaching using UCR 128 archive with Asym 50% noisy labels.

| ID | Dataset | fine | medium | coarse | a_t_b_f | a_f_b_t | a_t_b_t | b_t_c_f | b_f_c_t | b_t_c_t | c_t_a_f | c_f_a_t | c_t_a_t |
|----|---------|------|--------|--------|---------|---------|---------|---------|---------|---------|---------|---------|---------|
| 1 | ACSF1 | 0.4380 | 0.5400 | 0.5000 | 0 | 10 | 44 | 6 | 2 | 48 | 7 | 1 | 43 |
| 2 | Adiac | 0.0179 | 0.1857 | 0.2517 | 2 | 68 | 5 | 1 | 26 | 72 | 93 | 2 | 5 |
| 3 | AllGestureWiimoteX | 0.3714 | 0.4643 | 0.4683 | 31 | 96 | 229 | 16 | 19 | 309 | 99 | 31 | 229 |
| 4 | AllGestureWiimoteY | 0.3514 | 0.4494 | 0.4603 | 32 | 101 | 214 | 13 | 21 | 302 | 115 | 39 | 207 |
| 5 | AllGestureWiimoteZ | 0.3429 | 0.4494 | 0.4829 | 20 | 94 | 220 | 17 | 41 | 297 | 127 | 29 | 211 |
| 6 | ArrowHead | 0.3531 | 0.3760 | 0.3920 | 13 | 17 | 48 | 1 | 4 | 65 | 20 | 13 | 48 |
| 7 | BME | 0.4867 | 0.5000 | 0.4933 | 0 | 2 | 73 | 2 | 1 | 73 | 2 | 1 | 72 |
| 8 | Beef | 0.2000 | 0.2000 | 0.2333 | 1 | 1 | 5 | 0 | 1 | 6 | 2 | 1 | 5 |
| 9 | BeetleFly | 0.4000 | 0.4500 | 0.4000 | 2 | 3 | 6 | 3 | 2 | 6 | 0 | 0 | 8 |
| 10 | BirdChicken | 0.8000 | 0.8000 | 0.7500 | 0 | 0 | 16 | 1 | 0 | 15 | 0 | 1 | 15 |
| 11 | CBF | 0.7727 | 0.9398 | 0.9422 | 42 | 192 | 653 | 2 | 4 | 844 | 194 | 41 | 654 |
| 12 | Car | 0.4000 | 0.4433 | 0.4400 | 4 | 7 | 20 | 5 | 4 | 22 | 9 | 7 | 17 |
| 13 | Chinatown | 0.5681 | 0.6841 | 0.6725 | 10 | 50 | 186 | 16 | 12 | 220 | 58 | 22 | 174 |
| 14 | ChlorineConcentration | 0.5141 | 0.5090 | 0.5110 | 198 | 178 | 1776 | 15 | 23 | 1940 | 195 | 207 | 1767 |
| 15 | CinCECGTorso | 0.3372 | 0.3342 | 0.3290 | 114 | 109 | 352 | 28 | 21 | 433 | 119 | 131 | 335 |
| 16 | Coffee | 0.5357 | 0.5357 | 0.5357 | 0 | 0 | 15 | 0 | 0 | 15 | 0 | 0 | 15 |
| 17 | Computers | 0.5120 | 0.4976 | 0.4880 | 7 | 3 | 121 | 3 | 1 | 121 | 3 | 9 | 119 |
| 18 | CricketX | 0.2744 | 0.3431 | 0.3585 | 10 | 37 | 97 | 12 | 18 | 122 | 52 | 19 | 88 |
| 19 | CricketY | 0.2533 | 0.3318 | 0.3472 | 32 | 62 | 67 | 14 | 20 | 116 | 78 | 42 | 57 |
| 20 | CricketZ | 0.2154 | 0.3554 | 0.3549 | 9 | 64 | 75 | 11 | 10 | 128 | 71 | 17 | 67 |
| 21 | Crop | 0.4823 | 0.5471 | 0.5557 | 578 | 1666 | 7525 | 358 | 503 | 8833 | 1956 | 724 | 7380 |
| 22 | DiatomSizeReduction | 0.5850 | 0.5850 | 0.5850 | 0 | 0 | 179 | 0 | 0 | 179 | 0 | 0 | 179 |
| 23 | DistalPhalanxOutlineAgeGroup | 0.4676 | 0.4676 | 0.4676 | 0 | 0 | 65 | 0 | 0 | 65 | 0 | 0 | 65 |
| 24 | DistalPhalanxOutlineCorrect | 0.6043 | 0.5942 | 0.5964 | 7 | 4 | 160 | 1 | 2 | 163 | 6 | 8 | 159 |
| 25 | DistalPhalanxTW | 0.6691 | 0.6921 | 0.6403 | 6 | 9 | 87 | 8 | 1 | 88 | 10 | 14 | 79 |
| 26 | DodgerLoopDay | 0.2325 | 0.3500 | 0.3350 | 1 | 10 | 18 | 2 | 1 | 26 | 8 | 0 | 18 |
| 27 | DodgerLoopGame | 0.5623 | 0.7362 | 0.7478 | 7 | 31 | 70 | 2 | 3 | 100 | 34 | 8 | 69 |
| 28 | DodgerLoopWeekend | 0.7826 | 0.8261 | 0.6783 | 1 | 7 | 107 | 20 | 0 | 94 | 1 | 16 | 92 |
| 29 | ECG200 | 0.6280 | 0.6360 | 0.6100 | 0 | 1 | 63 | 4 | 1 | 60 | 2 | 4 | 59 |
| 30 | ECG5000 | 0.8978 | 0.8909 | 0.8918 | 51 | 20 | 3989 | 5 | 9 | 4004 | 23 | 50 | 3990 |
| 31 | ECGFiveDays | 0.5029 | 0.5145 | 0.5419 | 1 | 11 | 432 | 51 | 75 | 392 | 85 | 51 | 382 |
| 32 | EOGHorizontalSignal | 0.3354 | 0.4006 | 0.4232 | 13 | 37 | 108 | 9 | 17 | 136 | 40 | 8 | 113 |
| 33 | EOGVerticalSignal | 0.3033 | 0.3271 | 0.3481 | 11 | 20 | 99 | 6 | 14 | 112 | 31 | 15 | 95 |
| 34 | Earthquakes | 0.7482 | 0.7482 | 0.7482 | 0 | 0 | 104 | 0 | 0 | 104 | 0 | 0 | 104 |
| 35 | ElectricDevices | 0.6258 | 0.5982 | 0.5976 | 334 | 121 | 4492 | 126 | 122 | 4486 | 210 | 428 | 4398 |
| 36 | EthanolLevel | 0.2480 | 0.3632 | 0.4076 | 107 | 164 | 17 | 16 | 38 | 166 | 190 | 110 | 14 |
| 37 | FaceAll | 0.4463 | 0.4531 | 0.4574 | 48 | 60 | 706 | 35 | 42 | 731 | 79 | 61 | 694 |
| 38 | FaceFour | 0.4545 | 0.4614 | 0.5773 | 2 | 3 | 38 | 3 | 14 | 37 | 14 | 3 | 37 |
| 39 | FacesUCR | 0.3937 | 0.4758 | 0.4802 | 73 | 241 | 734 | 68 | 77 | 908 | 272 | 94 | 713 |
| 40 | FiftyWords | 0.1934 | 0.2901 | 0.3090 | 5 | 49 | 83 | 14 | 23 | 118 | 63 | 10 | 78 |
| 41 | Fish | 0.1371 | 0.3406 | 0.4457 | 5 | 41 | 19 | 1 | 19 | 59 | 60 | 6 | 18 |
| 42 | FordA | 0.8623 | 0.8608 | 0.8659 | 14 | 12 | 1124 | 5 | 12 | 1131 | 24 | 19 | 1119 |
| 43 | FordB | 0.5914 | 0.5916 | 0.6020 | 6 | 7 | 473 | 2 | 10 | 477 | 15 | 7 | 472 |
| 44 | FreezerRegularTrain | 0.7446 | 0.7595 | 0.7544 | 21 | 64 | 2101 | 46 | 31 | 2119 | 69 | 41 | 2081 |
| 45 | FreezerSmallTrain | 0.5000 | 0.4947 | 0.4996 | 17 | 2 | 1408 | 3 | 17 | 1407 | 0 | 1 | 1424 |
| 46 | Fungi | 0.1022 | 0.3226 | 0.2903 | 9 | 50 | 10 | 26 | 20 | 34 | 42 | 7 | 12 |
| 47 | GestureMidAirD1 | 0.1692 | 0.3385 | 0.4323 | 2 | 24 | 20 | 6 | 18 | 38 | 39 | 5 | 17 |
| 48 | GestureMidAirD2 | 0.1462 | 0.2785 | 0.2846 | 1 | 18 | 18 | 8 | 9 | 28 | 22 | 4 | 15 |
| 49 | GestureMidAirD3 | 0.1154 | 0.1385 | 0.1538 | 2 | 5 | 13 | 3 | 5 | 15 | 8 | 3 | 12 |
| 50 | GesturePebbleZ1 | 0.3605 | 0.2942 | 0.2988 | 19 | 8 | 43 | 3 | 3 | 48 | 10 | 21 | 41 |
| 51 | GesturePebbleZ2 | 0.5190 | 0.5013 | 0.4899 | 13 | 10 | 69 | 5 | 4 | 74 | 8 | 13 | 69 |
| 52 | GunPoint | 0.5067 | 0.5933 | 0.5667 | 0 | 13 | 76 | 4 | 0 | 85 | 9 | 0 | 76 |
| 53 | GunPointAgeSpan | 0.5057 | 0.4937 | 0.5671 | 33 | 29 | 127 | 2 | 25 | 154 | 30 | 11 | 149 |
| 54 | GunPointMaleVersusFemale | 0.8544 | 0.8165 | 0.8165 | 12 | 0 | 258 | 0 | 0 | 258 | 0 | 12 | 258 |
| 55 | GunPointOldVersusYoung | 1.0000 | 0.9524 | 0.8279 | 15 | 0 | 300 | 48 | 9 | 252 | 0 | 54 | 261 |
| 56 | Ham | 0.5143 | 0.5524 | 0.5143 | 2 | 6 | 52 | 6 | 2 | 52 | 0 | 0 | 54 |
| 57 | HandOutlines | 0.6589 | 0.6097 | 0.5865 | 37 | 19 | 206 | 26 | 17 | 200 | 36 | 63 | 181 |
| 58 | Haptics | 0.2292 | 0.3325 | 0.3162 | 30 | 62 | 40 | 8 | 3 | 94 | 60 | 34 | 37 |
| 59 | Herring | 0.4062 | 0.4531 | 0.4375 | 1 | 4 | 25 | 2 | 1 | 27 | 3 | 1 | 25 |
| 60 | HouseTwenty | 0.8756 | 0.9076 | 0.9076 | 4 | 8 | 100 | 1 | 1 | 107 | 8 | 4 | 100 |
| 61 | InlineSkate | 0.1636 | 0.2302 | 0.2509 | 45 | 82 | 45 | 30 | 42 | 96 | 112 | 64 | 26 |
| 62 | InsectEPGRegularTrain | 1.0000 | 1.0000 | 1.0000 | 0 | 0 | 249 | 0 | 0 | 249 | 0 | 0 | 249 |
| 63 | InsectEPGSmallTrain | 0.5606 | 0.5261 | 0.3574 | 14 | 5 | 126 | 42 | 0 | 89 | 5 | 56 | 84 |
| 64 | InsectWingbeatSound | 0.1376 | 0.2653 | 0.2631 | 70 | 323 | 202 | 109 | 105 | 416 | 351 | 103 | 170 |
| 65 | ItalyPowerDemand | 0.8707 | 0.8688 | 0.8698 | 16 | 14 | 880 | 5 | 6 | 889 | 18 | 19 | 877 |
| 66 | LargeKitchenAppliances | 0.5973 | 0.5600 | 0.5552 | 19 | 5 | 205 | 4 | 2 | 206 | 6 | 22 | 202 |
| 67 | Lightning2 | 0.6557 | 0.6230 | 0.6066 | 4 | 2 | 36 | 2 | 1 | 36 | 2 | 5 | 35 |
| 68 | Lightning7 | 0.4247 | 0.4110 | 0.4164 | 6 | 5 | 25 | 0 | 0 | 30 | 5 | 6 | 25 |
| 69 | Mallat | 0.2473 | 0.4583 | 0.4443 | 199 | 693 | 381 | 56 | 23 | 1018 | 678 | 216 | 364 |
| 70 | Meat | 0.5700 | 0.6167 | 0.5833 | 0 | 3 | 34 | 2 | 0 | 35 | 1 | 0 | 34 |
| 71 | MedicalImages | 0.4997 | 0.5987 | 0.5966 | 23 | 98 | 357 | 20 | 19 | 435 | 105 | 32 | 348 |
| 72 | MelbournePedestrian | 0.6867 | 0.6840 | 0.6741 | 24 | 18 | 1658 | 31 | 7 | 1645 | 20 | 50 | 1632 |
| 73 | MiddlePhalanxOutlineAgeGroup | 0.1883 | 0.1623 | 0.1792 | 5 | 1 | 24 | 2 | 5 | 23 | 6 | 7 | 22 |
| 74 | MiddlePhalanxOutlineCorrect | 0.5704 | 0.5883 | 0.6014 | 0 | 5 | 166 | 1 | 5 | 170 | 10 | 1 | 165 |
| 75 | MiddlePhalanxTW | 0.5675 | 0.4727 | 0.4740 | 22 | 8 | 65 | 1 | 1 | 72 | 8 | 22 | 65 |
| 76 | MixedShapesRegularTrain | 0.6712 | 0.6626 | 0.6609 | 85 | 64 | 1543 | 26 | 22 | 1581 | 82 | 107 | 1521 |
| 77 | MixedShapesSmallTrain | 0.2851 | 0.3651 | 0.4049 | 15 | 209 | 676 | 28 | 125 | 857 | 332 | 42 | 649 |
| 78 | MoteStrain | 0.4609 | 0.4609 | 0.4609 | 0 | 0 | 577 | 0 | 0 | 577 | 0 | 0 | 577 |
| 79 | NonInvasiveFetalECGThorax1 | 0.0244 | 0.4365 | 0.5565 | 1 | 811 | 47 | 15 | 251 | 843 | 1047 | 1 | 47 |
| 80 | NonInvasiveFetalECGThorax2 | 0.0461 | 0.4595 | 0.5502 | 1 | 813 | 90 | 20 | 198 | 883 | 992 | 1 | 90 |
| 81 | OSULeaf | 0.7545 | 0.7248 | 0.7289 | 9 | 1 | 174 | 4 | 5 | 171 | 4 | 10 | 172 |
| 82 | OliveOil | 0.4000 | 0.4333 | 0.4000 | 1 | 2 | 11 | 1 | 0 | 12 | 2 | 2 | 10 |
| 83 | PLAID | 0.2127 | 0.1534 | 0.1844 | 48 | 16 | 66 | 23 | 40 | 59 | 49 | 64 | 50 |
| 84 | PhalangesOutlinesCorrect | 0.6410 | 0.6522 | 0.6643 | 13 | 23 | 537 | 5 | 15 | 555 | 35 | 15 | 535 |
| 85 | Phoneme | 0.2013 | 0.1972 | 0.1963 | 91 | 84 | 290 | 55 | 53 | 319 | 96 | 105 | 277 |
| 86 | PickupGestureWiimoteZ | 0.2600 | 0.5000 | 0.4920 | 1 | 13 | 12 | 2 | 2 | 23 | 14 | 2 | 11 |
| 87 | PigAirwayPressure | 0.1058 | 0.0894 | 0.1298 | 11 | 7 | 11 | 6 | 15 | 12 | 15 | 10 | 12 |
| 88 | PigArtPressure | 0.0625 | 0.2077 | 0.3038 | 7 | 37 | 6 | 5 | 25 | 38 | 52 | 2 | 11 |
| 89 | PigCVP | 0.0337 | 0.0692 | 0.1433 | 4 | 11 | 3 | 4 | 20 | 10 | 26 | 3 | 4 |
| 90 | Plane | 0.4952 | 0.4952 | 0.4952 | 0 | 0 | 52 | 0 | 0 | 52 | 0 | 0 | 52 |
| 91 | PowerCons | 0.6578 | 0.6278 | 0.6356 | 8 | 3 | 110 | 2 | 3 | 111 | 1 | 5 | 113 |
| 92 | ProximalPhalanxOutlineAgeGroup | 0.7463 | 0.7395 | 0.7171 | 7 | 5 | 146 | 5 | 0 | 147 | 5 | 11 | 142 |
| 93 | ProximalPhalanxOutlineCorrect | 0.6838 | 0.7223 | 0.7203 | 6 | 17 | 193 | 10 | 9 | 201 | 26 | 16 | 183 |
| 94 | ProximalPhalanxTW | 0.5463 | 0.6537 | 0.6839 | 0 | 22 | 112 | 5 | 11 | 129 | 33 | 5 | 107 |
| 95 | RefrigerationDevices | 0.3995 | 0.4107 | 0.3947 | 11 | 15 | 139 | 8 | 2 | 146 | 15 | 17 | 133 |
| 96 | Rock | 0.3000 | 0.3000 | 0.3800 | 1 | 1 | 14 | 1 | 5 | 14 | 6 | 2 | 13 |
| 97 | ScreenType | 0.3733 | 0.3483 | 0.3424 | 17 | 8 | 123 | 8 | 6 | 122 | 11 | 22 | 118 |
| 98 | SemgHandGenderCh2 | 0.6227 | 0.6137 | 0.6227 | 13 | 8 | 361 | 8 | 14 | 360 | 12 | 12 | 361 |
| 99 | SemgHandMovementCh2 | 0.3733 | 0.4120 | 0.4089 | 14 | 32 | 154 | 16 | 14 | 170 | 44 | 28 | 140 |
| 100 | SemgHandSubjectCh2 | 0.3542 | 0.3791 | 0.4107 | 14 | 25 | 145 | 4 | 18 | 167 | 39 | 14 | 145 |
| 101 | ShakeGestureWiimoteZ | 0.3400 | 0.5800 | 0.5200 | 1 | 13 | 16 | 3 | 0 | 26 | 10 | 1 | 16 |
| 102 | ShapeletSim | 0.5000 | 0.5000 | 0.5000 | 0 | 0 | 90 | 0 | 0 | 90 | 0 | 0 | 90 |
| 103 | ShapesAll | 0.0633 | 0.4350 | 0.4900 | 2 | 225 | 36 | 23 | 56 | 238 | 259 | 3 | 35 |
| 104 | SmallKitchenAppliances | 0.6971 | 0.6939 | 0.6901 | 15 | 14 | 246 | 5 | 3 | 256 | 15 | 17 | 244 |
| 105 | SmoothSubspace | 0.7067 | 0.7467 | 0.7387 | 6 | 3 | 103 | 2 | 1 | 110 | 10 | 5 | 101 |
| 106 | SonyAIBORobotSurface1 | 0.6922 | 0.7544 | 0.7837 | 36 | 73 | 380 | 10 | 28 | 443 | 81 | 26 | 390 |
| 107 | SonyAIBORobotSurface2 | 0.6031 | 0.4716 | 0.4690 | 171 | 46 | 404 | 25 | 23 | 424 | 49 | 177 | 398 |
| 108 | StarLightCurves | 0.9613 | 0.9538 | 0.9526 | 96 | 35 | 7821 | 19 | 9 | 7836 | 36 | 107 | 7810 |
| 109 | Strawberry | 0.6432 | 0.6389 | 0.6692 | 13 | 11 | 225 | 4 | 15 | 232 | 25 | 16 | 222 |
| 110 | SwedishLeaf | 0.4800 | 0.6861 | 0.6829 | 28 | 157 | 272 | 30 | 28 | 398 | 182 | 55 | 245 |
| 111 | Symbols | 0.3176 | 0.4253 | 0.4529 | 1 | 108 | 315 | 46 | 73 | 377 | 137 | 2 | 314 |
| 112 | SyntheticControl | 0.8233 | 0.7573 | 0.7513 | 36 | 16 | 211 | 8 | 6 | 220 | 21 | 43 | 204 |
| 113 | ToeSegmentation1 | 0.4175 | 0.4219 | 0.4167 | 3 | 4 | 92 | 2 | 1 | 94 | 3 | 3 | 92 |
| 114 | ToeSegmentation2 | 0.7692 | 0.7385 | 0.7846 | 5 | 1 | 95 | 1 | 7 | 95 | 2 | 0 | 100 |
| 115 | Trace | 0.3900 | 0.3800 | 0.4240 | 3 | 2 | 36 | 0 | 4 | 38 | 5 | 2 | 37 |
| 116 | TwoLeadECG | 0.8520 | 0.8820 | 0.8611 | 5 | 39 | 965 | 31 | 8 | 973 | 32 | 21 | 949 |
| 117 | TwoPatterns | 0.7693 | 0.7747 | 0.7833 | 122 | 144 | 2955 | 12 | 46 | 3087 | 165 | 109 | 2968 |
| 118 | UMD | 0.3333 | 0.3472 | 0.6042 | 0 | 2 | 48 | 3 | 40 | 47 | 41 | 2 | 46 |
| 119 | UWaveGestureLibraryAll | 0.4893 | 0.5109 | 0.5260 | 148 | 226 | 1604 | 103 | 157 | 1727 | 344 | 213 | 1540 |
| 120 | UWaveGestureLibraryX | 0.5281 | 0.5540 | 0.5658 | 72 | 165 | 1820 | 71 | 113 | 1914 | 251 | 117 | 1775 |
| 121 | UWaveGestureLibraryY | 0.4818 | 0.4797 | 0.4840 | 213 | 205 | 1513 | 90 | 105 | 1628 | 279 | 272 | 1454 |
| 122 | UWaveGestureLibraryZ | 0.5240 | 0.5247 | 0.5229 | 93 | 96 | 1784 | 94 | 88 | 1786 | 167 | 171 | 1706 |
| 123 | Wafer | 0.8921 | 0.8933 | 0.9037 | 0 | 8 | 5499 | 0 | 64 | 5507 | 72 | 0 | 5499 |
| 124 | Wine | 0.5000 | 0.5000 | 0.5000 | 0 | 0 | 27 | 0 | 0 | 27 | 0 | 0 | 27 |
| 125 | WordSynonyms | 0.2313 | 0.2906 | 0.2759 | 10 | 47 | 138 | 27 | 18 | 158 | 50 | 22 | 126 |
| 126 | Worms | 0.5325 | 0.4987 | 0.5896 | 4 | 1 | 37 | 0 | 7 | 38 | 7 | 3 | 38 |
| 127 | WormsTwoClass | 0.5091 | 0.4935 | 0.5117 | 4 | 3 | 35 | 1 | 2 | 37 | 5 | 5 | 34 |
| 128 | Yoga | 0.5177 | 0.5190 | 0.5189 | 33 | 37 | 1520 | 24 | 24 | 1533 | 27 | 23 | 1530 |
| | **Avg Value** | 0.4775 | 0.5204 | 0.5281 | 31 | 71 | 532 | 18 | 26 | 585 | 88 | 39 | 524 |

Table 22: Multi-scale analysis of w/o cross-scale fusion based on Scale-teaching using UCR 128 archive with Asym 40% noisy labels.

| ID | Dataset | fine | medium | coarse | a_t_b_f | a_f_b_t | a_t_b_t | b_t_c_f | b_f_c_t | b_t_c_t | c_t_a_f | c_f_a_t | c_t_a_t |
|----|---------|------|--------|--------|---------|---------|---------|---------|---------|---------|---------|---------|---------|
| 1 | ACSF1 | 0.6000 | 0.0100 | 0.3000 | 59 | 0 | 1 | 1 | 30 | 0 | 1 | 31 | 29 |
| 2 | Adiac | 0.0665 | 0.0588 | 0.0205 | 22 | 19 | 4 | 23 | 8 | 0 | 8 | 26 | 0 |
| 3 | AllGestureWiimoteX | 0.3863 | 0.0917 | 0.0923 | 247 | 41 | 23 | 35 | 35 | 29 | 47 | 253 | 17 |
| 4 | AllGestureWiimoteY | 0.4200 | 0.1354 | 0.0383 | 227 | 28 | 67 | 83 | 15 | 12 | 21 | 288 | 6 |
| 5 | AllGestureWiimoteZ | 0.3811 | 0.1000 | 0.0894 | 208 | 11 | 59 | 50 | 42 | 20 | 38 | 242 | 25 |
| 6 | ArrowHead | 0.5097 | 0.3029 | 0.3029 | 89 | 53 | 0 | 53 | 53 | 0 | 1 | 37 | 52 |
| 7 | BME | 0.4867 | 0.3333 | 0.1387 | 23 | 0 | 50 | 49 | 20 | 1 | 20 | 72 | 1 |
| 8 | Beef | 0.2333 | 0.2000 | 0.1867 | 7 | 6 | 0 | 6 | 6 | 0 | 1 | 2 | 5 |
| 9 | BeetleFly | 0.4000 | 0.5000 | 0.5000 | 7 | 9 | 1 | 0 | 0 | 10 | 9 | 7 | 1 |
| 10 | BirdChicken | 0.6000 | 0.5000 | 0.5000 | 8 | 6 | 4 | 0 | 0 | 10 | 6 | 8 | 4 |
| 11 | CBF | 0.5589 | 0.3311 | 0.3311 | 205 | 0 | 298 | 0 | 0 | 298 | 0 | 205 | 298 |
| 12 | Car | 0.4000 | 0.2533 | 0.1200 | 23 | 14 | 1 | 15 | 7 | 0 | 3 | 20 | 4 |
| 13 | Chinatown | 0.5275 | 0.7275 | 0.7246 | 4 | 73 | 178 | 2 | 1 | 249 | 74 | 6 | 176 |
| 14 | ChlorineConcentration | 0.4660 | 0.3694 | 0.5380 | 904 | 533 | 885 | 399 | 1046 | 1020 | 769 | 493 | 1297 |
| 15 | CinCECGTorso | 0.3338 | 0.1733 | 0.2696 | 455 | 233 | 6 | 233 | 366 | 6 | 43 | 131 | 329 |
| 16 | Coffee | 0.5357 | 0.4643 | 0.4643 | 15 | 13 | 0 | 0 | 0 | 13 | 13 | 15 | 0 |
| 17 | Computers | 0.5024 | 0.4840 | 0.5000 | 8 | 3 | 118 | 1 | 5 | 120 | 2 | 3 | 123 |
| 18 | CricketX | 0.2790 | 0.0974 | 0.1308 | 73 | 2 | 36 | 38 | 51 | 0 | 47 | 105 | 4 |
| 19 | CricketY | 0.2738 | 0.1077 | 0.1149 | 83 | 18 | 24 | 42 | 45 | 0 | 35 | 97 | 10 |
| 20 | CricketZ | 0.2144 | 0.1364 | 0.1590 | 61 | 30 | 23 | 53 | 62 | 0 | 31 | 52 | 31 |
| 21 | Crop | 0.4882 | 0.0619 | 0.0342 | 7332 | 170 | 870 | 1040 | 574 | 0 | 386 | 8014 | 188 |
| 22 | DiatomSizeReduction | 0.3007 | 0.0974 | 0.3497 | 92 | 30 | 0 | 13 | 90 | 17 | 107 | 92 | 0 |
| 23 | DistalPhalanxOutlineAgeGroup | 0.6619 | 0.3655 | 0.5324 | 58 | 17 | 34 | 42 | 65 | 9 | 20 | 38 | 54 |
| 24 | DistalPhalanxOutlineCorrect | 0.5333 | 0.5978 | 0.5833 | 38 | 56 | 109 | 9 | 5 | 156 | 61 | 47 | 100 |
| 25 | DistalPhalanxTW | 0.6763 | 0.2806 | 0.3525 | 55 | 0 | 39 | 0 | 10 | 39 | 0 | 45 | 49 |
| 26 | DodgerLoopDay | 0.2700 | 0.1075 | 0.1625 | 13 | 0 | 8 | 0 | 4 | 9 | 1 | 10 | 12 |
| 27 | DodgerLoopGame | 0.5290 | 0.4783 | 0.4783 | 72 | 65 | 1 | 0 | 0 | 66 | 65 | 72 | 1 |
| 28 | DodgerLoopWeekend | 0.7739 | 0.2609 | 0.2609 | 72 | 1 | 35 | 0 | 0 | 36 | 1 | 72 | 35 |
| 29 | ECG200 | 0.6700 | 0.6400 | 0.6400 | 15 | 12 | 52 | 0 | 0 | 64 | 12 | 15 | 52 |
| 30 | ECG5000 | 0.8888 | 0.1998 | 0.0191 | 3115 | 14 | 885 | 897 | 84 | 2 | 86 | 4000 | 0 |
| 31 | ECGFiveDays | 0.5970 | 0.5029 | 0.5029 | 283 | 202 | 231 | 0 | 0 | 433 | 202 | 283 | 231 |
| 32 | EOGHorizontalSignal | 0.3138 | 0.0961 | 0.1320 | 96 | 17 | 18 | 35 | 48 | 0 | 13 | 78 | 35 |
| 33 | EOGVerticalSignal | 0.2873 | 0.0746 | 0.0801 | 84 | 7 | 20 | 20 | 22 | 7 | 25 | 100 | 4 |
| 34 | Earthquakes | 0.7482 | 0.2518 | 0.2518 | 104 | 35 | 0 | 0 | 0 | 35 | 35 | 104 | 0 |
| 35 | ElectricDevices | 0.6009 | 0.0691 | 0.1126 | 4578 | 477 | 56 | 533 | 868 | 0 | 487 | 4253 | 381 |
| 36 | EthanolLevel | 0.2528 | 0.2520 | 0.2544 | 56 | 56 | 70 | 125 | 126 | 1 | 71 | 70 | 56 |
| 37 | FaceAll | 0.5088 | 0.0677 | 0.0182 | 795 | 50 | 64 | 114 | 31 | 0 | 5 | 834 | 26 |
| 38 | FaceFour | 0.4705 | 0.0000 | 0.2045 | 41 | 0 | 0 | 0 | 18 | 0 | 4 | 27 | 14 |
| 39 | FacesUCR | 0.4839 | 0.1740 | 0.0062 | 661 | 26 | 331 | 357 | 13 | 0 | 3 | 982 | 10 |
| 40 | FiftyWords | 0.2382 | 0.1253 | 0.0659 | 77 | 26 | 31 | 57 | 30 | 0 | 22 | 100 | 8 |
| 41 | Fish | 0.4583 | 0.0834 | 0.1314 | 66 | 0 | 15 | 15 | 23 | 0 | 19 | 76 | 4 |
| 42 | FordA | 0.8917 | 0.7006 | 0.4932 | 365 | 113 | 812 | 289 | 15 | 636 | 111 | 637 | 540 |
| 43 | FordB | 0.6407 | 0.5568 | 0.5049 | 112 | 44 | 407 | 51 | 9 | 400 | 53 | 163 | 356 |
| 44 | FreezerRegularTrain | 0.6327 | 0.6138 | 0.5000 | 59 | 5 | 1744 | 482 | 157 | 1268 | 160 | 538 | 1265 |
| 45 | FreezerSmallTrain | 0.5004 | 0.7038 | 0.5000 | 192 | 772 | 1234 | 773 | 192 | 1233 | 4 | 5 | 1421 |
| 46 | Fungi | 0.0968 | 0.0591 | 0.1290 | 7 | 0 | 11 | 0 | 13 | 11 | 13 | 7 | 11 |
| 47 | GestureMidAirD1 | 0.1769 | 0.0138 | 0.0385 | 23 | 2 | 0 | 1 | 4 | 1 | 2 | 20 | 3 |
| 48 | GestureMidAirD2 | 0.1708 | 0.0385 | 0.0462 | 17 | 0 | 5 | 5 | 6 | 0 | 5 | 21 | 1 |
| 49 | GestureMidAirD3 | 0.1385 | 0.0615 | 0.0431 | 15 | 5 | 3 | 5 | 3 | 3 | 4 | 16 | 2 |
| 50 | GesturePebbleZ1 | 0.4814 | 0.1802 | 0.1453 | 53 | 1 | 30 | 31 | 25 | 0 | 7 | 65 | 18 |
| 51 | GesturePebbleZ2 | 0.5089 | 0.0481 | 0.1861 | 73 | 0 | 7 | 7 | 29 | 0 | 10 | 61 | 19 |
| 52 | GunPoint | 0.6133 | 0.5973 | 0.4933 | 22 | 20 | 70 | 53 | 37 | 37 | 57 | 75 | 17 |
| 53 | GunPointAgeSpan | 0.3829 | 0.4342 | 0.4937 | 72 | 88 | 49 | 2 | 21 | 135 | 90 | 55 | 66 |
| 54 | GunPointMaleVersusFemale | 0.8576 | 0.7266 | 0.4747 | 43 | 2 | 228 | 80 | 0 | 150 | 2 | 123 | 148 |
| 55 | GunPointOldVersusYoung | 1.0000 | 0.6540 | 0.5238 | 109 | 0 | 206 | 41 | 0 | 165 | 0 | 150 | 165 |
| 56 | Ham | 0.4381 | 0.5143 | 0.5143 | 20 | 28 | 26 | 0 | 0 | 54 | 28 | 20 | 26 |
| 57 | HandOutlines | 0.6200 | 0.6546 | 0.6405 | 104 | 117 | 126 | 5 | 0 | 237 | 117 | 109 | 120 |
| 58 | Haptics | 0.2591 | 0.2292 | 0.2338 | 67 | 58 | 13 | 70 | 71 | 1 | 20 | 28 | 52 |
| 59 | Herring | 0.4688 | 0.4062 | 0.4062 | 5 | 1 | 25 | 0 | 0 | 26 | 1 | 5 | 25 |
| 60 | HouseTwenty | 0.6353 | 0.4034 | 0.4202 | 28 | 1 | 47 | 0 | 2 | 48 | 1 | 26 | 49 |
| 61 | InlineSkate | 0.2073 | 0.1691 | 0.1873 | 109 | 88 | 5 | 44 | 54 | 49 | 88 | 99 | 15 |
| 62 | InsectEPGRegularTrain | 1.0000 | 0.4739 | 0.6426 | 131 | 0 | 118 | 0 | 42 | 118 | 0 | 89 | 160 |
| 63 | InsectEPGSmallTrain | 0.5719 | 0.4739 | 0.1687 | 131 | 107 | 11 | 118 | 42 | 0 | 0 | 100 | 42 |
| 64 | InsectWingbeatSound | 0.1622 | 0.0962 | 0.1188 | 283 | 152 | 39 | 177 | 222 | 14 | 133 | 219 | 102 |
| 65 | ItalyPowerDemand | 0.9261 | 0.5015 | 0.5015 | 463 | 26 | 490 | 0 | 0 | 516 | 26 | 463 | 490 |
| 66 | LargeKitchenAppliances | 0.6016 | 0.3557 | 0.2224 | 105 | 12 | 121 | 118 | 68 | 15 | 25 | 167 | 58 |
| 67 | Lightning2 | 0.6393 | 0.5410 | 0.5410 | 15 | 9 | 24 | 0 | 0 | 33 | 9 | 15 | 24 |
| 68 | Lightning7 | 0.3836 | 0.2575 | 0.2466 | 23 | 14 | 5 | 3 | 2 | 16 | 15 | 25 | 3 |
| 69 | Mallat | 0.2479 | 0.1243 | 0.0198 | 581 | 291 | 1 | 276 | 30 | 16 | 46 | 581 | 0 |
| 70 | Meat | 0.6000 | 0.3333 | 0.3167 | 20 | 4 | 16 | 1 | 0 | 19 | 3 | 20 | 16 |
| 71 | MedicalImages | 0.5503 | 0.4468 | 0.2171 | 99 | 21 | 319 | 234 | 59 | 106 | 61 | 314 | 104 |
| 72 | MelbournePedestrian | 0.7768 | 0.0600 | 0.0199 | 1772 | 16 | 131 | 133 | 35 | 14 | 15 | 1869 | 34 |
| 73 | MiddlePhalanxOutlineAgeGroup | 0.6169 | 0.5714 | 0.2143 | 7 | 0 | 88 | 84 | 29 | 4 | 22 | 84 | 11 |
| 74 | MiddlePhalanxOutlineCorrect | 0.6220 | 0.5918 | 0.5704 | 17 | 8 | 164 | 7 | 1 | 165 | 9 | 24 | 157 |
| 75 | MiddlePhalanxTW | 0.5065 | 0.1494 | 0.1455 | 63 | 8 | 15 | 7 | 6 | 16 | 12 | 68 | 10 |
| 76 | MixedShapesRegularTrain | 0.7980 | 0.0851 | 0.1298 | 1735 | 6 | 200 | 206 | 315 | 0 | 7 | 1627 | 308 |
| 77 | MixedShapesSmallTrain | 0.4144 | 0.1195 | 0.2501 | 997 | 282 | 8 | 286 | 603 | 4 | 23 | 422 | 583 |
| 78 | MoteStrain | 0.4609 | 0.4612 | 0.4609 | 2 | 2 | 575 | 2 | 2 | 575 | 0 | 0 | 577 |
| 79 | NonInvasiveFetalECGThorax1 | 0.1186 | 0.0214 | 0.0422 | 192 | 1 | 41 | 42 | 83 | 0 | 40 | 190 | 43 |
| 80 | NonInvasiveFetalECGThorax2 | 0.1445 | 0.0261 | 0.0453 | 233 | 0 | 51 | 51 | 89 | 0 | 1 | 196 | 88 |
| 81 | OSULeaf | 0.6901 | 0.1694 | 0.0950 | 129 | 3 | 38 | 36 | 18 | 5 | 4 | 148 | 19 |
| 82 | OliveOil | 0.4000 | 0.4000 | 0.4000 | 0 | 0 | 12 | 0 | 0 | 12 | 0 | 0 | 12 |
| 83 | PLAID | 0.2134 | 0.1088 | 0.1233 | 74 | 18 | 41 | 13 | 21 | 46 | 27 | 76 | 39 |
| 84 | PhalangesOutlinesCorrect | 0.6452 | 0.6142 | 0.6131 | 145 | 119 | 408 | 1 | 0 | 526 | 119 | 146 | 407 |
| 85 | Phoneme | 0.1786 | 0.0188 | 0.0179 | 336 | 33 | 2 | 35 | 33 | 1 | 30 | 335 | 4 |
| 86 | PickupGestureWiimoteZ | 0.1320 | 0.0360 | 0.2680 | 7 | 2 | 0 | 2 | 13 | 0 | 9 | 2 | 4 |
| 87 | PigAirwayPressure | 0.0788 | 0.0096 | 0.0192 | 16 | 2 | 0 | 0 | 2 | 2 | 4 | 16 | 0 |
| 88 | PigArtPressure | 0.0894 | 0.0192 | 0.0192 | 15 | 0 | 4 | 4 | 4 | 0 | 0 | 15 | 4 |
| 89 | PigCVP | 0.0577 | 0.0192 | 0.0337 | 9 | 1 | 3 | 4 | 7 | 0 | 4 | 9 | 3 |
| 90 | Plane | 0.4952 | 0.2152 | 0.1733 | 40 | 11 | 12 | 11 | 7 | 12 | 6 | 40 | 12 |
| 91 | PowerCons | 0.5833 | 0.5000 | 0.5056 | 61 | 46 | 44 | 0 | 1 | 90 | 46 | 60 | 45 |
| 92 | ProximalPhalanxOutlineAgeGroup | 0.7844 | 0.4098 | 0.5122 | 98 | 21 | 63 | 81 | 102 | 3 | 19 | 65 | 96 |
| 93 | ProximalPhalanxOutlineCorrect | 0.7491 | 0.6838 | 0.6838 | 23 | 4 | 195 | 0 | 0 | 199 | 4 | 23 | 195 |
| 94 | ProximalPhalanxTW | 0.7873 | 0.1951 | 0.1863 | 130 | 9 | 31 | 33 | 31 | 7 | 9 | 132 | 29 |
| 95 | RefrigerationDevices | 0.4299 | 0.3333 | 0.3333 | 112 | 76 | 49 | 125 | 125 | 0 | 89 | 125 | 36 |
| 96 | Rock | 0.4800 | 0.3000 | 0.1800 | 15 | 6 | 9 | 14 | 8 | 1 | 7 | 22 | 2 |
| 97 | ScreenType | 0.3899 | 0.3333 | 0.3280 | 65 | 44 | 81 | 92 | 90 | 33 | 97 | 121 | 26 |
| 98 | SemgHandGenderCh2 | 0.6033 | 0.3500 | 0.3777 | 230 | 78 | 132 | 0 | 17 | 210 | 78 | 213 | 149 |
| 99 | SemgHandMovementCh2 | 0.3409 | 0.1702 | 0.1236 | 109 | 33 | 44 | 77 | 56 | 0 | 25 | 122 | 31 |
| 100 | SemgHandSubjectCh2 | 0.2933 | 0.1418 | 0.2298 | 125 | 57 | 7 | 64 | 103 | 0 | 42 | 71 | 61 |
| 101 | ShakeGestureWiimoteZ | 0.2200 | 0.1400 | 0.1400 | 6 | 2 | 5 | 7 | 7 | 0 | 7 | 11 | 0 |
| 102 | ShapeletSim | 0.5000 | 0.5000 | 0.5000 | 90 | 90 | 0 | 0 | 0 | 90 | 90 | 90 | 0 |
| 103 | ShapesAll | 0.2513 | 0.0280 | 0.0133 | 139 | 5 | 12 | 17 | 8 | 0 | 1 | 144 | 7 |
| 104 | SmallKitchenAppliances | 0.6816 | 0.3333 | 0.2912 | 213 | 82 | 43 | 94 | 78 | 31 | 47 | 194 | 62 |
| 105 | SmoothSubspace | 0.6133 | 0.3333 | 0.2933 | 59 | 17 | 33 | 46 | 40 | 4 | 23 | 71 | 21 |
| 106 | SonyAIBORobotSurface1 | 0.5990 | 0.4293 | 0.4293 | 132 | 30 | 228 | 0 | 0 | 258 | 30 | 132 | 228 |
| 107 | SonyAIBORobotSurface2 | 0.6631 | 0.6233 | 0.6170 | 171 | 511 | 83 | 9 | 3 | 585 | 514 | 180 | 74 |
| 108 | StarLightCurves | 0.9631 | 0.2799 | 0.5772 | 5707 | 80 | 2225 | 2305 | 4754 | 0 | 24 | 3202 | 4730 |
| 109 | Strawberry | 0.7957 | 0.6432 | 0.6432 | 110 | 54 | 184 | 0 | 0 | 238 | 54 | 110 | 184 |
| 110 | SwedishLeaf | 0.5763 | 0.0672 | 0.0304 | 318 | 0 | 42 | 42 | 19 | 0 | 1 | 342 | 18 |
| 111 | Symbols | 0.4750 | 0.1648 | 0.2553 | 316 | 0 | 157 | 73 | 163 | 91 | 1 | 220 | 253 |
| 112 | SyntheticControl | 0.9280 | 0.0713 | 0.1307 | 260 | 3 | 18 | 13 | 31 | 8 | 2 | 241 | 37 |
| 113 | ToeSegmentation1 | 0.4000 | 0.4605 | 0.4737 | 58 | 72 | 33 | 0 | 3 | 105 | 75 | 58 | 33 |
| 114 | ToeSegmentation2 | 0.8923 | 0.1846 | 0.1846 | 99 | 7 | 17 | 0 | 0 | 24 | 7 | 99 | 17 |
| 115 | Trace | 0.7300 | 0.1100 | 0.3000 | 62 | 0 | 11 | 11 | 30 | 0 | 3 | 46 | 27 |
| 116 | TwoLeadECG | 0.8386 | 0.5004 | 0.5004 | 408 | 23 | 547 | 0 | 0 | 570 | 23 | 408 | 547 |
| 117 | TwoPatterns | 0.7727 | 0.0150 | 0.2976 | 3084 | 53 | 7 | 58 | 1189 | 2 | 328 | 2228 | 863 |
| 118 | UMD | 0.3347 | 0.3333 | 0.3333 | 48 | 48 | 0 | 48 | 48 | 0 | 19 | 19 | 29 |
| 119 | UWaveGestureLibraryAll | 0.4907 | 0.1680 | 0.0878 | 1213 | 56 | 545 | 600 | 313 | 1 | 83 | 1527 | 231 |
| 120 | UWaveGestureLibraryX | 0.5207 | 0.1304 | 0.1235 | 1445 | 47 | 420 | 464 | 439 | 3 | 186 | 1609 | 256 |
| 121 | UWaveGestureLibraryY | 0.4638 | 0.1439 | 0.0749 | 1279 | 134 | 382 | 482 | 235 | 34 | 92 | 1485 | 176 |
| 122 | UWaveGestureLibraryZ | 0.5313 | 0.1359 | 0.1161 | 1468 | 51 | 436 | 482 | 481 | 5 | 126 | 1613 | 290 |
| 123 | Wafer | 0.8734 | 0.8569 | 0.8921 | 158 | 57 | 5225 | 73 | 290 | 5209 | 132 | 16 | 5367 |
| 124 | Wine | 0.5000 | 0.5000 | 0.5000 | 27 | 27 | 0 | 0 | 0 | 27 | 27 | 27 | 0 |
| 125 | WordSynonyms | 0.2442 | 0.1398 | 0.0737 | 70 | 3 | 86 | 88 | 46 | 1 | 39 | 148 | 8 |
| 126 | Worms | 0.6442 | 0.0909 | 0.2078 | 44 | 1 | 6 | 7 | 16 | 0 | 9 | 43 | 7 |
| 127 | WormsTwoClass | 0.6234 | 0.5714 | 0.5714 | 15 | 11 | 33 | 0 | 0 | 44 | 11 | 15 | 33 |
| 128 | Yoga | 0.5539 | 0.5357 | 0.5357 | 587 | 532 | 1075 | 0 | 0 | 1607 | 532 | 587 | 1075 |
| | **Avg Value** | 0.4961 | 0.2901 | 0.2887 | 376 | 56 | 196 | 106 | 116 | 147 | 56 | 365 | 207 |

