# OpenReview forum: "Scale-teaching: Robust Multi-scale Training for Time Series Classification with Noisy Labels"
_NeurIPS.cc/2023/Conference — NeurIPS 2023 poster_

### Official Review · Reviewer_oEKq · 2023-06-29

**Soundness:** 3 good
**Presentation:** 2 fair
**Contribution:** 3 good
**Rating:** 5
**Confidence:** 3

**Summary:**

This paper proposes a deep learning paradigm, namely scale-teaching, to cope with time series noisy labels, with designing a fine-to-coarse cross-scale fusion mechanism for learning discriminative patterns by using time series at different scales to train multiple DNNs simultaneously. Additionally, each network is trained in a cross-teaching manner by using complementary information from different scales to select small-loss samples as clean labels. Meanwhile, this paper introduces a multi-scale embedding graph learning method via label propagation to correct their labels from unselected large-loss samples.  Extensive experiments demonstrate the superior performance of the proposed method.

**Strengths:**

1. The research problem is interesting and important.
2. The idea is simple and effective.
3. The paper is well written.
4. Extensive experiments with good results.


**Weaknesses:**

1. The meanings of some equations or symbols are not clear, like the operator || in Eq. (2).
2. The problem definition in Label Correction for Eq. (5) is difficult to understand.
3. The proposed method is relatively complex, and its procedure is not clear, including the Algorithm 1.
4. In Figure 2, it is better to introduce the flowchart of the proposed method.

**Questions:**

1. In Eq. (2), how the operator || to operate the r_i^k, v_i^k and so on?
2. How to obtain the Eq. (5)?

**Limitations:**

The proposed Scale-teaching paradigm only utilizes three scales, and its training time is increasing with the increase of the number of scales.

---

> ### Author Rebuttal · Authors · 2023-08-06
>
> **W1 & Q1**:  The meanings of the operator || in Eq. (2), and how the operator || to operate the r_i^k, v_i^k.
>
> **A1**: Thanks for your comment. The operator || in Equation (2) means to concatenate two vectors into a new vector. For example, vector **a** = [0,1], vector **b** = [2,3], then **a** || **b** = [0,1,2,3].
> r_i^k represents the single-scale k embedding, and v_i^k is the multi-scale ([1, 2, …, k], where k is downsampling scale) fusion embedding via Eq.(2). Particularly if k = 1, we use the single-scale for classification training and let v_i_k = r_i_k. If k > 1, v_i_k = f(r_i_k || v_i_k-t||(r_i_k - v_i_k-t)||(r_i_k * v_i_k-t)), where v_i_k-t denotes the multi-scale ([1,2, …, k-t], where k-t is downsampling scale) fusion embedding. And the source code for the implementation of Eq (2) is given below:
>
> ```
>  f = nn.Sequential(
>             nn.Linear(128 * 4, 256),
>             nn.BatchNorm1d(256),
>             nn.ReLU(inplace=True),
>             nn.Linear(256, 128),
>         )  ### the function of f(.)
>
> fea_concat = torch.cat((embed_x, scale_x, (embed_x - scale_x), (embed_x * scale_x)), dim=1) ### the function of operator ||. where embed_x is r_i^k, scale is v_i_k-t.
>
> v_i_k = f(fea_concat)
> ```
>
>
> **W2 & Q2**:  The problem definition in Label Correction for Eq. (5) and how to obtain the Eq. (5)?
>
> **A2**: Thanks for your comment. In Eq.(5), **Y** contains labeled and unlabeled samples, Y_ij=1 if x_i is labeled as y_i =j, otherwise Y_ij = 0. Label propagation theory utilizes the diffusion process [R1] to get the pseudo-labels of unlabeled samples via labeled samples and nearest-neighbor **Q**.
> Hence, we defined the sequence {F_t} to denote the diffusion process and described it as Eq.(5).
> When *t = 0*, *F_0* is initialized as **Y**. Further, when timestamp *t* in Eq. (5) is large, through the label information of **Y** and the graph edge weight of nearest-neighbor **Q**, the predicted pseudo-label *F_t* quality is better. Conveniently, we can know the sequence {F_t} has a closed-form solution [R1], which can be defined as Eq. (6).
>
> [R1] Learning with local and global consistency. NIPS, 2003.
>
>
> **W3 & W4**: The proposed method is relatively complex, and it is better to introduce the flowchart of the proposed method in Figure 2.
>
> **A3**: Thanks for your comment. The proposed Scale-teaching paradigm includes two processes: (i) clean label selection; (i) multi-scale graph embedding learning.
> For clean label selection,
> networks A, B, and C use clean labels provided by cross-teaching (A$\rightarrow$B, B$\rightarrow$C, C$\rightarrow$A) to guide each other's classification training.
> For multi-scale graph embedding learning, Scale-teaching first performs cross-scale fusion from fine to coarse (A$\rightarrow$B, B$\rightarrow$C).
> Then, we utilize pseudo labels obtained via multi-scale embeddings graph learning as corrected labels for those time series not selected as clean labelled data.
> In addition, we give a pseudo-code for the Scale-teaching paradigm in Figure 2:
>
> ```
> Step 1: Obtain single-scale embeddings r_A, r_B, r_C
> r_A = encoder_1(x_A)
> r_B = encoder_2(x_B)
> r_c = encoder_3(x_C)
> Step 2: Obtain cross-scale embeddings v_A, v_B, v_C
> v_A = r_A
> v_B = Eq.5(r_B, v_A)
> v_C = Eq.5(r_C, v_B)
> Step 3: Obtain clean labels y_A, y_B, y_C for cross-teaching training
> y_A = classifier_3(v_C)  ## Using small loss criterion
> y_B = classifier_1(v_A)  ## Using small loss criterion
> y_C = classifier_2(v_B)  ## Using small loss criterion
> Step 4: Obtain corrected labels yc_A, yc_B, yc_C for classification training
> yc_A = Eq.6(v_A, y_A)
> yc_B = Eq.6(v_B, y_B)
> yc_C = Eq.6(v_C, y_C)
> Step 5: Overall training
> Update encoder_1 and classifier_1 via cross-entropy loss(v_A, yc_A)
> Update encoder_2 and classifier_2 via cross-entropy loss(v_B, yc_B)
> Update encoder_3 and classifier_3 via cross-entropy loss(v_C, yc_C)
> ```

---

> > ### Comment · Reviewer_oEKq · 2023-08-16
> >
> > Thank you for the response of all the authors. Most of my concerns have been addressed. However, it is necessary to clearly present them in the next version.

---

> > > ### Author Response · Authors · 2023-08-17
> > >
> > > Thank you very much for your reply.

---

### Official Review · Reviewer_L5T6 · 2023-07-07

**Soundness:** 3 good
**Presentation:** 4 excellent
**Contribution:** 2 fair
**Rating:** 4
**Confidence:** 4

**Summary:**

This paper a paradigm to improve the small-loss criterion for time series noisy labels, which mainly addresses the problem that external noises easily distort time series’ discriminative patterns. The motivation is clear, and experiments have shown good results. However, the reviewer is still concerned about some issues.

**Strengths:**

1. DNNs with different random initializations have high consistency in classification results for clean labeled samples in the early training period; while there is disagreement in the classification of noisy labeled samples, the designation of cross-scale fusion using inputs with different scales is lighter than other methods using different networks like co-teaching which are time-consuming.

**Weaknesses:**

1. Although the proposed method is useful, the network is similar to multi-scale feature learning like FPN and the label propagation part is also not novel.
2. The way of downsampling methods needs more comparison.
3. Momentum updating is also useful for other methods, although the goal of using it in this paper is reasonable, and scale-teaching achieves SOTA even w.o. momentum updating. But it is still an unfair comparison.


**Questions:**

1. What is the fundamental difference between the proposed method with FPN and traditional label propagation is graph learning?

**Limitations:**

The authors seem to discuss some limitations of the method in terms of computation efficiency. The reviewer suggests more discussion in energy consumption for a more general usage.

---

> ### Author Rebuttal · Authors · 2023-08-06
>
> **W1 & Q1**:  The network is similar to multi-scale feature learning like FPN and the label propagation part is also not novel. Also, what is the fundamental difference between the proposed method with FPN and traditional label propagation in graph learning?
>
> **A1**:  Thanks for your comment.  Based on the keyword *FPN*, we searched for the article [R1], which is expected to be the FPN. First, we summarize the differences between Scale-teaching and FPN as follows:
> ```
> (1) FPN and Scale-teaching solve different problems. FPN is used for image object detection, while Scale-teaching is used for time series classification with noisy labels. In addition, time series data is a collection of data points sorted in chronological order, which is different from image data. In general, it is hard to use the image model directly for time series.
> (2) FPN takes a single-scale image as input and uses a top-down pyramid network to learn the multi-scale embeddings of the image. Scale-teaching takes different scale sequences of the time series as input and uses the cross-scale fusion mechanism to learn the multi-scale embeddings of the time series.
> (3) For different scale embeddings, FPN uses shared classifiers/regressors for object detection. In contrast, Scale-teaching employs multiple classifiers with different initializations for classification training, thus exploiting the divergence information of the DNNs on the noisy labels to obtain more robust multi-scale embeddings.
> (4) FPN uses one randomly initialized DNN to learn multi-scale embeddings of an image. Scale-teaching uses several different randomly initialized DNNs to learn multi-scale embeddings of a time series, and uses selected clean labels for cross-teaching classification training.
> ```
>
> For the differences between Scale-teaching and Traditional Label Propagation (TLP) in graph learning, we summarize as follows:
> ```
> (1) TLP is generally used for semi-supervised learning. Scale-teaching solves the time-series noise label learning problem with label propagation.
> (2) TLP generally uses single-scale data for graph construction. Scale-teaching utilizes learned multi-scale embeddings for graph learning.
> (3) TLP requires explicitly correctly labeled data for modeling. Scale-teaching utilizes selected small-loss data as correctly labeled data for modeling.
> ```
>
> [R1] Feature Pyramid Networks for Object Detection. CVPR, 2017.
>
>
> **W2**: The way of downsampling methods needs more comparison.
>
> **A2**: Thanks for your comment. In our supplementary material, we have provided the comparison analysis of downsampling methods in Section E.
>
> Firstly, we analyze the impact of the downsampling scale sequence list. From Figure 2 in Section E, we find that Scale-teaching has the highest classification accuracy using four different scales for training, indicating that more input scales do not necessarily improve the classification performance. In addition, using three or four scales of sequences can effectively improve the classification performance of Scale-teaching compared with using two different scales.
>
> Then, we analyze the impact of input scales of sequence order. As shown in Figure 3 in Section E, We can find that the classification performance of finer-to-coarser is better overall, which is due to its ability to use a single fine-scale sequence with an excellent classification performance from the beginning to promote the classification performance of multi-scale fusion embeddings gradually.
>
>
> **W3**: Momentum updating is also useful for other methods.
>
> **A3**: Thanks for your comment.
> Existing noise-label learning methods rarely consider utilizing robust feature representations to cope with noisy labels.
> Specifically, existing noise-label learning methods mainly design robust learning paradigms at the loss level, while Scale-teaching focuses on handling noise labels with robust multi-scale feature representations.
> Therefore, Scale-teaching designs the cross-scale mechanism and combines it with momentum updating to learn robust multi-scale feature representations.

---

### Official Review · Reviewer_QaFC · 2023-07-07

**Soundness:** 3 good
**Presentation:** 3 good
**Contribution:** 2 fair
**Rating:** 5
**Confidence:** 3

**Summary:**

The paper proposes a time series classification algorithm that works across multiple scales with a noise-robust loss function. The authors claim improvements due to the multi-scale network architecture and the objective function capable to handle label noise on a variety of datasets in UCR128 and beyond.

**Strengths:**

The paper is mostly well organized. Statistical tests were performed in all experiments, and the paper has a good mix of summary statistics for large evaluation runs (like in Table 1), and more qualitative results showing properties of the algorithm (e.g. the training dynamics in Figure 5). The authors also perform ablation studies and attempt to test the effect of different components of their algorithm.

**Weaknesses:**

- presentation can be generally improved. In section 3, it is not fully clear which components are preliminaries and present in existing work, and which are the new algorithmic components of the new approach. According to title and abstract, these should be the multi-scale and label noise corrections parts of the model. Maybe the headings could be further adapted to make this clear (e.g. right now, it is not clear why "cross-scale fusion" mostly addresses the multi-scale aspect, while "multi-scale embedding graph learning" mostly covers the label noise robustness part).

- The figure caption for Figure 2 could be a bit longer and actually explain the figure content.

**Minor**:

- typo: emebeddings (l. 56)
- the definition of the considered noise model comes too late and should be directly mentioned in Section 3.
- Table 1 has a lot of numbers. Given that you ran a statistical test, consider to e.g. gray out the non-signficant results for easier readability.
- sec 4.5 has a few typos (missing spaces, etc)

**Questions:**

- The work seems large incremental, given that the method (3.4) is composed of techniques used in [44], [40], and label propagation. Could you highlight the main conceptual novelty of your algorithm compared to previous approaches?
- What is the test performed in Table 1? I would suggest to include this directly into the caption.
- l. 327 in the conclusion claims "... can utilize the multi-scale properties of time series to effectively handle noisy labels". Where is this particular point validated, in your opinion?
- How are hyperparameters selected? Can this be made more explicit in the main paper?
- What are additional limitations of the assumed noise model and correction mechanism?

---

> ### Author Rebuttal · Authors · 2023-08-07
>
> **Q1**:  The work seems large incremental, given that the method (3.4) is composed of techniques used in [40], [44], and label propagation. Could you highlight the main conceptual novelty of your algorithm compared to previous approaches?
>
> **A1**: Thanks for your question.
> Literature [40] uses original sample features for graph construction, thus for semi-supervised learning. Literature [44] designs an unsupervised reconstruction loss for semi-supervised learning using a momentum update strategy.
> In contrast to [40], we construct the graph using learned multi-scale embeddings. Unlike [44], we perform graph construction using momentum-updated multi-scale embeddings to exploit robust multi-scale embeddings for noisy label correction.
> Compared to existing noisy label learning methods, our conceptual innovation is to exploit the complementary information between sequences of different scales and design a cross-scale fusion mechanism for learning more robust multi-scale embeddings. Furthermore, we use multi-scale embeddings to introduce graph learning based on label propagation for noisy label correction.
>
> [40] Label propagation through linear neighborhoods. ICML, 2006.
>
> [44] Temporal ensembling for semi-supervised learning. arXiv, 2016.
>
> **Q2**: What is the test performed in Table 1? I would suggest to include this directly into the caption.
>
> **A2**: Thanks for your question and suggestion.
> The results in Table 1 show the test classification accuracy statistics of different baselines on four individual large datasets, the UCR 128 archive and the UEA 30 archive.
> The Avg. Rank indicates the average ranking of the corresponding baseline's test classification accuracy among all baselines for each benchmark. In the next version, we will update the description in the caption of Table 1.
>
> **Q3**: l. 327 in the conclusion claims "... can utilize the multi-scale properties of time series to effectively handle noisy labels". Where is this particular point validated, in your opinion?
>
>  **A3**: Thanks for your question.
> In Section 4.3, we analyze the cross-scale fusion mechanism for Scale-teaching. Figure 3 in Section 4.3 illustrates that the classification results of different scale sequences exhibit evident complementary information. Specifically, Figure 3 (c) demonstrates that Scale-teaching (larger shaded circle area) has significantly more samples with correct category predictions for the classifiers corresponding to the coarse and fine scales on the UCR 128 archive than w/o cross-scale fusion (smaller shaded circle area). Additionally, Scale-teaching on the coarse $\rightarrow$ fine scale also shows a significantly larger cross-circle area (525) compared to w/o cross-scale fusion (207). These results indicate that Scale-teaching can improve the model's classification performance by effectively utilizing the complementary information between different scales in the presence of noisy labels.
>
> Furthermore, the t-SNE visualization in Figure 4 reveals that the multi-scale embeddings learned by Scale-teaching are more discriminative across categories. Hence, we claim that Scale-teaching can effectively leverage the multi-scale properties of time series to handle noisy labels.
>
>
> **Q4**: How are hyperparameters selected? Can this be made more explicit in the main paper?
>
> **A4**: Thanks for your question.
> Our experiment contains 162 datasets. It would be time-consuming to perform hyperparameter selection for each dataset. Therefore, the hyperparameters of Scale-teaching are not carefully tuned for each dataset, and most of the hyperparameters are set based on the default hyperparameters of related works.
> The learning rate and epoch are set based on the parameters of existing noise-label learning methods, such as FINE and CULCU.
> $\alpha$ in Eq. 3, $\sigma$ in Eq. 4 and $\beta$ in Eq. 5 are set based on the default hyperparameters of related label propagation works.
> $e_{warm}$ is based on FINE settings. $e_{update}$, $\gamma$ and batch size are based on manual empirical settings without specific hyperparameter analysis.
> The largest neighbor $K$ is set based on human experience, and we had a simple test on several datasets, and found that a larger value of $K$ does not improve the classification performance, but instead increases the running time of the model. In our next release, we will explain in detail the basis for choosing each hyperparameter.
>
> **Q5**: What are additional limitations of the assumed noise model and correction mechanism?
>
> **A5**: Thanks for your question.
> It may be difficult for our proposed method to achieve good classification performance with noise ratios greater than 50%. In particular, the literature [R3] suggests that the small-loss criterion selects clean labeled samples that still contain many noisy labels when the noise ratio is greater than 50%. Therefore, existing work on label-noise learning based on the small-loss criterion (e.g., Co-teaching, CULCU) has experimentally set label noise ratios of less than 50%.
>
> [R3] Towards understanding deep learning from noisy labels with small-loss criterion. IJCAI, 2021.
>
> **W1**:  presentation can be generally improved.
>
> **A6**: Thanks for your comment.
> We think your suggestions are excellent. In our next release, we will update the title and organization of Section 3 based on your suggestions.
>
> **W2**:  The figure caption for Figure 2 could be a bit longer and actually explain the figure content.
>
> **A7**: Thanks for your comment. We will update the caption of Figure 2 in the next version.
>
> **W3**:  Minor: typo: emebeddings (l. 56), etc.
>
> **A8**: Thanks for your comment. We will fix the above issues in the next version.

---

### Official Review · Reviewer_Rktg · 2023-07-08

**Soundness:** 2 fair
**Presentation:** 3 good
**Contribution:** 2 fair
**Rating:** 5
**Confidence:** 4

**Summary:**

This paper presents a "scale-teaching" framework aimed at addressing label noise in time series classification tasks. The proposed approach involves multiple encoders to extract multi-scale time series features. These features are then concatenated for the final loss calculation, enabling the selection of clean samples. The identified clean and noisy samples are further processed using a label propagation algorithm, incorporating graph learning techniques to correct the noisy labels. The entire dataset is subsequently trained using cross-entropy as the loss function. Experimental evaluations are conducted on the UCR 128 archive and UEA 30 archive datasets, both containing synthetic label noise.

**Strengths:**

1. While the problem of learning with noisy labels (LNL) has been extensively studied in the domain of image classification, the exploration of label noise in time series classification remains relatively limited in current literature. This work makes significant contributions by conducting extensive experiments to investigate how the LNL experience gained in image classification can be effectively generalized to time series classification.

2. The utilization of multi-scale operations is particularly noteworthy in the context of time series classification with label noise, considering the inherent differences in modality between images and time series data.

3. The paper demonstrates a clear and coherent writing style, ensuring ease of comprehension for readers. Additionally, the release of the accompanying code promotes reproducibility, while the appendix provides comprehensive details on the conducted experiments.


**Weaknesses:**

1. While I appreciate the contribution made by generalizing learning with noisy labels (LNL) from image classification to time series classification, the level of technical novelty in this work appears somewhat limited. Specifically, the utilization of a multi-scale framework has already been explored in time series forecasting tasks, and the application of a semi-supervised approach to enhance performance is not novel in the context of general LNL tasks [R1].

2. The paper lacks sufficient ablation studies to thoroughly examine the individual components of the proposed method. For instance, in order to demonstrate the efficacy of the multi-scale framework in sample selection, it would be valuable for the authors to conduct controlled experiments using only one scale. While accuracy is reported in Table 3, it is not necessarily the most appropriate criterion for evaluating sample selection. Therefore, I suggest that the authors also report selection quality metrics such as precision, recall, and F1 score. Additionally, it would be worthwhile to explore alternative label correction techniques, such as pseudo-labeling or solely utilizing clean samples for training, to ascertain their competitiveness in comparison to the label propagation method employed in the paper.

3. It would be insightful to evaluate the performance of the "scale-teaching" framework on datasets that do not contain label noise. I noticed that such experiments were conducted in [R2], and I encourage the authors to perform similar settings to provide a comprehensive analysis in their work.


[R1] DivideMix: Learning with Noisy Labels as Semi-supervised Learning

[R2 ]Estimating the Electrical Power Output of Industrial Devices with End-to-End Time-Series Classification in the Presence of Label Noise

**Questions:**

See **weaknesses**

**Limitations:**

This work extensively addresses its limitations, and based on the information provided, it appears that the paper does not pose potential negative societal impacts.

---

> ### Author Rebuttal · Authors · 2023-08-07
>
> **W1**:  The utilization of a multi-scale framework has already been explored in time series forecasting tasks, and the application of a semi-supervised approach to enhance performance is not novel in the context of general LNL tasks [R1].
>
> **A1**: Thanks for your comment.
> In Section 2, we have discussed existing work related to multi-scale time series modeling. In particular, the multi-scale properties of time series have been widely used for various time series downstream tasks, such as time series prediction,  classification, and anomaly detection tasks.
> The above studies show that exploiting the multi-scale properties of time series can improve the performance of downstream tasks. Nevertheless, the above studies do not consider how to utilize the multi-scale properties of time series for label-noise learning. Unlike the above work, we employ multiple DNNs with the same structure to learn discriminative patterns of the time series at different scales separately, and design a cross-scale fusion strategy to obtain robust embeddings for handling noisy labels.
>
> DivideMix is a classical work for image label-noise learning that combines MixMatch semi-supervised learning method for noisy label classification. In contrast, our proposed Scale-teaching paradigm utilizes learned multi-scale embeddings to cope with time series noisy labels.
> In other words, Scale-teaching focuses on utilizing robust feature representations for label-noise learning rather than a semi-supervised learning paradigm for label-noise learning.
> In particular, Scale-teaching utilizes its learned multi-scale embeddings to select more reliable clean labels. Further, Scale-teaching uses the selected clean labels in conjunction with multi-scale graph learning to achieve noisy label correction. Hence, the focus of Scale-teaching exploits the multi-scale nature of time series for label-noise learning, and does not rely exclusively on the semi-supervised learning paradigm for noisy label classification.
>
> [R1] DivideMix: Learning with Noisy Labels as Semi-supervised Learning
>
>
> **W2**: (i) It would be valuable for the authors to conduct controlled experiments using only one scale. (ii) I suggest that the authors also report selection quality metrics such as precision, recall, and F1 score. (iii) Additionally, it would be worthwhile to explore alternative label correction techniques, such as pseudo-labeling or solely utilizing clean samples for training.
>
> **A2**: Thanks for your comment.
>
> (i) From Table 3 in Section 4.5, we have conducted controlled experiments using only one scale. For details, please refer to the *only single scale* method in Table 3.
>
> (ii) That's a good suggestion. Following [R2], we select averaged F1-score on the test set as a new metric for ablation analysis. Like Table 3 in Section 4.5, we give the corresponding test classification F1-score (%) as follows:
>
> |Dataset | HAR |  | UniMiB-SHAR |  |
> |:---:|:---:|:---:|:---:|:---:|
> | Method  | Sym 50% | Asym 40% | Sym 50% | Asym 40% |
> | Scale-teaching | **90.05** | **89.14** | **77.56** | **65.89** |
> | w/o cross-scale fusion | 88.16 (-1.89) | 87.05 (-2.09) | 68.23 (-9.33) | 57.76 (-8.13) |
> | only single scale | 87.56 (-2.49) | 86.75 (-2.39) | 66.87 (-10.69) | 54.12 (-11.77) |
> | w/o graph learning | 87.79 (-2.26) | 87.41 (-1.73) | 74.62 (-2.94) | 63.15 (-2.74) |
> | w/o moment | 89.34 (-0.71) | 88.27 (-0.87) | 76.67 (-0.89) | 64.92 (-0.97) |
> | w/o dynamic threshold | 88.93 (-1.12) | 88.29 (-0.85) | 73.11 (-4.45) | 64.76 (-1.17) |
>
> (iii) In Section 3.4, we claim that $\mathcal{F}$ in Eq. (7) is the estimated pseudo-labels which inevitably contain some incorrect labels. To address the above issue, we utilize a dynamic threshold pseudo-labeling strategy to promote label correction performance.
> Further, we add an ablation study method (w/o dynamic threshold) in Table 3. The result shows that the dynamic threshold pseudo-labelling strategy can improve classification performance.
> For solely utilizing clean samples for training, please refer to the ablation study method named *w/o graph learning* in Table 3.
>
> [R2] Estimating the Electrical Power Output of Industrial Devices with End-to-End Time-Series Classification in the Presence of Label Noise
>
> **W3**: It would be insightful to evaluate the performance of the "scale-teaching" framework on datasets that do not contain label noise.
>
> **A3**: Thanks for your comment.
> We select the four individual large datasets without noisy labels for evaluation. The detailed test classification accuracy (Acc, %) and F1-score (F1, %) are shown as follows:
>
> |  | Standard  |  | Mixup  |  | Co-teaching  |  | FINE |  |  SREA  |  | SELC  |  | CULCU  |  | Scale-teaching |  |
> |:---:|:---:|:---:|:---:|:---:|:---:|:---:|:---:|:---:|:---:|:---:|:---:|:---:|:---:|:---:|:---:|:---:|
> | Dataset | Acc | F1 | Acc | F1 | Acc | F1 | Acc | F1 | Acc | F1 | Acc | F1 | Acc | F1 | Acc | F1 |
> | HAR | 93.29 | 93.27 | **95.42** | **95.39** | 93.77 | 93.75 | 93.13 | 93.19 | 93.02 | 92.91 | 93.76 | 93.71 | 94.75 | 94.72 | 94.72 | 94.18 |
> | UniMiB-SHAR | 89.14 | 86.37 | _84.84_ | _80.17_ | 88.24 | 84.43 | 88.14 | 84.03 | _65.51_ | _66.54_ | 89.28 | 89.19 | 89.46 | 86.45 | **93.61** | **93.62** |
> | FD-A | 99.93 | 99.93 | 99.91 | 99.91 | **99.96** | **99.96** | _68.22_ | _64.05_ | _90.25_ | _90.14_ | 99.82 | 99.82 | 99.95 | 99.95 | **99.96** | **99.96** |
> | Sleep-EDF | 84.93 | 81.99 | 84.67 | 82.11 | 85.37 | 82.52 | 84.62 | 83.07 | _79.42_ | _77.67_ | 84.82 | 82.17 | **85.54** | 83.26 | 85.34 | **84.76** |
>
> As shown in the above table, Scale-teaching's classification performance is still better than most baselines. In addition, SREA employs an unsupervised time series reconstruction loss as an auxiliary task, which reduces the model's classification performance without noisy labels.

---

> > ### Comment · Reviewer_Rktg · 2023-08-18
> > **Thanks for the reply**
> >
> > I appreciate the comprehensive explanations and meticulously designed experiments provided. These experiments further bolster the validity of the proposed method, and I strongly encourage the authors to incorporate these new experiments into the next version. However, my initial concern remains unresolved. While I understand the explanations presented by the authors in the Rebuttal, I still find the method lacking in significant novelty. Beyond the aspect of multi-scale feature modeling, the other techniques appear to be direct extensions from the existing LNL/semi-supervised practices. I don't intend to disregard the potential utility of these techniques, but I'm eager to see a more profound analysis comparing the conventional noisy label setting in image classification with the noisy label setting in time series classification. In the current submission, these analyses seem insufficiently comprehensive or deeply explored.
> >
> > Overall, I consider this paper to be on the borderline. However, I have raised my score to 5 as my other concerns have been effectively addressed by the authors.

---

> > > ### Author Response · Authors · 2023-08-18
> > >
> > > Thank you very much for your reply and for acknowledging our work. In our next version, we plan to include an exposition on the distinctions between time series noise labeling and conventional noisy label setting in image classification.

---

### Comment · Area_Chair_KyBT · 2023-08-12
**Discussion period**

Dear Reviewers,

I would like to express my sincere gratitude for your thorough examination of this paper. Now that the authors have provided their rebuttal, I kindly ask you to evaluate whether their response sufficiently addresses the concerns you have raised. Should you require any additional information or have further questions, please feel free to request clarification directly from the authors. Your insights and contributions to this process are greatly appreciated!

Best regards,

AC

---

### Comment · Area_Chair_KyBT · 2023-08-18
**Awaiting Your Feedback on Authors' Rebuttal**

Dear Reviewer L5T6 and Reviewer QaFC,

Thank you for your hard work. The Author-Reviewer discussion ends on August 21. The authors and I are eager to learn whether their responses have adequately addressed your concerns. You are encouraged to directly reply to the authors' rebuttal.

Please note that this is a public thread. If you prefer to reply to me individually, please use the internal discussion thread.

Kind Regards,

AC

---

### Decision · Program_Chairs · 2023-09-21

**Decision:**

Accept (poster)

**Comment:**

This is a borderline paper. The core idea is to learn robust multi-scale representations, select small loss examples as confident examples, and build graphs to correct labels of large-loss examples, handling the issue of noisy labels in time series.

Most reviewers acknowledge the empirical performance of this method. However, reviewers have concerns about its technical contribution.

AC has carefully checked the content of this paper, all comments from reviewers, and the authors' responses. After consideration, AC recommends accepting this paper. The reasons are as follows:
- The paper skillfully melds diverse techniques like co-training, multi-scale feature learning, graph learning. This combination, driven by a clear rationale and motivation, surpasses the performance of  prevalent baselines. Harmoniously integrating these methods, especially when each brings its own complexity, is not easy.
- It targets a practical and significant problem: reducing the negative impact of time series label errors.  Targeting this issue holds  implications for real-world applications, ensuring more reliable data interpretations and outcomes.

Seen through this lens, the method proposed by this paper represents a novel combination of well-known techniques and contributes a solution to a important problem. With the proposed post-acceptance modifications (clarifying definitions, distinguishing from existing methods, and additional experiments), the paper can be further enhanced, making it a valuable contribution to the conference.

For the camera-ready version, the authors should integrate the reviewers' feedback and implement the suggested changes. Additionally, for the readers' convenience, it would be beneficial if the related work section paints a broader overview of methods for learning with noisy labels, e.g., also covering semi-supervised-based methods and transition-matrix-based methods.